# The Space Omics and Medical Atlas (SOMA) and international astronaut biobank

Spaceflight induces molecular, cellular and physiological shifts in astronauts and poses myriad biomedical challenges to the human body, which are becoming increasingly relevant as more humans venture into space[1–6]. Yet current frameworks for aerospace medicine are nascent and lag far behind advancements in precision medicine on Earth, underscoring the need for rapid development of space medicine databases, tools and protocols. Here we present the Space Omics and Medical Atlas (SOMA), an integrated data and sample repository for clinical, cellular and multi-omic research profiles from a diverse range of missions, including the NASA Twins Study[7], JAXA CFE study[8,9], SpaceX Inspiration4 crew[10–12], Axiom and Polaris. The SOMA resource represents a more than tenfold increase in publicly available human space omics data, with matched samples available from the Cornell Aerospace Medicine Biobank. The Atlas includes extensive molecular and physiological profiles encompassing genomics, epigenomics, transcriptomics, proteomics, metabolomics and microbiome datasets, which reveal some consistent features across missions, including cytokine shifts, telomere elongation and gene expression changes, as well as mission-specific molecular responses and links to orthologous, tissue-specific mouse datasets. Leveraging the datasets, tools and resources in SOMA can help to accelerate precision aerospace medicine, bringing needed health monitoring, risk mitigation and countermeasure data for upcoming lunar, Mars and exploration-class missions.

With an exponential increase in launches since 2019, space is rapidly becoming more accessible[1]. Multiple commercial and state-sponsored groups are developing roadmaps to construct space stations, moon bases, Mars colonies and other permanent establishments beyond Earth[1]. Although innovation across the aerospace sector makes these ambitions technologically achievable, the biomedical challenges for crews in these extraplanetary habitats still need to be addressed, as humans did not evolve to survive in such extreme environments. The clinical consequences of this evolutionary mismatch of spaceflight exposure and adaptation have revealed a plethora of challenges to long-term space habitation, including a loss in bone density and muscle mass[2], spaceflight-associated neuro-ocular syndrome[3], perturbed immune function[4] and spaceflight anaemia[5]. These physiological changes during spaceflight appear to imprint on the health status in humans; a chief example of this long-term effect is the increased risk of cardiovascular pathology observed in astronauts compared with age-matched controls[6].

Before long-term space habitation is feasible, these biomedical challenges must be understood and mitigated. However, the aetiologies driving them are not understood, with the low number of astronauts yielding limited opportunities for in-depth biomolecular characterization. For example, two of the largest multi-omic studies to date have been the NASA Twins Study, which has published an in-depth molecular and cognitive profile of a single astronaut[7], and the Japanese Aerospace Exploration Agency (JAXA) Cell-Free Epigenome (CFE) project, which has profiled cell-free DNA (cfDNA) and cfRNA in six astronauts[8,9]. Thus, achieving statistical power requires integrating data from other cohorts, such as the MARROW study, and other missions[5,10]. Furthermore, these analyses are complicated by the substantial variation in physiological responses to spaceflight among astronauts. Therefore, there is a need for increasingly large, detailed multi-omic profiles of astronauts to characterize the diversity of physiological shifts as a function of spaceflight[11].

To achieve this end, we have leveraged the burgeoning commercial spaceflight industry. With the launch of SpaceX's 2021 Inspiration4 (I4) mission, a cohort of all-civilian astronauts successfully completed a high elevation (585 km), 3-day orbital mission within a SpaceX Dragon capsule. Using recently developed protocols, the crew participated in a range of biospecimen collections before, during and after their mission[12]. We used the I4 biospecimens to deeply profile the effect of the stressors of spaceflight (for example, microgravity and space radiation) on crew physiology and health. We also compared these results with previous missions and control datasets, creating the largest-to-date molecular atlas of the effect of spaceflight on the human body, encompassing almost 3,000 samples and over 75 billion sequenced nucleic acids. Collectively, these resources are referred to as SOMA, and the samples are linked to a Cornell Aerospace Medicine Biobank (CAMbank) that stores viably frozen specimens for future, additional analyses.

As with Earth-based cohorts[13], these accessible data — when profiled and aggregated at scale — will enable the development of both personalized and general medical guidance for astronauts[14]. A large group of subject matter experts in artificial intelligence has recently released recommendations focused on the importance of generating and archiving space data into the NASA Open Science Data Repository

(OSDR)[15,16] to enable autonomous and intelligent precision space health systems, and to monitor, aggregate and assess biomedical statuses for future deep space missions[17]. In addition, the study of the parallels between the physiological effects of spaceflight and ageing, chronic disease and immune system disorders using omics data can pave the way for therapeutics applicable to conditions on Earth.

Here we present a detailed guide to the SOMA resource, which includes the 2,911 samples collected during the I4 mission[11,12], as well as spatial transcriptomics data, long-read profiles of astronaut RNA, microbiome data, exosome profiles and in-depth immune diversity maps. Additional spaceflight data were annotated and compiled into the SOMA portal to help contextualize gene, protein or metabolite dynamics, including data from the NASA Twins Study[7], JAXA's CFE mission[8,9], single-cell RNA sequencing (scRNA-seq) data after simulated microgravity on peripheral blood mononuclear cells (PBMCs)[18], and rat or mouse spaceflight data matched to human orthologues. In addition to rigorous dataset annotations, we detail (1) a comparison of conclusions on NASA Twins Study and flight dynamics comparing short-duration and long-duration missions, (2) cell-type-specific responses to spaceflight previously undocumented in astronauts, (3) cfRNA expression profiles showing haematological responses during recovery from spaceflight, and (4) additional analyses on individual responses to spaceflight from proteomic, transcriptomic and microbiome data. Data and samples generated in this study are available through SOMA (https://soma.weill.cornell.edu), NASA OSDR (https://osdr.nasa.gov/bio/) and CAMbank (https://cambank.weill.cornell.edu/), which offer an unprecedented view of the multi-system omics changes before, during and after spaceflight.

## Comprehensive astronaut data resource

To generate a comprehensive profile of the physiological changes of the I4 crew (29, 38, 42 and 51 years of age), 13 biospecimen sample types were collected and processed, including whole blood, serum, PBMCs, plasma, extracellular vesicles and particles (EVPs) derived from plasma, dried blood spots, oral swabs, nasal swabs, skin biopsies, skin swabs, capsule (SpaceX Dragon) swabs, urine and stool specimens[12]. After collection, samples were subject to a battery of multi-omic assays, including clinical (CLIA) whole-genome sequencing, a clonal haematopoiesis panel, direct RNA-seq (dRNA-seq), single-nucleus RNA-seq (snRNA-seq), single-nucleus assay for transposase-accessible chromatin with sequencing (snATAC-seq), single-cell B cell repertoire (BCR) and T cell repertoire (TCR) V(D)J sequencing, untargeted plasma proteomics (liquid chromatography–tandem mass spectrometry), untargeted plasma metabolomics, cfDNA sequencing, cfRNA, metagenomics, metatranscriptomics and spatially resolved transcriptomics. In addition, chemokine, cytokine and cardiovascular biomarkers were quantified, and a CLIA lab (Quest Diagnostics) was used to perform a complete blood count and comprehensive metabolic panel (Fig. 1a). Datasets were generated across ten timepoints: three pre-flight (L−92, L−44 and L−3), three in-flight (flight day 1 (FD1), FD2 and FD3), one immediately post-flight (R+1) and three recovery (R+45, R+82 and R+194) spanning 289 days (Fig. 1b). Assays were performed on all crew members unless otherwise noted (Fig. 1b and Supplementary Table 1).

A total of 2,911 samples were banked, with 1,194 samples processed for sequencing, imaging and biochemical analysis (Supplementary Table 1). These results and assays subsume and expand on work and protocols from previous missions, including the JAXA CFE study, the NASA Twins Study and some NASA astronauts (Fig. 1c). This latter category spans studies primarily from the International Space Station (ISS) that lack certain metadata, primarily duration spent in space and launch dates, to maintain astronaut anonymity. These studies include chemokine/cytokine biomarker panels (n = 46 astronauts), comprehensive metabolic panels, telomere length quantitative PCR (qPCR) and ISS-surface metagenomic profiling (Fig. 1c).

The SOMA resources were first compared with the NASA OSDR database, which contains all publicly accessible human omics data from spaceflight and ground analogue studies. OSDR hosts 76 human omics studies, of which 11 are from human primary cells exposed to spaceflight. The other studies encompassed cell line and ground studies, including high-altitude studies (Extended Data Fig. 1a and Supplementary Table 2), which were all merged with the SOMA dataset. Once merged, the total number of sequenced nucleic acid molecules from this study represents a more than tenfold increase in the total amount of human omics data in the OSDR (Extended Data Fig. 1b), across all spaceflight studies, ground studies, cell line and primary cell experiments (Extended Data Fig. 1 and Supplementary Tables 2 and 3).

The data from the missions were then divided into three analysis timeframes: (1) flight profiles, (2) recovery profiles and (3) longitudinal profiles (Extended Data Fig. 1c). Flight profiles reveal the most immediate effect of spaceflight, recovery profiles catalogue changes that occur after return to Earth, and the longitudinal profiles identify changes that have not returned to baseline after returning to Earth. We focused on several outputs for the resource, including first calculating differentially expressed genes (DEGs) for (1) PBMC snRNA-seq, (2) whole-blood dRNA-seq, (3) skin spatially resolved transcriptomics, and (4) cfRNA. We also mapped differentially methylated genes from whole-blood dRNA-seq, differentially accessible regions from PBMC snATAC-seq, isotype identification from TCR and BCR V(D)J sequencing, differentially abundant proteins from plasma and EVP proteomics, differential metabolites from liquid chromatography–mass spectrometry metabolomics, and microbial differentials from metagenomic and metatranscriptomic assays (Extended Data Fig. 2), with all raw and processed data annotated in the OSDR (Supplementary Table 4).

## I4 reproduces NASA Twins Study

Telomere elongation has been previously described in three astronauts who stayed for 6 months to 1 year aboard the ISS[7,19,20], but it was unclear how quickly such a phenotype appeared in astronauts. The average telomere length in all I4 crew members increased during spaceflight (17–22% longer), and this trend was statistically significant (mixed-effects linear model P = 0.0048; Fig. 2a). This finding is particularly notable, given the shorter mission duration (3 days total) and higher elevation of the I4 mission than the ISS studies, indicating that telomere length dynamics respond much more rapidly to spaceflight than previously observed.

We then compared the DEGs and cytokine changes from the Twins Study with those observed in the snRNA-seq data from the I4 mission, as well as compared with the expected DEGs of the assay from replicate negative control donor PBMCs (see Methods). The cross-mission DEG comparison highlighted a consistent response between both types of T cells, including CD4[+] and CD8[+] markers (552 and 608 DEGs, respectively, both <2.2 × 10$^{-16}$), across both sorted T cells or single-cell annotated cells (Fig. 2b). Conversely, B cells were less responsive to spaceflight, as expected from previous work in the Twins Study[7], which showed B cells as either not significant or less responsive to spaceflight. For the four overlapping cytokines measured in the Twins Study with our panel, we found significant increases in three: IL-6 (P = 0.014), IL-10 (P = 0.021) and CCL2 (P = 0.040) (Fig. 2c); these cytokines also showed changes and similar increases in other long-duration (more than 6 months) crews (Fig. 2c). However, we also ran a differential analysis of all cytokines detected on the I4 mission, to detect any differences from the Twins Study. Indeed, the levels of BDNF showed a statistically significant decrease (P = 0.00011, q = 0.0153), and IL-19 levels showed a statistically significant increase (P = 0.00015, q = 0.0153) during the post-flight (R+1) timepoint that returned to baseline during recovery (R+45 and R+82; Fig. 2d).

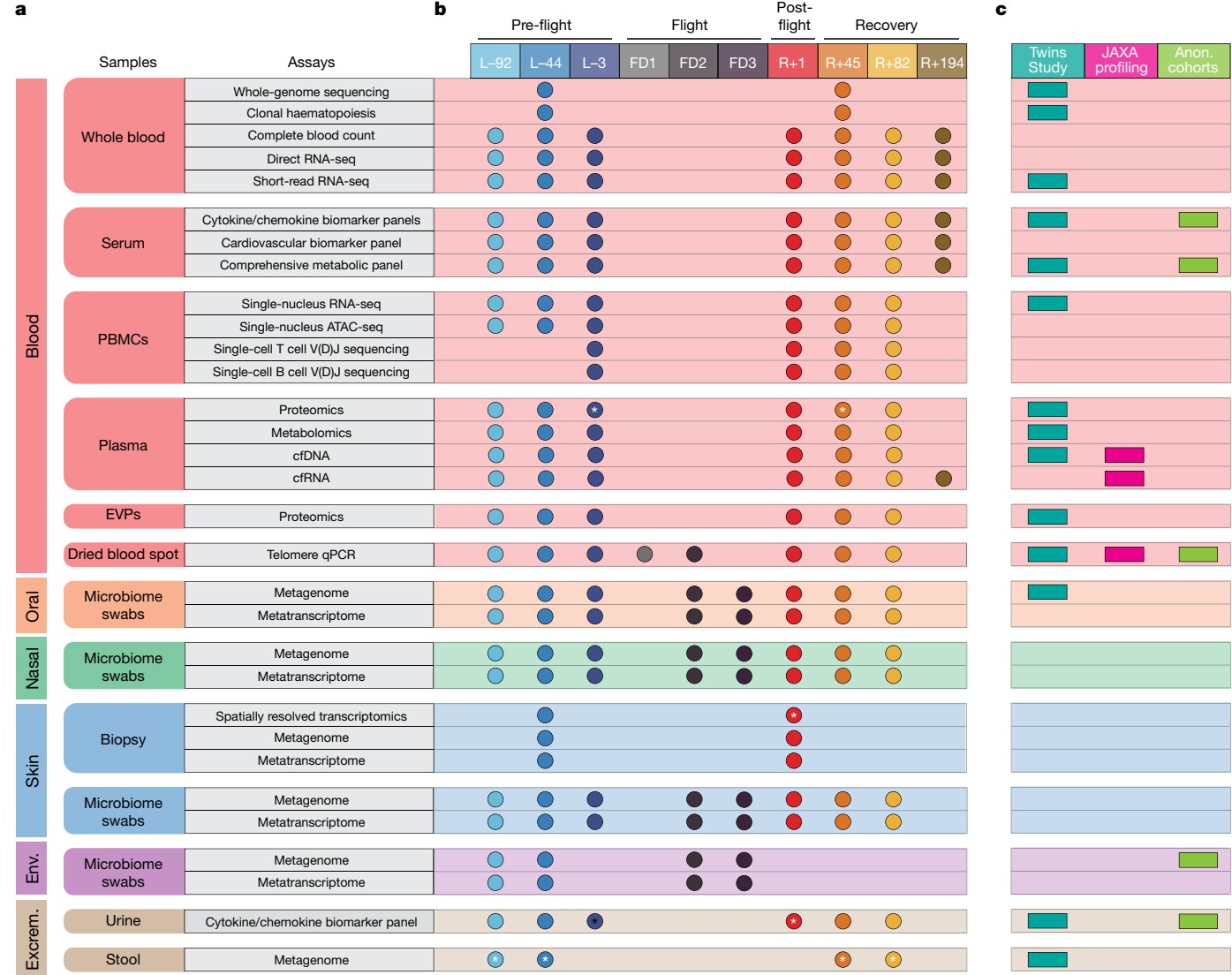

**Fig. 1 | Compendium of astronaut omic data and time-series analysis paradigms. a**, Omics and biochemical assays were performed on blood (whole blood, serum, PBMCs, plasma, plasma-derived EVPs and dried blood spots), oral (microbiome swabs), nasal (microbiome swabs), skin (biopsy and microbiome swabs), environmental (env.; microbiome swabs) and excrement (excrem.; urine and stool) samples. **b**, The timepoints of this study are separated into four different categories: pre-flight (L−92, L−44 and L−3), in-flight (FD1, FD2 and FD3), post-flight (R+1) and recovery (R+45, R+82 and R+194). The coloured circles indicate which assay was performed at each timepoint. Assays were performed on all crew members, unless denoted with an asterisk. **c**, Indicator for which assay types have been previously performed in spaceflight studies, broken down by the NASA Twins Study, JAXA studies and anonymized NASA cohort studies. Anon., anonymized.

## Distinct RNA fingerprints of spaceflight

Beyond recapitulating known biomarkers of spaceflight, the atlas integrated newer assays that were not available in previous missions, with a particular emphasis on RNA profiling. The first novel assay was spatially resolved transcriptomics on skin biopsies, which were obtained from all crew members during one pre-flight timepoint (L−44) and the day after landing back on Earth (R+1). The 4-mm biopsies were stained with markers for DNA, PanCK, FAP and α-SMA and then processed with the NanoString/Bruker GeoMx Digital Spatial Profiler, where regions of interest were selected based on the tissue structures identified by the fluorescence staining (Extended Data Fig. 3a). After filtering out outliers, the RNA counts were used for downstream data analysis, generating 95 regions of interest across four skin compartments: outer epidermis, inner epidermis, outer dermis and vasculature (Extended Data Fig. 3b,c). This analysis revealed a distinct set of DEGs, including JAK–STAT signalling, and melanocyte signatures (Fig. 3).

A second RNA assay for spaceflight integrated into SOMA was cfRNA profiling, which has recently been established as a dynamic tool for mapping temporal alterations in cfRNA composition and cell lysis[21]. However, bulk cfRNA had not been utilized to measure the response of spaceflight until the JAXA CFE study[8,9] and the I4 mission[12]. Using principal component analysis, we identified a distinct separation in cfRNA profiles pre-flight versus post-flight and recovery for I4, suggesting a systemic physiological shift probably induced by space travel (Extended Data Fig. 3d). This was further reflected in the differential abundance of cfRNA genes across various timepoints, revealing specific patterns of noncoding expression (Extended Data Fig. 3e) and RNA types (Extended Data Fig. 3f) that correspond with the spaceflight timeline. The cell-type proportions inferred from the cfRNA profiles also exhibited spaceflight-associated variation over time and showed variation distinct from a set of healthy blood donor controls (n = 35; Fig. 3a and Supplementary Table 5). Cell types that showed significant post-flight shifts in proportion included hepatocytes,

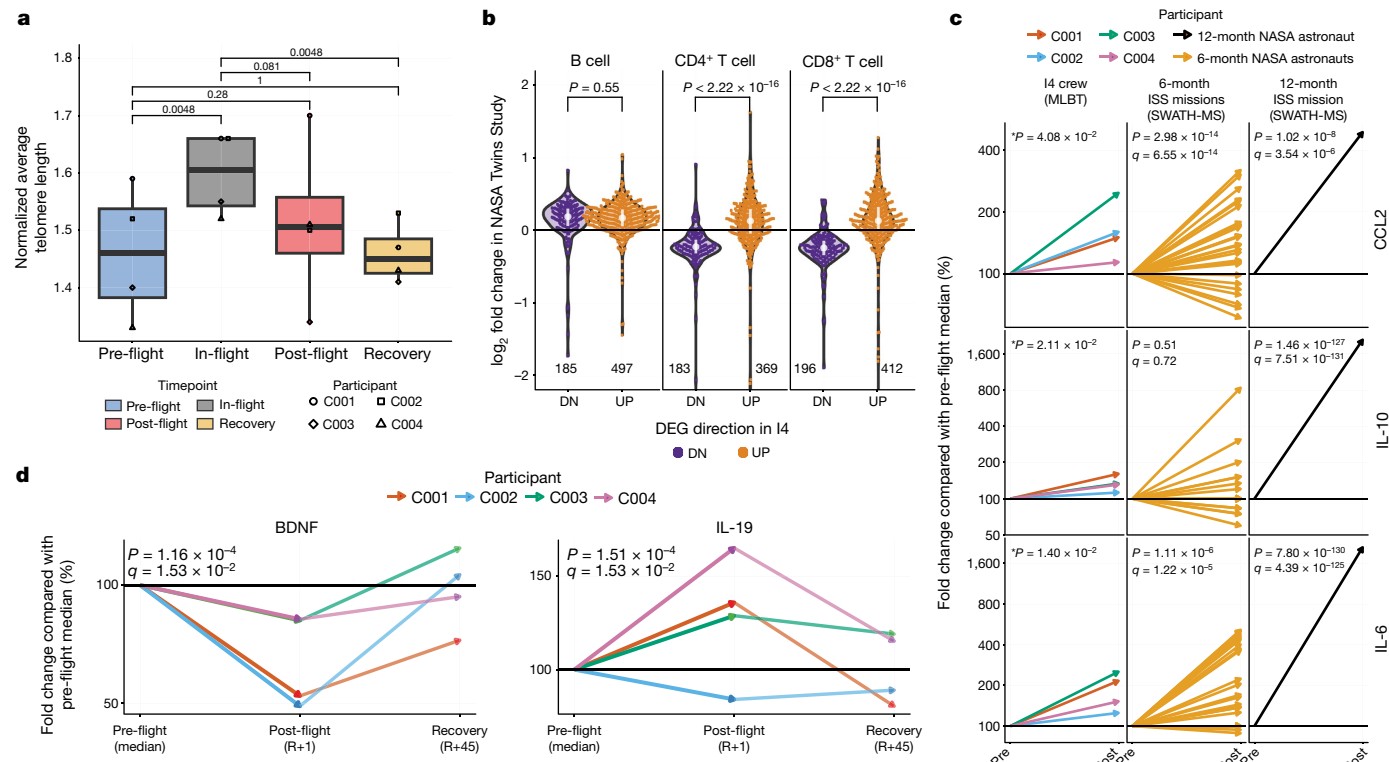

**Fig. 2 | Telomere and cytokine Twins Study comparison. a**, Normalized average telomere lengths for I4 crew members, pre-flight, during flight and post-flight, determined by qPCR analyses of blood (DNA) collected on dried blood spot (DBS) cards ($n = 32$ samples for 4 independent participants across 8 timepoints). Two-sided $P$ values were derived using a mixed-effects linear model that incorporated fixed effects for different timepoints (pre-flight, in-flight, post-flight and recovery) and random effects to account for variations among participants. The centre of the boxplots represents the median, the box hinges encompass the first and third quartiles, and the whiskers extend to the smallest and largest values no further than $1.5 \times$ the interquartile range (IQR) away from the hinges. **b**, Changes in downregulated (DN; purple) and upregulated (UP; orange) gene expression $\log_2$ fold-change directionality post-flight from the Twins Study versus I4 in CD19 B cells, CD4+ T cells and CD8+ T cells (statistical significance was determined by a two-sided Wilcoxon rank-sum test).

The number of genes is shown below the violin plots. The centre white dot represents the median, and the white line shows the range of the first and third quartiles. **c**, Relative cytokine/chemokine abundance pre-flight, post-flight and during recovery in the I4 crew versus the NASA Twins Study and anonymized NASA astronaut cohorts for CCL2, IL-10 and IL-6. MLBT, multiplexing LASER bead technology. Pre, pre-flight median; Post, post-flight (R+1). **d**, Relative abundance of BDNF and IL-19 pre-flight, post-flight and during recovery in the I4 crew. In panels **c** and **d**, the two-sided $P$ values and adjusted $q$ values were derived using a mixed-effects model that incorporated fixed effects for different timepoints (pre-flight, in-flight, post-flight and recovery) and random effects to account for variations among participants, except in the Twins Study, which had a single participant ($n = 1$). $P$ values with an asterisk have a $q > 0.05$ after multiple correction testing.

kidney endothelial cells, haematopoietic stem cells and melanocytes (Fig. 3b and Supplementary Table 5). Of note, the melanocyte cell proportions that demonstrated significant changes post-flight (Fig. 3c) were also found in the spatial skin transcriptomics data, providing additional evidence of adaptive skin responses to the space environment.

A third novel RNA method applied to these spaceflight samples focused on RNA isoforms and RNA modifications (epitranscriptome), through dRNA-seq on the Oxford Nanopore Technologies PromethION and deep RNA-seq (more than 400 million reads per sample) on the Ultima Genomics UG100. These data quantified genes that were differentially expressed and displayed differential $N^6$-methyladenosine methylation (Extended Data Fig. 4a,b), or both, and were analysed for enriched Gene Ontology pathways. We identified a set of sites (set M-I; Extended Data Fig. 4b) that undergoes hypomethylation during recovery, another set (set M-II) that is detectably hypermethylated from spaceflight, and a set (set M-III) that exhibits novel hypermethylation during recovery and longitudinally as well. The common pathways in all three sets (Extended Data Fig. 4b) showed evidence of radiation and telomere response, including 'TSAI response to radiation therapy' and 'Wiemann telomere shortening'[22]. In addition, the set of downregulated pathways after landing (recovery) was distinct, including genes associated with breathing regulation (for example, $CO_2$ and $O_2$

take-up and release by erythrocytes), which matches those pathways associated with crew response ranges (below, Fig. 5). Although further studies are needed to validate and delineate the potential mechanisms of these RNA dynamics (cfRNA, spatial and RNA modifications), these pathways suggest a potential relationship between RNA expression and methylation in regulating haematological and dermatological functions upon return to Earth.

## Gene regulatory changes during recovery

Leveraging the time-series data, we next analysed PBMC gene expression from snRNA-seq to discern whether unique DEGs are present at each timepoint. We examined DEGs from immediately post-flight (FP1) through the recovery profiles (RP1 and RP2) to observe how gene expression profiles are re-established after spaceflight. The number of DEGs was used to quantify the severity and recovery of crew response, and we also compared identified DEGs from a negative control group, including two healthy donors and read-depth- and cell-count-matched permutation group (Extended Data Fig. 5), with an average of more than 700 cells per crew member, per timepoint, per cell type. The DEG count (adjusted $P < 0.05$, |$\log_2$ fold change (FC)| > 0.5) decreased from FP1 to RP1 in CD16+ monocytes, dendritic cells, natural killer cells,

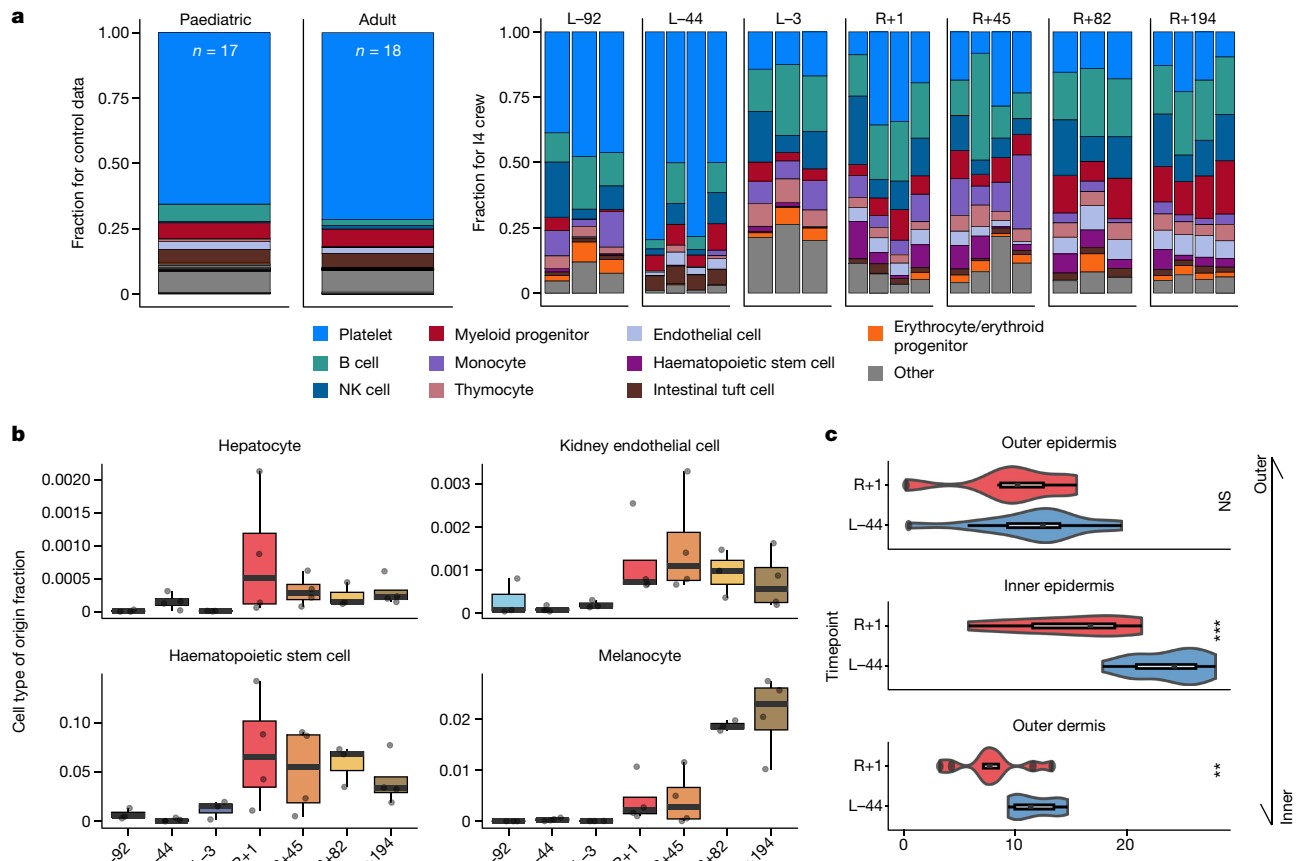

**Fig. 3 | Body-wide tissue stress map with cfRNA. a**, Cell-type deconvolution using Bayes Prism with Tabula Sapiens as a reference. Top ten cell types by average fraction across all samples with all remaining cell types summed together as 'other'. **b**, Cell type of origin for hepatocytes, endothelial cells, haematopoietic stem cells and melanocytes, which all show increased abundance during post-flight and recovery timepoints. **c**, Cell proportion changes in different layers of the skin from spatially resolved transcriptomics on skin biopsies. Predicted melanocyte abundance changes are significant in the inner epidermal and outer dermal skin compartments. In panels **b** and **c**, $n = 4$ independent participants across 7 timepoints. The centre of the boxplots represent the median, the box hinges encompass the first and third quartiles, and the whiskers extend to the smallest and largest values no further than $1.5 \times$ IQR away from the hinges. NS, not significant; **$P \leq 0.01$; ***$P \leq 0.001$.

'other' T cells (but not CD4+ or CD8+), uncategorized ('other') cells and in pseudobulk (calculated from additive counts across cell types to represent PBMCs). However, the DEG count increased in B cells, CD4+ T cells, CD8+ T cells and CD14+ monocytes (Fig. 4a); by RP2, all cell types had lower DEG counts than at FP1 and began to approach the expected noise range of single-cell DEGs (Fig. 4a).

As DEG counts were higher at RP1 than at FP1 for several cell types, we hypothesized the introduction of distinct DEGs after landing back on Earth. First, we calculated the percentage of DEGs unique to FP1 at both RP1 and RP2, focusing on cell-type-specific DEGs (Fig. 4b). By LP2 (L−92, L−44 and L−3 versus R+82), we observed that nearly all differential gene expression was driven by the same set of genes that were differentially expressed at FP1, with the exception of CD14+ monocytes, which had a unique longitudinal profile not strongly connected with the DEG responses in other cells. With the exception of CD14+ monocytes, the RP1 profile is distinct in the abundance of DEGs not present in FP1, but this perturbation almost entirely (more than 95% of genes) disappeared after flight (RP2 and LP2). From the DEGs shared with FP1, we observed that gene expression directionality reversed for nearly all DEGs between FP1 and each recovery profile (Fig. 4c), indicating a return to baseline for those DEGs. The exception to this were two genes from the CD14+ monocyte population − AHR (log2FC of 0.742 (FP1) and log2FC of 0.808 (RP1)) and PELI1 (log2FC of 0.539 (FP1) and log2FC of 0.560 (RP1); Fig. 4c) − in which there was a positive log2FC expression at both FP1 and RP1.

From the DEGs that were unique to FP1, we next quantified the uniformity of the pathway enrichment between two PBMC lineages: T cells (CD4+ and CD8+) and monocytes (CD14+ and CD16+; Supplementary Table 6). Of note, 211 (70.8%) genes were unique to individual cell types and 87 (29.2%) were shared between two or more cell types (Extended Data Fig. 6a). To identify enriched pathways among the gene set for each cell type, overrepresentation analysis was performed on Kyoto Encyclopedia of Genes and Genomes (KEGG) pathways (Supplementary Table 6). Even though 70.8% of genes were unique to each cell type, their response was more convergent; unique pathways only ranged from 4.2% (CD4+ T cells) to 42.5% (CD14+ monocytes; Extended Data Fig. 6b). Yet some pathways were cell-type specific, such as the circadian entrainment pathway in CD14+ monocytes, with genes *GNG2*, *GNAS*, *PRKCB*, *CREB1* and *CAMK2D* driving the overrepresentation from the gene set for this pathway (Fig. 4d). In addition, our data suggest that overrepresented pathways are sometimes more associated with cell lineage (for example, all T cells) than with the individual cell type (for example, CD4). This is evident from the longevity-regulating pathway that shares the *PRKACB*, *NFKB1* and *PIK3CA* genes between both CD4+ and CD8+ T cell populations (Fig. 4d).

In addition, we found that certain pathways can be enriched across specific cell types belonging to different lineages. The inflammatory mediator regulation of the 'TRP channel' pathway, for example, was overrepresented in both CD8+ T cells and CD14+ monocytes, with each

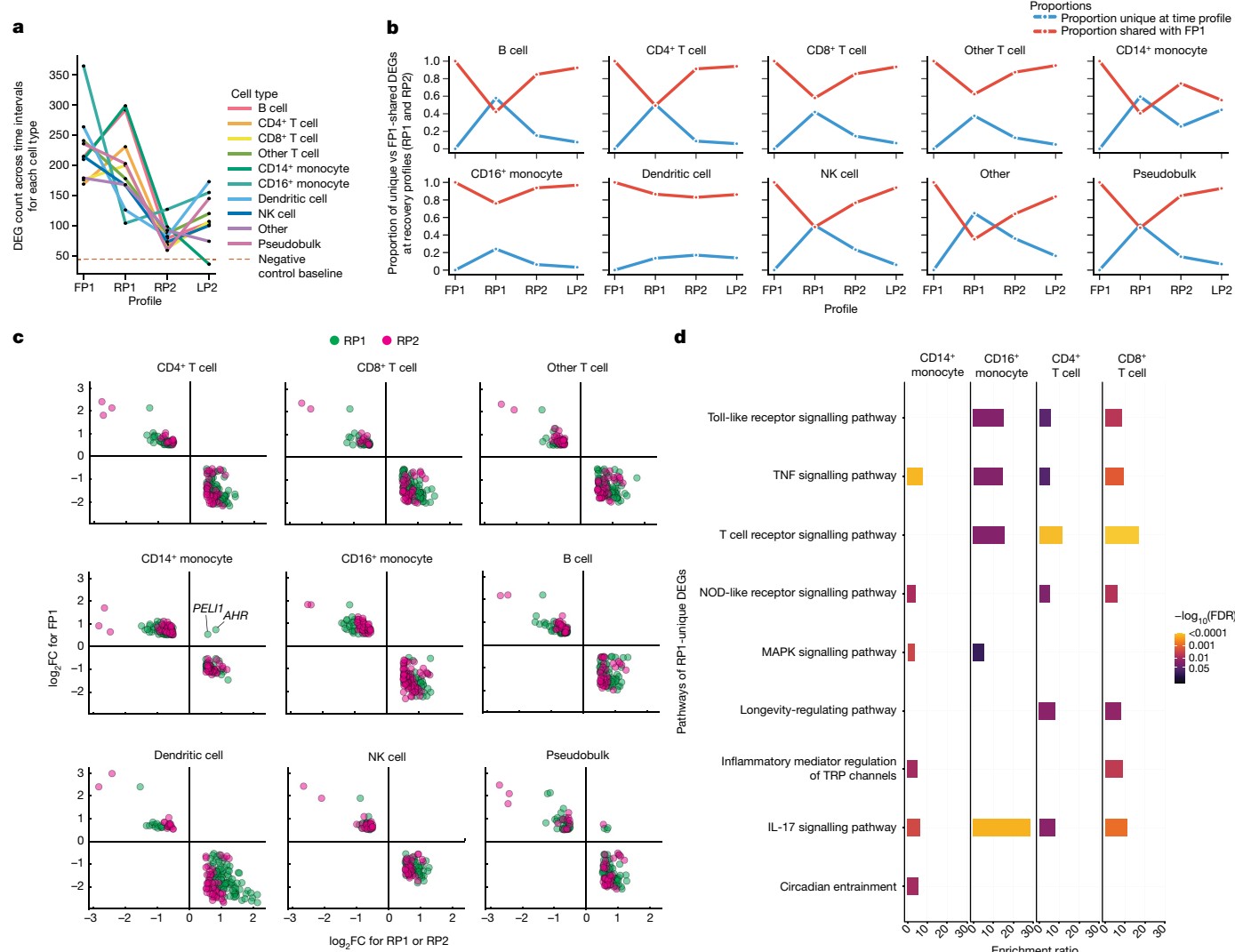

**Fig. 4 | Recovery profile dynamics in PBMCs. a**, Number of DEGs from PBMC snRNA-seq for each cell type during the flight, recovery and longitudinal profiles (adjusted two-sided $P < 0.05$, $|\log_2 FC| > 0.5$). NK, natural killer. **b**, Fraction of DEGs shared with FP1 at RP1, RP2 and LP2 for each cell type.

**c**, Directionality of $\log_2 FC$ between FP1 and RP1 and RP2 for DEGs present in both profiles. **d**, Bar chart of pathways of DEGs present in RP1 that were absent in FP1 in monocytes and T cells. Bars are shaded by the false discovery rate (FDR) value, and the enrichment ratio is on the *x* axis.

cell type contributing different gene sets associated with the pathway (Fig. 4d). Other pathways, such as the IL-17 signalling pathway, contain mixtures of shared genes and unique genes and are significantly overrepresented across all cell types (Fig. 4d). This also indicates that although cell types are distinct, they appear to have a set of core pathways in response to spaceflight, with 30–60% of overrepresented pathways shared both within and between lineages (Extended Data Fig. 6b).

To examine the chromatin accessibility and regulatory landscape in each cell population, we then analysed the transcription factor-binding site (TFBS) motif accessibility changes from snATAC-seq data to identify the top motifs in flight and recovery profiles (Extended Data Fig. 7). We first observed that increased gene expression was correlated with more accessibility at the transcription start sites, and closed chromatin was associated with downregulated genes (all $P < 0.05$, Wilcoxon rank-sum test, for all cell types; Extended Data Fig. 7a). The motif accessibility changes associated with recovery profiles recapitulated the trend in gene expression and chromatin accessibility data, which showed lower accessibility in regions of the genome with closed chromatin and higher expression in regions of more accessibility (Extended Data Fig. 7b). In addition, we showed the top five upregulated and downregulated TFBS

motifs per cell type in FP1, RP1 and RP2, which revealed both common and distinct motifs and their accessibility across cell types (Extended Data Fig. 7c–e). These data also provided further evidence of cell-type specificity in the differences of chromatin and transcription factor accessibility dynamics.

## Intra-individual spaceflight responses

To understand individual variation during spaceflight, the coefficient of variation was measured across microbial, proteomic, cytokine and gene expression normalized count data, calculated by time interval (pre-flight, in-flight, post-flight and recovery). Normalized coefficient of variation scores (see Methods) were calculated for each body area from the microbial swabs from both metagenomic and metatranscriptomic data. The oral and forearm microbial variation (Extended Data Fig. 8a,b) showed *Rothia mucilaginosa* and *Staphylococcus epidermidis* as leading variable strains, but each body site has distinct higher coefficient of variation species (other body sites are shown in Extended Data Fig. 8c–j and Supplementary Table 7). Similarly, abundance standardized coefficient of variation scores were calculated for EVP proteomic, plasma proteomic, metabolomic, RNA-seq, dRNA-seq, and

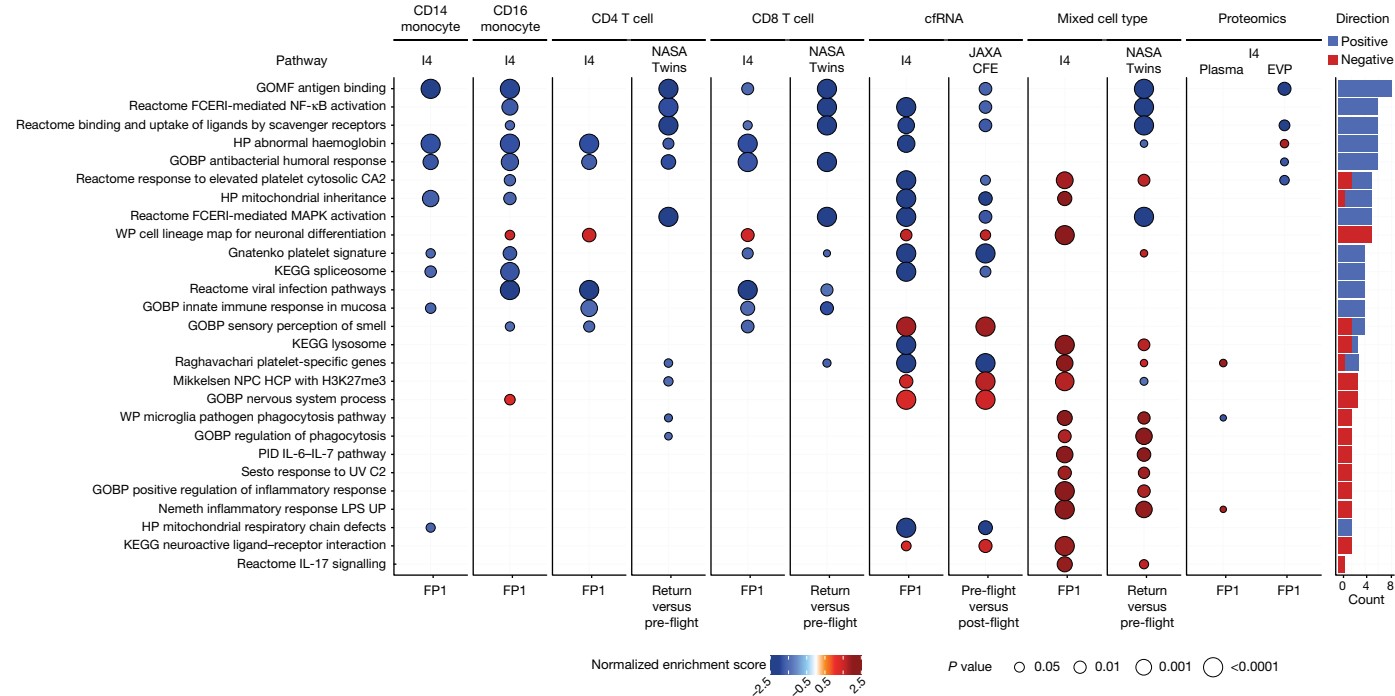

**Fig. 5 | Pathway enrichment of most variable genes.** Enriched pathways in post-flight compared with pre-flight across various assays and missions, analysed using fast gene set enrichment analysis (fGSEA). The colour represents the normalized enrichment score, whereas the dot size indicates Benjamini–Hochberg adjusted *q* values. Only the pathways with unadjusted *P* < 0.01 are shown. The barplot shows the total number of comparisons with *q* < 0.05 for every pathway, coloured by the direction of the enrichment. The column 'mixed cell type' refers to whole blood for I4 data and lymphocyte-depleted cells for the NASA Twins Study. GOBP, gene ontology and biological process; GOMF, gene ontology molecular function; HP, human phenotype; LPS, lipopolysaccharides; NPC, neural progenitor cells; PID, pathway interaction database; WP, wikipathways.

cytokine abundance normalized protein and gene counts (Extended Data Fig. 9a–f). To characterize the participant-to-participant variation and null distributions, we also calculated the differentials for FP1 along with label permutation testing on these calculations to confirm that post-flight coefficient of variation was not higher than other timepoints (see Methods; Supplementary Table 8).

We then utilized the data across missions (I4, Twins Study and JAXA), assays (scRNA-seq, cfRNA, bulk RNA-seq and proteomics) and cell types (CD4, CD8, CD14 and CD16) to find the most recurrent pathways associated with spaceflight. The most enriched pathways were antigen binding, haemoglobin, cytokine signalling and immune activation, which confirmed the DEG-related signatures and also validated the signatures from our I4 mission (Fig. 5). Moreover, the NASA Twins Study data and cfRNA profiles from the JAXA study confirmed many of the top-ranked pathways, while also showing differences between bulk blood RNA and protein markers and purified cell populations. These data indicate that leveraging both bulk and sorted cell populations can help to clarify signals coming from crew blood dynamics related to spaceflight.

## Access to datasets and crew samples

This study introduces full accessibility of astronaut data to the scientific community. Datasets are accessible through an online web portal and scientific data repositories, as well as controlled access to more sensitive data (for example, genetic sequence data) and physical specimens. The online web portal splits the data into three data browsers: the SOMA browser, the I4 single-cell browser and the microbiome browser (Extended Data Fig. 10a–c). The SOMA browser (https://soma.weill.cornell.edu/apps/SOMA_Browser/) enables visualization of gene expression (bulk RNA-seq, snRNA-seq, cfRNA-seq and spatially resolved transcriptomics), mass spectrometry (plasma

proteomics, plasma metabolomics and plasma EVP proteomics) and microbial (metagenomic and metatranscriptomic) data. Gene expression and protein abundance are visualized for each astronaut by time-point, with fold-change values, statistical significance (*q* value) and summary tables. Any selected gene is also then plotted across data for other missions, including the JAXA CFE study, the NASA Twins Study, mouse spaceflight datasets from NASA OSDR and Genelab, and control cohorts. For additional granularity, the single-cell browser (https://soma.weill.cornell.edu/apps/I4_Multiome/) provides visualizations specific to single-cell gene expression (scRNA-seq) and chromatin accessibility (scATAC-seq) data, and includes quality metrics, cell-type annotations, gene co-expression and chromatin accessibility magnitude estimates. Finally, the microbiome browser contains metagenomic and metatranscriptomic boxplots (https://soma.weill.cornell.edu/apps/I4_Microbiome/) from each timepoint of the study, spanning eight skin locations, deltoid swabs collected pre-skin biopsy, swabs of the SpaceX Dragon capsule and stool samples.

The remaining biospecimens from this study have been preserved and catalogued for continued use by the scientific community (https://cambank.weill.cornell.edu/). These samples include venous blood plasma, venous blood serum, viably frozen PBMCs, vacutainer red blood cell pellets, urine (both crude and with nucleic acid preservative), extracted saliva nucleic acids (DNA and RNA), extracted whole-blood total RNA, extracted skin swab nucleic acids (DNA and RNA) and extracted stool nucleic acids (DNA and RNA). A subset of these samples will be available for additional assays and hypothesis testing by other groups, and the remainder are allocated for long-term biobanking. These data and specimen resources for astronauts can help enable larger cohorts for increased statistical power and also for new biomedical technologies that will emerge in the future.

## Discussion

Overall, these data represent a comprehensive clinical and multi-omic resource from commercial and non-commercial astronaut cohorts, creating, to our knowledge, the first-ever aerospace medicine biobank, while providing a platform for private citizens to contribute to future astronaut biomedical studies. In addition, we have demonstrated that short-duration, high-elevation (585 km) spaceflight results in broad-ranging molecular changes, in which some of these changes mirror what has been observed during longer-duration spaceflight, including elevated cytokines, telomere elongation and gene expression changes for immune activation, DNA damage response and oxidative stress. Although more than 95% of markers return to baseline in the months after the mission had ended, some proteins, genes and cytokines appear to be activated only in the recovery period after spaceflight and persist post-flight for at least 3 months.

These results collectively indicate a dynamic recovery profile that substantially reverses the direction of differential gene expression in multiple key biological pathways from the post-flight timepoint (R+1) and afterward. This suggests that re-adaptation to Earth activates a range of restorative mechanisms that help to recover, at least in part, the physiological stress imposed by exposure to the space environment. The systematic analysis of the molecular and cellular changes observed post-flight afford us with a unique opportunity to capture naturally occurring health-restoring mechanisms, which can be used for therapeutic target discovery. Furthermore, we observed a nuanced regulatory landscape, in which enriched recovery pathways are both unique to individual cell types and span cell types in unique combinations. We have carefully indexed various profiles for flight, recovery and longitudinal analysis that are annotated in processed data files available in the NASA OSDR. The RP1 profile indicates that we need more frequent sampling in the 1–2 months directly after flight to untangle the gene and pathway responses during re-adaptation from spaceflight. This will be especially true during longer missions, in which re-adaptation will probably be more intense.

Of note, these data are, to our knowledge, the first-ever joint single-nucleus chromatin profiles (RNA and ATAC) for astronauts, and they also leverage new methods that can track gene expression and epigenetic changes within the same cells. This single-cell, dual-measurement assay provides new data on the molecular changes and regulatory response to spaceflight (for example, chromatin and TFBS accessibility), and the data revealed distinct levels of stress and adaptation by each cell type. Specifically, the T cell and monocyte cell populations (CD14 and CD16) had the largest changes in expression and response of any cell type. The differences between PBMC subpopulations also suggest that single-cell sequencing can be helpful for delineating unique cell-type responses in future studies as well. Indeed, although the immune system and haematopoietic systems both show thousands of transient changes at the gene expression level, the chromatin architecture is distinctly disrupted, in both scale and duration, in these CD14[+] and CD16[+] monocyte populations. These cell types were also found to be disrupted in the NASA Twins Study[7,23], and thus represent a key cell type to be studied for future missions.

The cfRNA, dRNA and spatial RNA profiles revealed unique profiles that differed based on recovery and longitudinal analysis, suggesting that the multi-omic footprint of spaceflight is much wider than previously observed. Although some focused studies of cfRNA have shown that it can be used to detect mitochondrial increase in blood related to spaceflight[7–9,24], to our knowledge, it had never been applied as a 'full-body molecular scan' to detect differential tissue and cell stress. Of note, the SOMA resource enables comparisons of the cfRNAs and cell-encapsulated RNAs, as well as the exosomal fractions, all within the same framework, which is essential for delineating and ranking the cells and tissues in the body that are the most disrupted by spaceflight. Similar to the utility of single-cell data[23], the granularity of seeing cell lysis from all tissues across the body in one assay makes cfRNA profiling an ideal addition to some of the proposed standard measures for spaceflight monitoring. So far, cfRNA data indicate that each part of the body may show its own transcriptional response, shedding rate or RNA excretion rates, and thus each tissue should be examined on its own and then compared with other sites and assays. Similarly, in the microbiome data, the taxonomic classifications yielded differences in variation by body site, timepoint and crew member (Extended Data Fig. 8), indicating the need for multi-site and multi-omic sampling for ideal understanding of microbiome changes associated with spaceflight.

However, this study is not without limitations. Although cell-specific comparisons could be made between the I4 mission and previous studies, the assays and collection protocols were slightly different (for example, column purification versus droplet-sorted cells), and thus comparisons will be imperfect. In addition, although the same tube types and methods were used whenever possible, cross-mission comparisons will inevitably include noise from various other types of technical variation, including batches of library preparation and extraction kits, slightly different collection intervals (L−3 versus L−10) and different sequencing or profiling technologies (for example, Illumina versus Ultima or liquid chromatography–tandem mass spectrometry versus NULISASeq). Finally, as our comparisons to other NASA and JAXA datasets span, at most, 64 astronauts, these data are not sufficient to guide medical interventions or inference of mechanisms. As such, these SOMA data and resources should be viewed as preliminary molecular maps of the response of the human body to spaceflight.

## Future directions

Nonetheless, the data from the I4 mission will be an invaluable resource for future studies. Deep analyses of secretome profiling[25], single-nucleus multiome[26], viral activation and ecological restructuring[27], skin spatial transcriptomic profiling[28], epitranscriptomic profiling[22], and genome integrity and clonal haematopoiesis[29] have already leveraged this resource. In future missions, additional biomedical profiling can help to delineate the short-term and long-term health effects of spaceflight, including changes in telomere length dynamics, DNA methylation, non-coding RNAs, as well as additional sample types, such as hair follicles, tears, sperm and other biospecimens. Indeed, the remaining samples from the I4 mission have been biobanked for just this reason and to help the scientific community tackle future objectives[12]. In addition, aliquots of DNA, RNA, protein, serum, urine and stool from the protocols performed in this paper have acquired consent for release and are available (https://cambank.weill.cornell.edu/) for request, in which researchers can then append new results to this extensible SOMA repository.

Differences between the biomedical and cellular responses of crew members may be caused by several factors, including inter-individual genetic differences (Fig. 5), the duration of the mission, the higher flight elevation (585 km), the unique environment of the SpaceX Dragon capsule, or a combination of these and other factors. To address these hypotheses, other molecular assays can be informative for future missions, including other epigenetics and chromatin conformation assays. For example, we have investigated using the method cleavage under targets and release using nuclease (Cut&Run) to profile histone modifications using 5,000 and 10,000 T cells as assay input (Supplementary Fig. 3a), down from the 10 million cells recommended in the original protocol[30]. T cells were collected from C002 during the R+194 recovery timepoint, but showed high variance at lower input. Nonetheless, the combination of current modifications profiled can be used to annotate active enhancer regions in the genome (Supplementary Fig. 3b), which is a novel profile for astronauts, and opens the door to better understanding of the gene-regulatory changes induced by spaceflight for each crew member.

Molecular changes described here can also help to guide research and countermeasure development, but are only the start of the process of mitigating risks, especially as year-long and multi-year space habitat missions will represent greater biomedical challenges. To aid in this effort, the SOMA resource will continue to expand as further samples are sequenced from the I4 mission[11,12] and samples are collected from future missions that travel farther into space, and then compared with other longitudinal, multi-omics cohorts[31]. Biospecimen samples have been collected and processed from Polaris and Axiom crews, and this Atlas also represents an open call for research participation for astronauts from any commercial or governmental programs.

Finally, although the multi-omic data and resources from the I4 crew represent the largest release of data from astronauts to date, the I4 crew is still a small cohort, and represents only the first step towards resolving the many hazards and needs for long-term missions[32], building towards enough statistical power and contextualization of normal human biological variation[33,34]. Fortunately, one of the crew members for I4 will continue to donate to the SOMA Biobank for multiple missions, including the Polaris Dawn mission, and data and samples from additional crews (for example, long-term Twins Study follow-up, Axiom Saudi, JAXA and Malta missions) are now being collected and integrated. These cross-mission datasets create a unique opportunity for long-term, in-depth analysis of the effect of spaceflight on the human body. Such data are especially important as missions travel farther away from Earth[35] and for longer periods of time, in which the data and biomedical discoveries can help to prepare commercial and state-sponsored agencies for the lunar, Mars and exploration-class missions.

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

Eliah G. Overbey[1,2,3,4✉], JangKeun Kim[1,2], Braden T. Tierney[1,2], Jiwoon Park[1,2], Nadia Houerbi[1,2], Alexander G. Lucaci[1,2], Sebastian Garcia Medina[1,2], Namita Damle[1], Deena Najjar[5], Kirill Grigorev[1,2], Evan E. Afshin[1,2], Krista A. Ryon[1], Karolina Sienkiewicz[2,6], Laura Patras[7,8], Remi Klotz[9], Veronica Ortiz[9], Matthew MacKay[6], Annalise Schweickart[2,6], Christopher R. Chin[1], Maria A. Sierra[6], Matias F. Valenzuela[10], Ezequiel Dantas[11,12], Theodore M. Nelson[13], Egle Cekanaviciute[14], Gabriel Deards[6], Jonathan Foox[1,2], S. Anand Narayanan[15], Caleb M. Schmidt[16,17,18], Michael A. Schmidt[16,17], Julian C. Schmidt[16,17], Sean Mullane[19], Seth Stravers Tigchelaar[19], Steven Levitte[19,20], Craig Westover[1], Chandrima Bhattacharya[6], Serena Lucotti[7], Jeremy Wain Hirschberg[1], Jacqueline Proszynski[1], Marissa Burke[1,21], Ashley S. Kleinman[1], Daniel J. Butler[1], Conor Loy[22], Omary Mzava[22], Joan Lenz[22], Doru Paul[23], Christopher Mozsary[1], Lauren M. Sanders[14], Lynn E. Taylor[24], Chintan O. Patel[25], Sharib A. Khan[25], Mir Suhail Mohamad[25], Syed Gufran Ahmad Byhaqui[25], Burhan Aslam[25], Aaron S. Gajadhar[26], Lucy Williamson[26], Purvi Tandel[26], Qiu Yang[26], Jessica Chu[26], Ryan W. Benz[26], Asim Siddiqui[26], Daniel Hornburg[26], Kelly Blease[26], Juan Moreno[27], Andrew Boddicker[27], Junhua Zhao[27], Bryan Lajoie[27], Ryan T. Scott[28], Rachel R. Gilbert[28], San-huei Lai Polo[28], Andrew Altomare[27], Semyon Kruglyak[27], Shawn Levy[27], Ishara Ariyapala[29], Joanne Beer[29], Bingqing Zhang[29], Briana M. Hudson[30], Aric Rininger[30], Sarah E. Church[30], Afshin Beheshti[31,32], George M. Church[33], Scott M. Smith[34], Brian E. Crucian[34], Sara R. Zwart[35], Irina Matei[7,12], David C. Lyden[7,12], Francine Garrett-Bakelman[36,37], Jan Krumsiek[1,2,6], Qiuying Chen[38], Dawson Miller[38], Joe Shuga[39], Stephen Williams[39], Corey Nemec[39], Guy Trudel[40,41,42], Martin Pelchat[43],

Odette Laneuville[44], Iwijn De Vlaminck[22], Steven Gross[38], Kelly L. Bolton[45], Susan M. Bailey[24,46], Richard Granstein[47], David Furman[10,48,49,50], Ari M. Melnick[12,23], Sylvain V. Costes[14], Bader Shirah[51], Min Yu[9], Anil S. Menon[35], Jaime Mateus[19], Cem Meydan[1,2,23 ✉] & Christopher E. Mason[1,2,3,52,53 ✉]

[1]Department of Physiology and Biophysics, Weill Cornell Medicine, New York, NY, USA. [2]The HRH Prince Alwaleed Bin Talal Bin Abdulaziz Alsaud Institute for Computational Biomedicine, Weill Cornell Medicine, New York, NY, USA. [3]BioAstra Inc., New York, NY, USA. [4]Center for STEM, University of Austin, Austin, TX, USA. [5]Albert Einstein College of Medicine, Bronx, NY, USA. [6]Tri-Institutional Biology and Medicine Program, Weill Cornell Medicine, New York, NY, USA. [7]Children's Cancer and Blood Foundation Laboratories, Departments of Pediatrics and Cell and Developmental Biology, Drukier Institute for Children's Health, Weill Cornell Medicine, New York, NY, USA. [8]Department of Molecular Biology and Biotechnology, Center of Systems Biology, Biodiversity and Bioresources, Faculty of Biology and Geology, Babes-Bolyai University, Cluj-Napoca, Romania. [9]Department of Stem Cell Biology and Regenerative Medicine, Keck School of Medicine, University of Southern California, Los Angeles, CA, USA. [10]Buck Institute for Research on Aging, Novato, CA, USA. [11]Department of Medicine, Division of Endocrinology, Weill Cornell Medicine, New York, NY, USA. [12]Meyer Cancer Center, Weill Cornell Medicine, New York, NY, USA. [13]Department of Microbiology & Immunology, Vagelos College of Physicians & Surgeons, Columbia University Irving Medical Center, New York, NY, USA. [14]Space Biosciences Division, NASA Ames Research Center, Moffett Field, CA, USA. [15]Department of Health, Nutrition, and Food Sciences, Florida State University, Tallahassee, FL, USA. [16]Sovaris Aerospace, Boulder, CO, USA. [17]Advanced Pattern Analysis and Human Performance Group, Boulder, CO, USA. [18]Department of Systems Engineering, Colorado State University, Fort Collins, CO, USA. [19]Space Exploration Technologies Corporation (SpaceX), Hawthorne, CA, USA. [20]Division of Pediatric Gastroenterology, Stanford University, Palo Alto, CA, USA. [21]Department of Neurosurgery, Houston Methodist Research Center, Houston, United States. [22]Nancy E. and Peter C. Meinig School of Biomedical Engineering, Cornell University, Ithaca, NY, USA. [23]Department of Medicine, Division of Hematology/Oncology, Weill Cornell Medicine, New York, NY, USA. [24]Department of Environmental and Radiological Health Sciences, Colorado State University, Fort Collins, CO, USA. [25]TrialX Inc., New York, NY, USA. [26]Seer Inc., Redwood City, CA, USA. [27]Element Biosciences, San Diego, CA, USA. [28]KBR, Space Biosciences Division, NASA Ames Research Center, Moffett Field, CA, USA. [29]Alamar Biosciences Inc., Fremont, CA, USA. [30]NanoString Technologies, Seattle, WA, USA. [31]Blue Marble Space Institute of Science, Space Biosciences Division, NASA Ames Research Center, Moffett Field, CA, USA. [32]Stanley Center for Psychiatric Research, Broad Institute of MIT and Harvard, Cambridge, MA, USA. [33]Harvard Medical School and the Wyss Institute, Boston, MA, USA. [34]National Aeronautics and Space Administration, Johnson Space Center, Human Health and Performance Directorate, Biomedical Research and Environmental Sciences Division, Houston, TX, USA. [35]University of Texas Medical Branch, Galveston, TX, USA. [36]Department of Biochemistry and Molecular Genetics, University of Virginia, Charlottesville, VA, USA. [37]Department of Medicine, Division of Hematology and Oncology, University of Virginia, Charlottesville, VA, USA. [38]Department of Pharmacology, Weill Cornell Medicine, New York, NY, USA. [39]10x Genomics, Pleasanton, CA, USA. [40]Bone and Joint Research Laboratory, Ottawa Hospital Research Institute, Ottawa, Ontario, Canada. [41]Department of Medicine, Division of Physiatry, The Ottawa Hospital, Ottawa, Ontario, Canada. [42]Department of Cellular and Molecular Medicine, Faculty of Medicine, University of Ottawa, Ottawa, Ontario, Canada. [43]Department of Biochemistry, Microbiology, and Immunology, University of Ottawa, Ottawa, Ontario, Canada. [44]Department of Biology, University of Ottawa, Ottawa, Ontario, Canada. [45]Department of Medicine, Division of Oncology, Washington University School of Medicine, St Louis, MO, USA. [46]Cell and Molecular Biology Program, Colorado State University, Fort Collins, CO, USA. [47]Department of Dermatology, Weill Cornell Medicine, New York, NY, USA. [48]Cosmica Biosciences Inc., San Francisco, CA, USA. [49]Stanford 1000 Immunomes Project, Stanford School of Medicine, Stanford University, Stanford, CA, USA. [50]Institute for Research in Translational Medicine, Universidad Austral and CONICET, Buenos Aires, Argentina. [51]Department of Neuroscience, King Faisal Specialist Hospital and Research Centre, Jeddah, Saudi Arabia. [52]The Feil Family Brain and Mind Research Institute, Weill Cornell Medicine, NY, USA. [53]WorldQuant Initiative for Quantitative Prediction, Weill Cornell Medicine, New York, NY, USA. ✉e-mail: elo4004@med.cornell.edu; cem2009@med.cornell.edu; chm2042@med.cornell.edu

## Methods

### Institutional Review Board statement

All participants consented at an informed consent briefing at SpaceX, and samples were collected and processed under the approval of the Institutional Review Board at Weill Cornell Medicine, under protocol 21-05023569. All crew members consented to data and sample sharing.

### I4 data compendium data generation

Full methodology for sample preparation, nucleic acid/protein extraction, sequencing, mass spectrometry and analysis are reported in Supplementary Note 1. Sample collection has been previously reported[12].

### I4 versus NASA Twins Study cytokine analysis

Differential abundance analyses of cytokines and other analytes were conducted using a mixed-effects model. This model included fixed effects for time, treated as a categorical variable with levels corresponding to preflight (L−92, L−44 and L−3), immediate return (R+1) and recovery periods (R+45, R+82 and R+194). In addition, participant-specific effects were incorporated as random effects in the model. The $P$ values for the coefficients obtained from this model were adjusted for multiple comparisons using the Benjamini–Hochberg procedure.

### 10x Genomics snRNA-seq negative control data

We acquired cryovials of PBMCs from a healthy man 22 years of age (AllCells) and stored them in vapour-phase liquid nitrogen cryotanks. Two cryovials from this donor were thawed and processed on two different days in the same week, at the same laboratory, using the 10x Genomics demonstrated protocol called Nuclei Isolation for Single Cell Multiome ATAC + Gene Expression Sequencing (CG000365). From each day's nuclei suspension, ATAC and gene expression libraries were generated (in technical triplicates on the same chip) according to the Chromium Next GEM Single Cell Multiome ATAC + Gene Expression User Guide (CG000338) and sequenced on a NovaSeq 6000 Sequencer. The resulting single-nucleus GEX and single-nucleus ATAC files were aligned using the cellranger-arc pipeline (v2.0.2) from 10x Genomics against the human reference genome hg38. Quality control and cell annotation were performed on the snGEX gene-cell matrices using the R Seurat package (v4.2.0)[36]. Subpopulations were clustered and labelled using a publicly available Azimuth human PBMC reference[37] in conjunction with Seurat's supervised clustering functionality. DEG analysis for labelled subpopulations was performed using the FindMarkers functionality of Seurat, with a log fold change (logFC) cut-off point of 0.5. $P$ values of resultant genes were measured using the Wilcoxon rank-sum test and deemed significant for $P < 0.05$. Correlation of chromatin accessibility and gene expression changes in FP1 was characterized by summing up the accessibility in a promoter window consisting of the transcription start site ± 500 bp for every cell in a given cell type across every gene, normalizing it to counts per million, taking the $\log_2$ difference between the different timepoints and matching it to the $\log_2$FC from the differential RNA analysis.

### Recovery profile analysis

DEGs from each cell type from the PBMC 10x Genomics single-cell data were filtered for |logFC| > 0.5 and adjusted $P < 0.05$ for the FP1, RP1, RP2 and LP2 profiles. An UpSet plot of shared genes was generated using Intervene[38]. Overrepresented pathways were calculated using WebGestalt[39] using the default parameters and the KEGG database. Pathways included in the analysis were within the top 40 lowest false discovery rate values or all pathways with a false discovery rate under 0.05.

### Transcription factor motif accessibility analysis

chromVAR[40] was used for the analysis of sparse chromatin accessibility from 10x Genomics single-cell data. The FindMarkers function of the Seurat package was used for differential analysis with setting mean. fxn as rowMeans and fc.name = "avg_diff". The top five differentially accessible transcription factor motifs from each cell type from the PBMC 10x Genomics single-cell data for the FP1, RP1 and RP2 were selected for visualization by heatmaps. Heatmaps were generated by the ComplexHeatmap R package.

### Calculation of individual variation

To distinguish the analytes that varied with the greatest magnitude between individual crew members (C001, C002, C003 and C004), a coefficient of variation (CV) calculation was applied across measurements collected for each: metagenomic species, metatranscriptomic species, plasma protein abundance, EVP protein abundance, metabolite abundance, cytokine abundances (Alamar Bio) and gene expression values (Oxford Nanopore dRNA-seq and Ultima RNA-seq). CV calculations were performed using the formula 'np.std($x$, ddof = 0)/np.mean($x$)', where '$x$' is the array of normalized analyte values and 'np' refers to the numpy scientific computing package ($v = 1.26$). CV calculations were split into pre-flight (L−92, L−44 and L−3), post-flight (R+1) and recovery (R+45 and R+82) intervals. The metagenomic and metatranscriptomic data also include the in-flight (FD2 and FD3) interval.

Before performing the CV calculation, data were first normalized. The 'limma' R package was applied to the plasma proteomic, EVP proteomic and metabolite data (v3.52). DESeq2 (v1.36.0) was applied to the Oxford Nanopore dRNA-seq and Ultima RNA-seq data. For microbiome data, to measure individual contributions per astronaut and for each tissue type ('armpit', 'forearm', 'nasal', 'oral', 'post-auricular', 'T-zone', among others), we averaged MetaPhlAn4, species-level, relative abundance values across our collected sample points. CVs were calculated for individual genes, proteins and microbial organisms.

An abundance standardized CV calculation was also performed to correct for the mean/CV relationship in the assays. First, the normalized abundance values across all the analytes of a given dataset were split into approximately 100 mean-abundance quantiles ($M\_q$ where $q$ is 1–100 or fewer, in case it resulted in less than 10 points per quantile group). For each quantile group, the mean and the standard deviation of CV values for the analytes belonging to $M\_q$ were calculated (CVmean_$q$ and CVs.d._$q$, respectively) to prepare the mean/CV distribution reference set across all timepoints and samples. Then, for a given timepoint, the CV calculated for each analyte was matched to the corresponding CVmean_$q$ and CVs.d._$q$ based on cross-referencing to the $M\_q$ interval overlap. Finally, the abundance standardized CV was calculated as the $z$ score of the CV as (CV − CVmean_$q$)/(CVs.d._$q$). Permutation testing of the post-flight versus pre-flight difference was performed by repeating this procedure 10,000 times on shuffled labels for samples, taking the difference. Rank and $z$ score of the observed value were calculated against the random permutations to order the genes and used as input to fGSEA for pathway enrichment analysis. Gene Ontology analysis on abundance standardized CVs was performed with Enrichr[41,42] using the default settings.

### Additional methods and details

In-depth methods and protocol information for all assays in Fig. 1a is located in Supplementary Note 1.

### Reporting summary

Further information on research design is available in the Nature Portfolio Reporting Summary linked to this article.

## Data availability

Datasets have been uploaded to two data repositories: the NASA OSDR (https://osdr.nasa.gov; comprising NASA GeneLab[15] and the NASA Ames Life Sciences Data Archive[16,32]). Identifiers for publicly downloadable datasets in the OSDR are documented in Supplementary Table 9. Data

can be visualized online through the SOMA Data Browser (https://soma.weill.cornell.edu/apps/SOMA_Browser), the single-cell expression and chromatin browser (https://soma.weill.cornell.edu/apps/I4_Multiome/) and the microbiome browser (https://soma.weill.cornell.edu/apps/I4_Microbiome/).

## Code availability

Code for data processing is available at https://github.com/eliah-o/inspiration4-omics.

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

**Acknowledgements** C.E.M. thanks the WorldQuant Foundation, NASA (NNX14AH50G, NNX17AB26G, 80NSSC22K0254, 80NSSC23K0832 and NNH18ZTT001N-FG2), the US National Institutes of Health (R01MH117406, P01CA214274, R01AI151059, and R01CA249054), the LLS (MCL7001-18 and 9238-16, 7029-23) and the GI Research Foundation (GIRF). E.G.O. thanks Quest Diagnostics for logistical support across multiple biospecimen collection locations and NASA (80NSSC21K0316). The NASA OSDR thanks the NASA Space Biology Program and the NASA Human Research Program. S.A.N. thanks NASA (80NSSC19K0426 and 80NSSC19K1322). We thank the Scientific Computing Unit at Weill Cornell Medicine, including R. Ahmed, A. Mahmoud and J. Hargitai; J. J. Hastings for collections; I4 logistics and K. J. Venkateswaran for microbiome guidance; and the NASA GeneLab and NASA ALSDA teams for customizing solutions for data accessibility in the OSDR. J. Kim thanks the MOGAM Science Foundation and was supported by the Basic Science Research Program through the National Research Foundation of Korea (NRF) funded by the Ministry of Education (RS-2023-00241586). The astronaut cytokine data were provided from the Nutritional Status Assessment project and the Biochemical Profile Projects, which were funded by the Human Health Countermeasures Element of the NASA Human Research Program. N.H. thanks Z. Wan and J. Gu for providing the lists of tissue-specific and tissue-enriched proteins compiled from the Human Protein Atlas and help with the GSEA analysis. R.G. receives funding from Pfizer, Biohaven Pharmaceuticals, Leo Pharma A/S and the Howard and Abby Milstein Foundation. L.P. thanks the Association of Transdisciplinary Society of Personalized Oncology for Combating Cancer for financial support through the postdoctoral research fellowship STOP Cancer. D.C.L. acknowledges support from the Children's Cancer and Blood Foundation, the Malcolm Hewitt Weiner Foundation, the Manning Foundation, the Sohn Foundation, the Theodore A. Rapp Foundation and the AHEPA Vth District Cancer Research Foundation. S.M.B. thanks NASA (80NSSC19K0434). A.M.M. is funded by NCI R35-CA220499. We thank F. Wu and D. Winer for data analysis and interpretation of data.

**Author contributions** C.E.M., M.M. and J.M. conceptualized the study. C.E.M., E.G.O., C. Meydan, S.M. and J.M. conducted the methodology. E.G.O., C.E.M., B.T.T., J. Park, N.H., J. Kim, I.M. and K.G. wrote the original draft of the manuscript. E.G.O., B.T.T., C. Meydan, M.A. Schmidt, M.B., J. Kim, M.A. Sierra, C.M.S., J.C.S., S.M.B., K.A.R. and B.S. reviewed and edited the manuscript. E.G.O., A.G.L., C. Meydan, J. Kim, N.H., J. Park, B.T.T., S.G.M., K.S. and C.W. performed the visualization. C.E.M., J.M., A.S.M., E.G.O., E.E.A., C. Mozsary and D.J.B. provided project administration. C.E.M., A.M.M., S.M.B., A.S.G., L.W., P.T., Q.Y., J.C., R.W.B., A. Siddiqui, D.H., K.B., J.M., A. Boddicker, J.Z., B.L., A.A., S.K., S.L., I.A., J.B. and B.Z. provided resources. C. Meydan, A.G.L., B.T.T., J. Kim, C.R.C., K.G., E.G.O., J. Park, N.H., S.G.M., K.S., A. Schweickart, T.M.N., E.C., J.F., S.A.N., S.S.T., C.B., S. Levitte, C.L., D.P. and J. Kim performed formal analysis. E.G.O., B.M.H., A.R., S.E.C., S.G., B.T.T., J. Kim, N.D., D.N., K.A.R., J.W.H., L.P., R.K., V.O., E.D., S. Lucotti, O.M., J.L., C. Meydan, L.E.T., J. Proszynski, F.G.-B., A.S.K., S.R.Z., B.E.C., S.M.S., A. Beheshti, Q.C., D.M., K.L.B., S.M.B., R.G., I.M., J. Krumsiek, M.F.V., J.S., S.W., C.N., G.T., M.P., O.L. and D.F. conducted the investigation. E.G.O., J. Kim, L.M.S., R.T.S., S.V.C., R.R.G., S.-h.L.P., C.O.P., G.D., S.A.K., M.S., S.G.A.B. and B.A. curated the data. C.E.M., J.M., A.M.M., D.C.L., M.Y., S.M.B., I.D.V., K.L.B. and G.M.C. acquired funding. C.E.M. supervised the study. All authors read and approved the final manuscript.

**Competing interests** B.T.T. is compensated for consulting with Seed Health and Enzymetrics Biosciences on microbiome study design and holds an ownership stake in the former. K.L.B. receives research funding from Servier and Bristol Myers Squibb, serves on the medical advisory board of GoodCell. SC Employee and is a shareholder at NanoString Technologies. K.B., J.M., A. Boddicker, J.Z., B.L., A.A., S.K. and S.L. are employees of and have a financial interest in Element Biosciences. E.E.A. is a consultant for Thorne HealthTech. A.S.G., L.W., P.T., Q.Y., J.C., R.B., A. Siddiqui and D.H. are employees of and have a financial interest in Seer Inc. and Prognomiq Inc. C. Meydan is compensated by Thorne HealthTech. C.E.M. is co-founder of Cosmica Biosciences. D.C.L. and I.M. receive research grant support/funding from Atossa Inc. R.G. is on the scientific advisory board of Elysium Health, is an advisor to Gore Range Capital and is also an informal advisor to BelleTorus Corporation, but has no financial ties to BelleTorus at this time. C.M.S., J.C.S. and M.A. Schmidt hold shares in Sovaris Holdings LLC. J. Krumsiek holds equity in Chymia LLC, intellectual property in PsyProtix and is co-founder of iollo. M.Y. is the founder and president of CanTraCer Biosciences Inc. The GC COI list is available at arep.med.harvard.edu/gmc/tech.html. A.M.M. has research funding from Jannsen, Epizyme and Daiichi Sankyo and has consulted for Treeline, AstraZeneca and Epizyme. All other authors declare no competing interests.

**Additional information**
**Correspondence and requests for materials** should be addressed to Eliah G. Overbey, Cem Meydan or Christopher E. Mason.

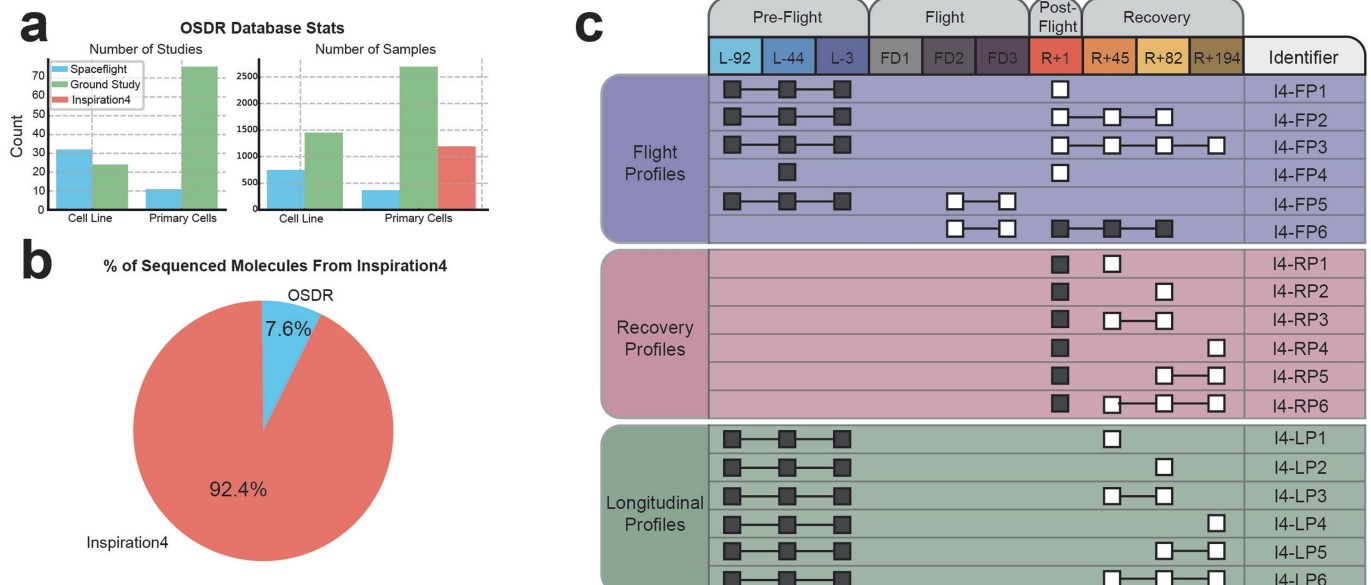

**Extended Data Fig. 1 | Prior Work and Comparative Profiles. (a)** Prior human spaceflight omics study and sample counts with publicly available data, which is housed in OSDR. **(b)** Total number of sequence nucleic acid molecules in all prior studies (OSDR, blue) compared to this study (Inspiration4, red).

**(c)** Visualization of the different analysis paradigms used when analyzing the time-series spaceflight data. The database identifier is provided as a shorthand to reference each comparison.

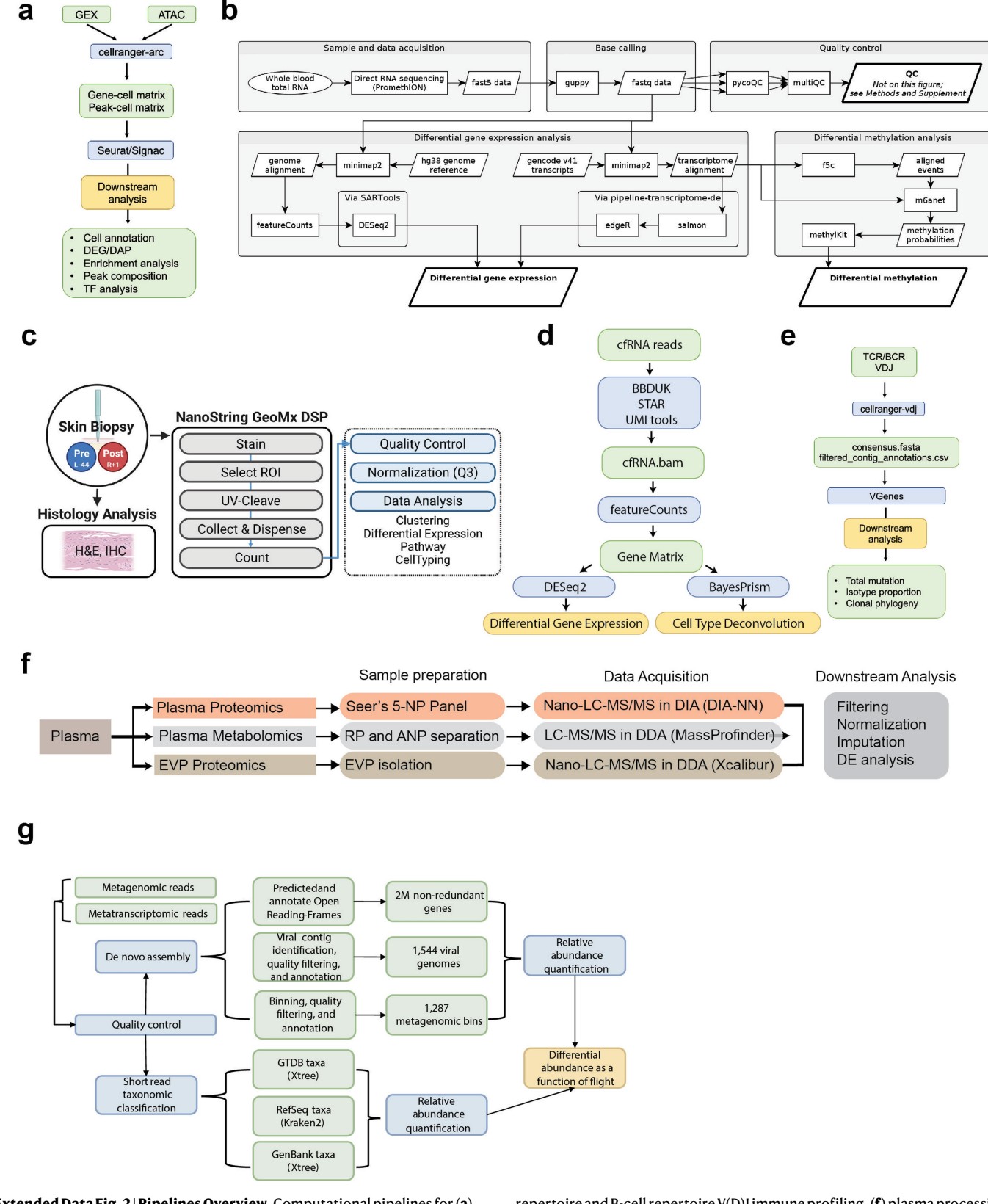

**Extended Data Fig. 2 | Pipelines Overview.** Computational pipelines for (**a**) 10x Genomics Multiome sequencing (snRNA and snATAC), (**b**) ONT direct RNA-sequencing gene expression and m6A detection, (**c**) Nanostring GeoMx whole transcriptome atlas profiling, (**d**) cfRNA gene abundances, (**e**) T-cell repertoire and B-cell repertoire V(D)J immune profiling, (**f**) plasma processing for proteomic, metabolomic, and EVP proteomic profiling, and (**g**) microbial profiling.

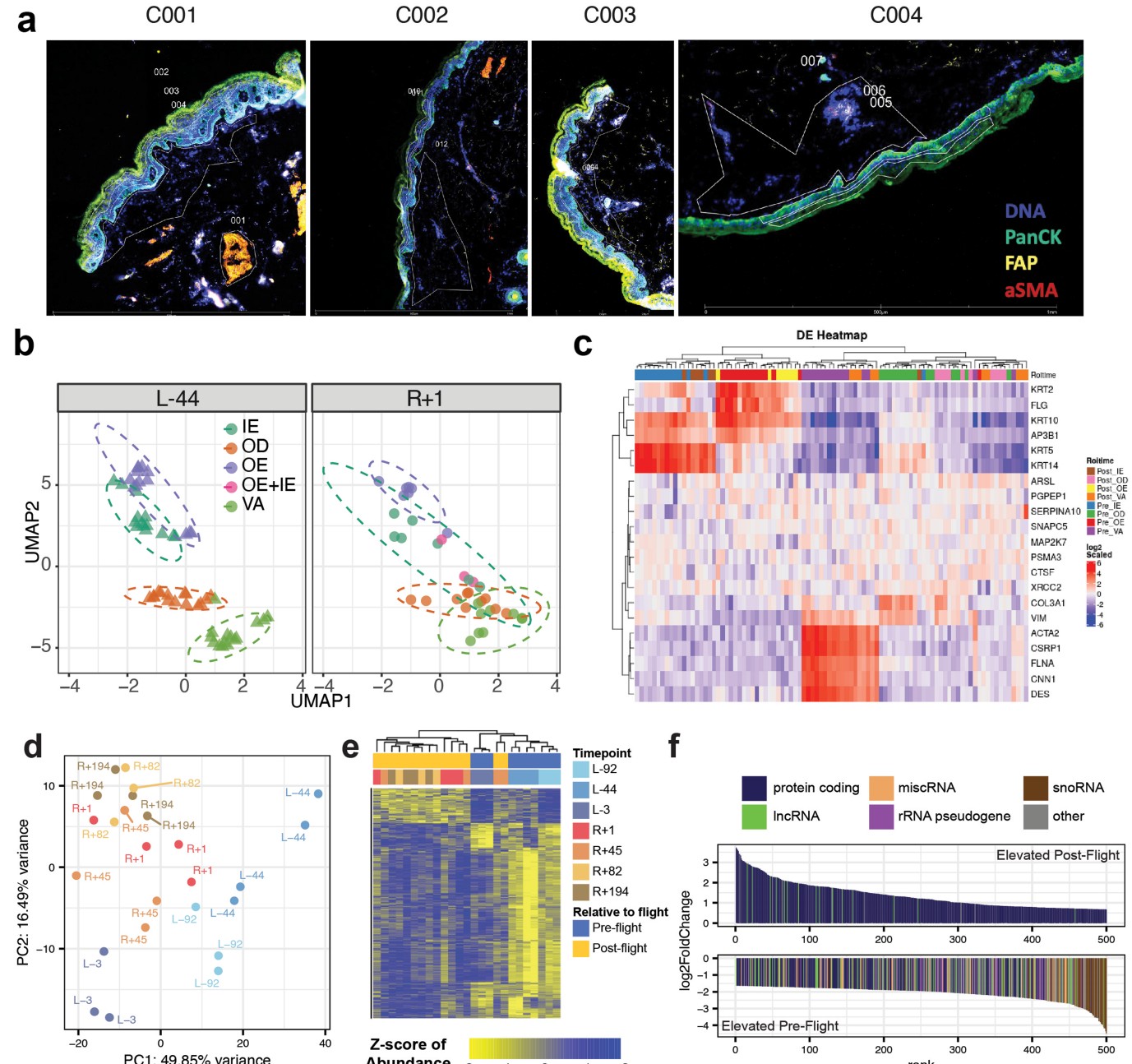

**Extended Data Fig. 3 | Transcriptomic Fingerprint of Short Duration Spaceflight.** (**a**) Representative spatial imaging of skin biopsy tissue for each crew member (C001, C002, C003, and C004). Processing was done in two batches and across four ROI types. (**b**) UMAP projection of the ROIs. The colors represent ROI types and shapes represent time points. Most of the ROIs showed good clustering around ROI types in both time points. (**c**) Heatmap visualization of top variable genes across ROIs and time points in skin biopsies. (**d**) PCA on scaled vst normalized counts of top 500 variable genes in cfRNA data. (**e**) Z-score of vst normalized cfRNA abundances from DESeq2 (BH adjusted two-sided p-value < 0.01, |log2FC | >1). Total of 927 genes. (**f**) cfDNA RNA species elevated pre-flight vs post-flight. Top 500 displayed for each group ranked by $\log_2$ fold-change.

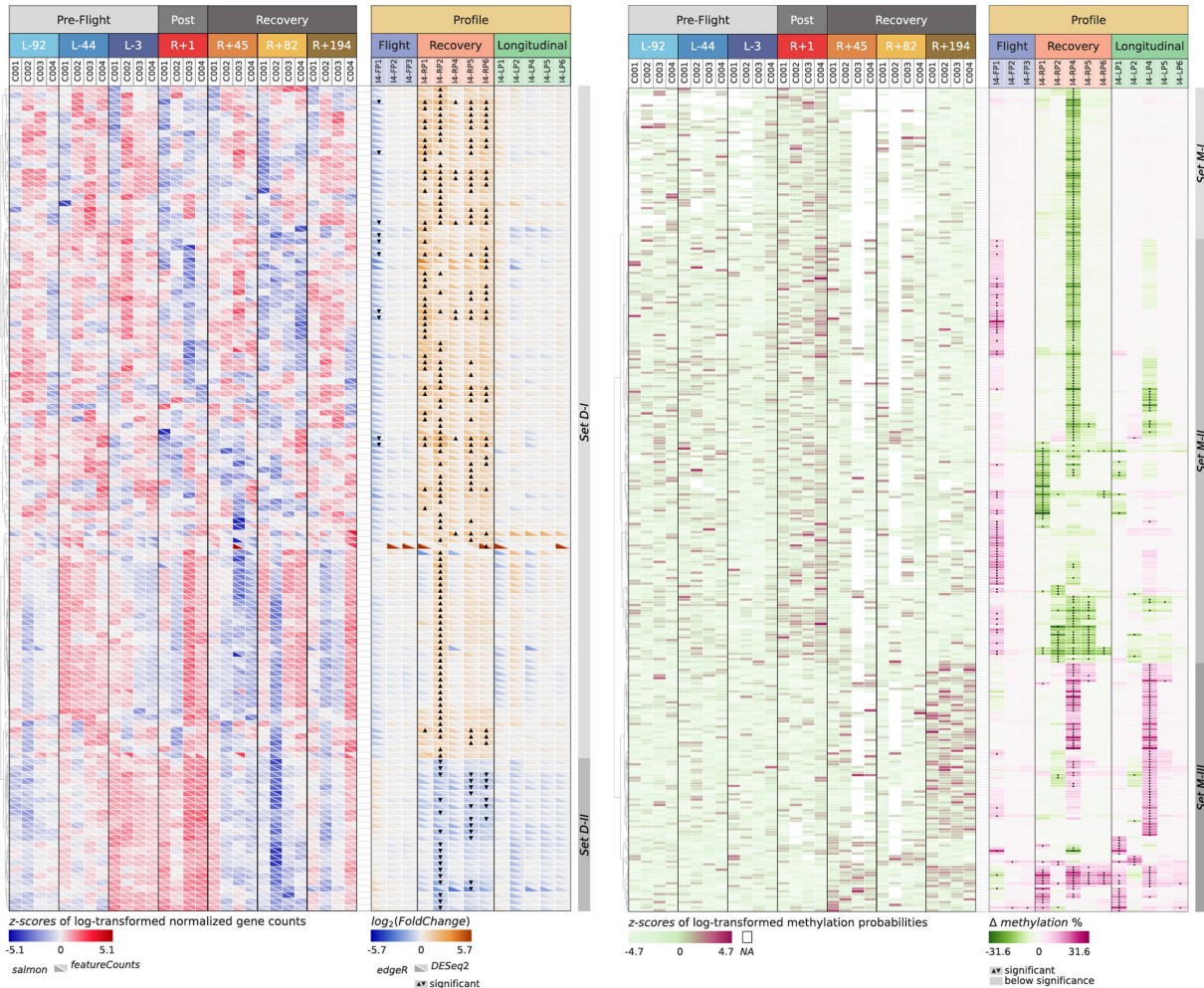

**a** Differential Gene Expression

**b** Differential m6A Abundance

*z-scores* of log-transformed normalized gene counts

-5.1   0   5.1

*salmon*   *featureCounts*

*log₂(FoldChange)*

-5.7   0   5.7

edgeR   DESeq2

significant

below significance

*z-scores* of log-transformed methylation probabilities

-4.7   0   4.7   NA

Δ *methylation %*

-31.6   0   31.6

significant

below significance

**Extended Data Fig. 4 | Direct RNA-seq Gene Expression and RNA m⁶A Modifications Across 13 Comparative Profiles. (a)** Patterns of gene expression across seven time points and 13 comparisons. *Left:* z-scored log-transformed normalized gene counts obtained from salmon (bottom left of each cell) and featureCounts (top right of each cell). *Right:* log₂(fold-change) values obtained from edgeR (bottom left of each cell) and from DESeq2 (top right of each cell). The genes are clustered by z-scored log-transformed normalized counts using the correlation distance metric. **(b)** Patterns of base-level m⁶A modifications across seven time points and 13 comparisons.

*Left:* z-scored log-transformed positional methylation probabilities obtained from m6anet. *Right:* percentage of change in methylation between the conditions in each comparison, obtained from methylKit. The sites are clustered by the pattern of differential methylation across all comparisons using the correlation distance metric. On both panes **(a)** and **(b)**, only the genes and sites with significant differences in expression and/or methylation in at least one comparison are plotted; the significance of individual comparisons is annotated with up and down arrows.

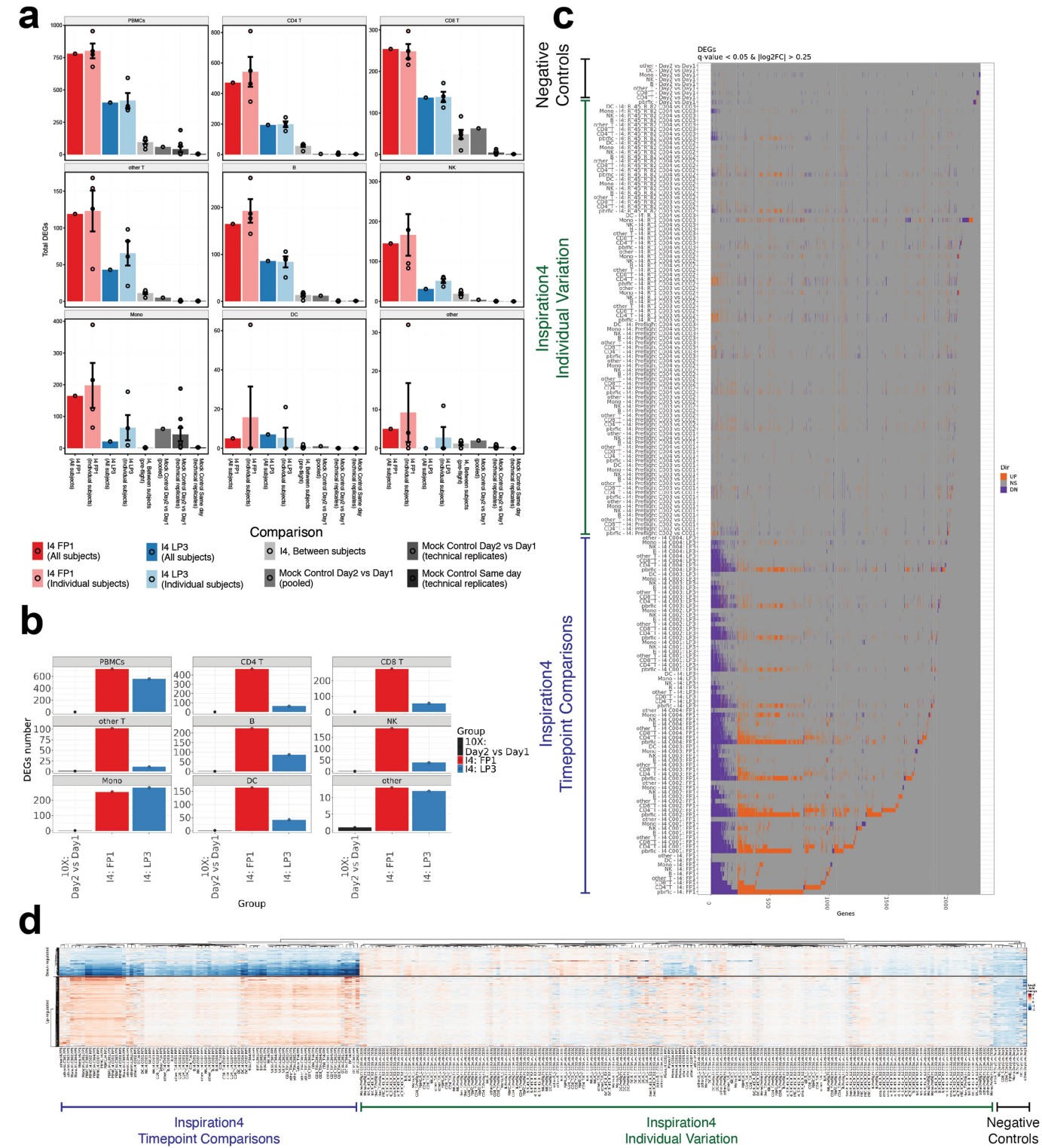

**Extended Data Fig. 5 | Single-Nuclei RNA-Sequencing Controls. (a)** The number of DEGs in I4 flight profiles (FP1: grouped crew or individual crew members), I4 longitudinal profiles (LP3: grouped crew or individual crew members), and negative control groups (mock control day 2 vs day 1, mock control same day technical replicate differences, and I4 inter-subject comparisons in preflight). All comparisons were done with downsampling to the same number of cells. Intra-timepoint subject comparisons are crew-to-crew comparison in the pre-flight (L-92, L-44, L-3), immediately post-flight (R + 1), and recovery (R + 45, R + 82) time intervals. The bars show the mean of the total number of DEGs, and error bars show the standard error for groups that have 3 or more comparisons summarized. (n = 4 independent subjects with 6 timepoints, and n = 1 control subject with 2 timepoints and 3 technical

scRNAseq replicates for each timepoint.) **(b)** The number of total DEGs identified by DESeq2 and pseudobulk counts in I4 flight profiles (FP1), I4 longitudinal profiles (LP3), and negative control groups (mock control day 2 vs day 1). All comparisons were done with aggregation into a single sample for each crew and each cell type. The bars show the mean of the total number of DEGs. **(c)** DEG directionality heatmap represents the overlap of up-regulated (orange) and down-regulated (purple) DEGs across comparison groups (I4 timepoint comparisons, I4 individual variation, 10x negative controls). **(d)** Heatmap representing the log$_2$ fold-change of I4-FP1 up-regulated and down-regulated PBMC DEGs in each comparison group (I4 timepoint comparisons, I4 individual variation, 10x Genomics negative controls).

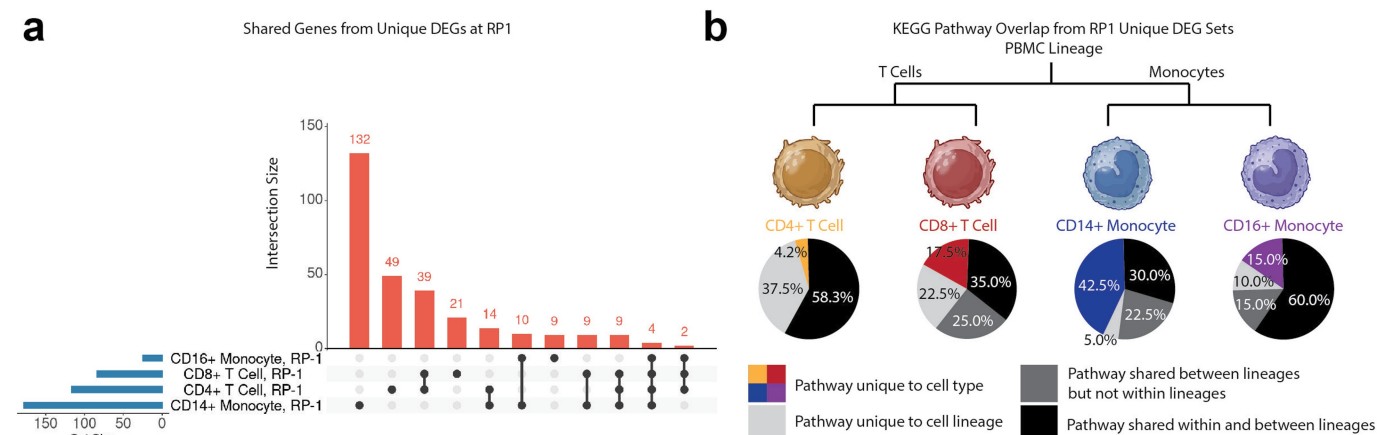

**a** Shared Genes from Unique DEGs at RP1

**b** KEGG Pathway Overlap from RP1 Unique DEG Sets

**Extended Data Fig. 6 | Recovery Profile Analysis. (a)** DEGs shared between the T cell and monocyte lineages for DEGs present in RP1, but not present in the FP1 profile. **(b)** Overrepresented KEGG pathways from DEG sets unique to the RP1 profile in the t-cell and monocyte lineages. The percent of pathways unique to each cell type are quantified along with the various configurations in which the pathway is shared between cell types and lineages.

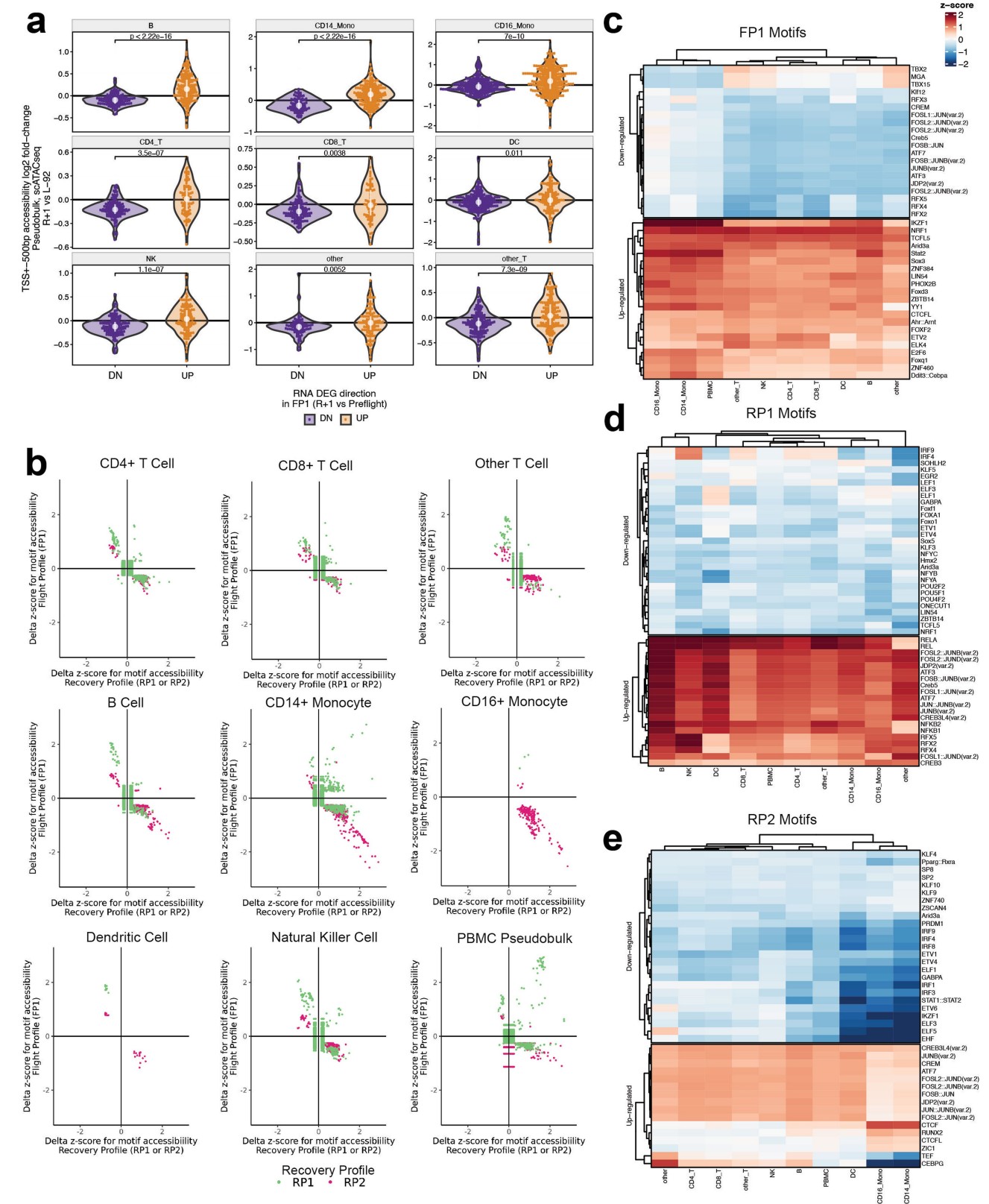

**Extended Data Fig. 7 | Single-Nuclei ATAC-seq and TFBSs.** (**a**) log2 fold-change of chromatin accessibility at promoters of genes that are differentially expressed in FP across different cell types. Promoters are defined as the transcription start site ± 500 bp of a given gene. Two-sided p-values were calculated by Wilcoxon rank-sum test. Violin plots show the density of the points, and the center white dot represents the median, and the white line shows the range of the first and third quartiles. (**b**) Directionality of delta z-score for motif accessibility between flight profile FP1 and recovery profiles RP1 and RP2 for significant TF motifs of FP1 present in either RP1 or RP2. (**c**) Heatmap of accessibility z-score of top5 significantly increased and decreased TFs in each cell type in FP1. (**d**) Heatmap of accessibility z-score of top5 significantly increased and decreased TFs in each cell type in RP1. (**e**) Heatmap of accessibility z-score of top5 significantly increased and decreased TFs in each cell type in RP2.

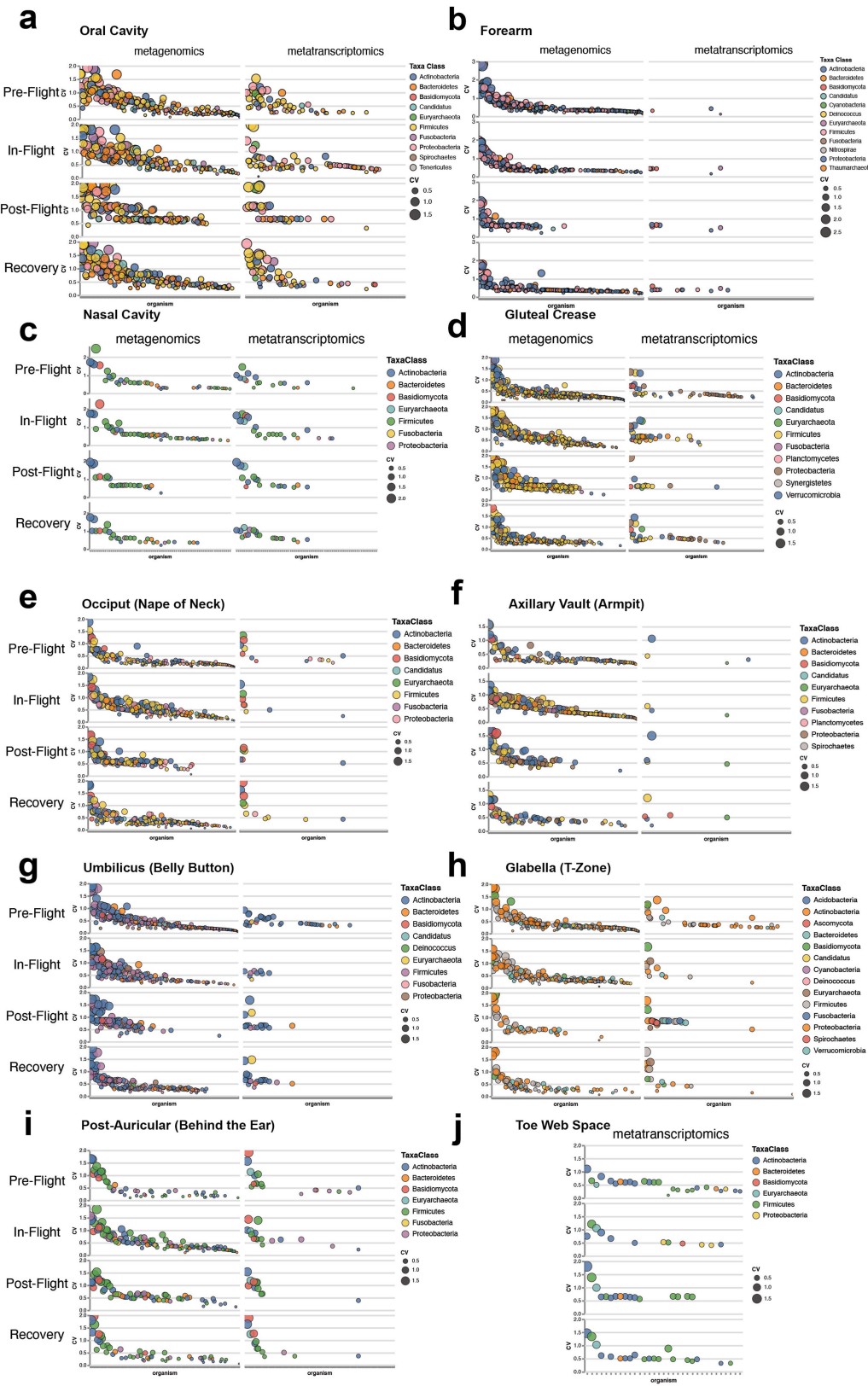

**Extended Data Fig. 8 | Coefficients of Variation (CVs) for Microbial Taxa Across Body Sites. (a-j)** CVs for different microbial taxa from oral, forearm, nasal, gluteal crease, occiput, axillary vault, umbilicus, glabella, post-auricular, and toe web space regions generated from skin swabs. CVs are calculated from both metagenomic and metatranscriptomic sequencing data.

**Extended Data Fig. 9 | CV Analysis of Datasets by Time Interval.**
(**a-f**) Abundance standardized CVs for the human omics assays across pre-flight, post-flight, and recovery time intervals. The most variable analytes are labeled at the top of each violin plot. CV is calculated across n = 4 independent subjects in 6 to 7 timepoints for p = 16283, 8464, 527, 1765, 656, 203 analytes respectively from a to f. The center dot represents the median, and the black line shows the range of the first and third quartiles.

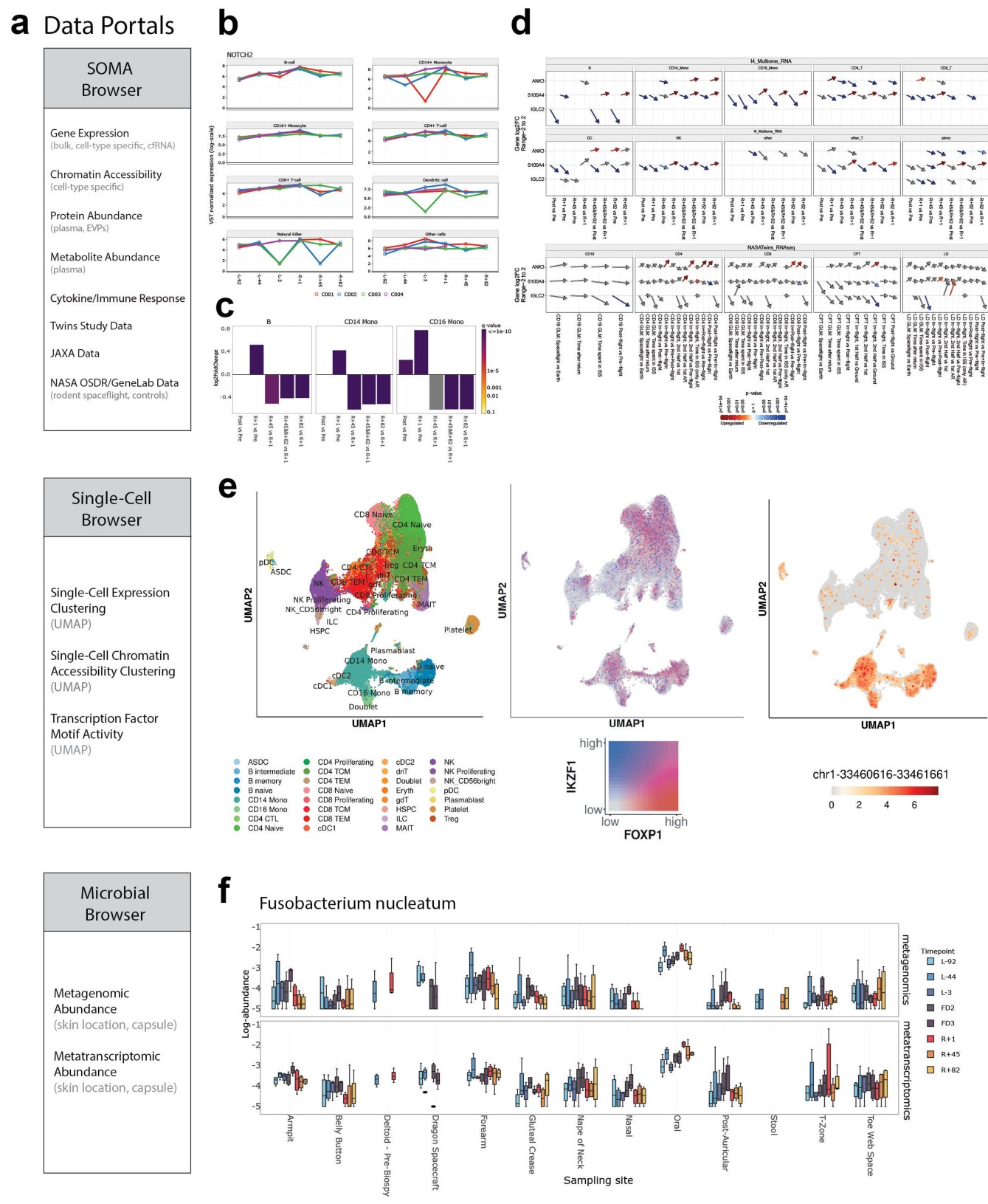

**a** Data Portals

**Extended Data Fig. 10 | SOMA Web Portals. (a)** Three different web portals were created: SOMA Browser, Single-Cell Browser, and Microbial Browser. The SOMA Browser includes **(b)** gene expression and protein abundance measurements in line chart and volcano plot formats, **(c)** log fold change calculations, and **(d)** comparison of DEGs across different contrasts, assays, studies, and organisms (statistical tests dependent on comparison, please see website). **(e)** The Single-Cell Browser enables visualization of cell type specific information, including gene co-expression and ATAC-seq region peak visualization. **(f)** The Microbial Browser includes microbial abundances across timepoints from different annotation databases for metagenomic and metatranscriptomic datasets (n = 4 independent subjects for 8 timepoints and 12 collection sites, or n = 9 environmental samples for 4 timepoints). The center of the boxplots represent the median, the box hinges encompass the first and third quartiles, and whiskers extend to the smallest and largest value no further than the 1.5x interquartile range (IQR) away from the hinges.

Cem Meydan
Christopher E. Mason

# Reporting Summary

## Statistics

For all statistical analyses, confirm that the following items are present in the figure legend, table legend, main text, or Methods section.

| n/a | Confirmed | |
|---|---|---|
| ☐ | ☒ | The exact sample size (*n*) for each experimental group/condition, given as a discrete number and unit of measurement |
| ☐ | ☒ | A statement on whether measurements were taken from distinct samples or whether the same sample was measured repeatedly |
| ☐ | ☒ | The statistical test(s) used AND whether they are one- or two-sided *Only common tests should be described solely by name; describe more complex techniques in the Methods section.* |
| ☐ | ☒ | A description of all covariates tested |
| ☐ | ☒ | A description of any assumptions or corrections, such as tests of normality and adjustment for multiple comparisons |
| ☐ | ☒ | A full description of the statistical parameters including central tendency (e.g. means) or other basic estimates (e.g. regression coefficient) AND variation (e.g. standard deviation) or associated estimates of uncertainty (e.g. confidence intervals) |
| ☐ | ☒ | For null hypothesis testing, the test statistic (e.g. *F*, *t*, *r*) with confidence intervals, effect sizes, degrees of freedom and *P* value noted *Give P values as exact values whenever suitable.* |
| ☒ | ☐ | For Bayesian analysis, information on the choice of priors and Markov chain Monte Carlo settings |
| ☒ | ☐ | For hierarchical and complex designs, identification of the appropriate level for tests and full reporting of outcomes |
| ☒ | ☐ | Estimates of effect sizes (e.g. Cohen's *d*, Pearson's *r*), indicating how they were calculated |

*Our web collection on statistics for biologists contains articles on many of the points above.*

## Software and code

Policy information about availability of computer code

| Data collection | No software was used for data collection |
|---|---|
| Data analysis | ani-cluster (https://github.com/snayfach/MGV/tree/master/ani_cluster)<br>Bakta (v1.5.1)<br>bbtools (v38.92)<br>BCFtools (v1.9)<br>bedtools (v2.29.2)<br>Bowtie2 (v2.2.3)<br>bracken (v2.6.2)<br>BWA MEM (v0.7.15)<br>Cell Ranger (v6.1.1)<br>CheckV (v0.8.1)<br>CheckM (v1.2)<br>deepTools (v3.5.1)<br>DESeq2 (v1.36.0)<br>Diamond (v2.0.14)<br>FastQC (v0.11.9)<br>fgsea (v1.22)<br>GSVA (v1.42.0)<br>HOMER (v4.11)<br>Kraken2 (v2.1.2) |

limma (v3.52)
m6Anet (v1.1.1)
MACS2 (v2.2.6)
MassProfinder 8.0
MetaQUAST (v5.0.2)
MetaSPAdes (v3.14.3)
methylKit (v3.18)
minimap2 (v2.24)
MMseqs2 (v13.4511)
MultiQC (v1.13)
pheatmap (v1.0)
PycoQC (v2.5.0.21)
samtools (v1.0, v1.9)
SARTools (v1.8.1)
SEACR (v1.3)
Sentieon TNscope RNA-seq variant pipeline (v202010)
Seurat (v4.2.0)
SnpEff (v4.3)
subread (v2.0.1)
trimmomatic (v0.39)
Varient-Effect-Predictor VEP (v107)
vegan (v2.6.2)
VGenes (vPre-release3)
vironomy (https://github.com/b-tierney/vironomy)
XTree (v0.92i)

Additional codes and scripts can be found in https://github.com/eliah-o/inspiration4-omics

For manuscripts utilizing custom algorithms or software that are central to the research but not yet described in published literature, software must be made available to editors and reviewers. We strongly encourage code deposition in a community repository (e.g. GitHub). See the Nature Portfolio guidelines for submitting code & software for further information.

# Data

Policy information about availability of data

All manuscripts must include a data availability statement. This statement should provide the following information, where applicable:
- Accession codes, unique identifiers, or web links for publicly available datasets
- A description of any restrictions on data availability
- For clinical datasets or third party data, please ensure that the statement adheres to our policy

Datasets have been uploaded to the NASA Open Science Data Repository (osdr.nasa.gov; comprised of NASA GeneLab and the NASA Ames Life Sciences Data Archive)

OSD-569
https://osdr.nasa.gov/bio/repo/data/studies/OSD-569/
OSD-570
https://osdr.nasa.gov/bio/repo/data/studies/OSD-570/
OSD-571
https://osdr.nasa.gov/bio/repo/data/studies/OSD-571/
OSD-572
https://osdr.nasa.gov/bio/repo/data/studies/OSD-572/
OSD-573
https://osdr.nasa.gov/bio/repo/data/studies/OSD-573/
OSD-574
https://osdr.nasa.gov/bio/repo/data/studies/OSD-574/
OSD-575
https://osdr.nasa.gov/bio/repo/data/studies/OSD-575/
OSD-630
https://osdr.nasa.gov/bio/repo/data/studies/OSD-630/
OSD-656
https://osdr.nasa.gov/bio/repo/data/studies/OSD-656/

In addition the JAXA Cell-Free Epigenome (CFE) Study data and results were obtained from accession OSD-530.
https://osdr.nasa.gov/bio/repo/data/studies/OSD-530/

Processed data are also available for visualization in the following web portals.
https://soma.weill.cornell.edu/apps/SOMA_Browser/
https://soma.weill.cornell.edu/apps/I4_Multiome/
https://soma.weill.cornell.edu/apps/I4_Microbiome/

## Human research participants

Policy information about studies involving human research participants and Sex and Gender in Research.

| | |
|---|---|
| Reporting on sex and gender | Sex information was collected and annotated in the NASA OSDR repository for each sample collected. |
| Population characteristics | The crew member composition was of two races and ages ranged from 29-51. |
| Recruitment | Participants were recruited by SpaceX and mission commander Jared Isaacman through various task-specific criteria and raffle processes. There are no self-selection biases relevant to sample the collection protocols outlined in this paper. |
| Ethics oversight | All subjects were consented at an informed consent briefing (ICB) at SpaceX (Hawthorne, CA), and samples were collected and processed under the approval of the Institutional Review Board (IRB) at Weill Cornell Medicine, under Protocol 21-05023569. All crew members have consented for data and sample sharing. |

Note that full information on the approval of the study protocol must also be provided in the manuscript.

# Field-specific reporting

Please select the one below that is the best fit for your research. If you are not sure, read the appropriate sections before making your selection.

☒ Life sciences    ☐ Behavioural & social sciences    ☐ Ecological, evolutionary & environmental sciences

For a reference copy of the document with all sections, see nature.com/documents/nr-reporting-summary-flat.pdf

# Life sciences study design

All studies must disclose on these points even when the disclosure is negative.

| | |
|---|---|
| Sample size | No sample size calculations were performed. The entire Inspiration4 crew was profiled, which was limited by the size of the Dragon capsule (n=4). |
| Data exclusions | No data has been excluded. |
| Replication | Replication tests are difficult as mission parameters cannot be repeated. Where possible, data validation was performed via western blots to validate proteomic findings. Western blots were done only once due to limited material. |
| Randomization | No randomizations were performed, all of the subjects were profiled longitudinally and there were no experimental groups or interventions that separated the subjects. |
| Blinding | Blinding was not possible because all subjects were astronauts in the same crew. |

# Reporting for specific materials, systems and methods

We require information from authors about some types of materials, experimental systems and methods used in many studies. Here, indicate whether each material, system or method listed is relevant to your study. If you are not sure if a list item applies to your research, read the appropriate section before selecting a response.

## Materials & experimental systems

| n/a | Involved in the study |
|---|---|
| ☐ | ☒ Antibodies |
| ☒ | ☐ Eukaryotic cell lines |
| ☒ | ☐ Palaeontology and archaeology |
| ☒ | ☐ Animals and other organisms |
| ☒ | ☐ Clinical data |
| ☒ | ☐ Dual use research of concern |

## Methods

| n/a | Involved in the study |
|---|---|
| ☒ | ☐ ChIP-seq |
| ☒ | ☐ Flow cytometry |
| ☒ | ☐ MRI-based neuroimaging |

## Antibodies

| | |
|---|---|
| Antibodies used | Abcam H3K27ac (CAT:ab245911)<br>Epicypher H3K4me1 (SKU:13-0040) |

Cutana Rabbit IgG antibody (SKU: 13-0042)

Cell Signaling, human CD9 (rabbit monoclonal IgG, CAT:13174S)
Santa Cruz Biotechnology, Galectin 3 Binding Protein (LGALS3BP) (mouse monoclonal IgG, CAT:sc-374541)
Abcam, Fibronectin (FN1) (rabbit polyclonal IgG, CAT:ab2413)
Cell Signaling, β-actin (rabbit monoclonal IgG, CAT:8457S)
Jackson Laboratory, horseradish peroxidase (HRP)-labeled IgG goat anti-rabbit or goat anti-mouse

Validation

H3K27ac and H3K4me1 antibodies were compared to the Cutana Rabbit IgG antibody, which served as a negative control.

Information from the manufacturer:
Abcam H3K27ac (CAT:ab245911)
This antibody meets the Abcam promise guarantee for immunocytochemistry and immunofluorescence applications. and manufactrer notes it specifically binds to K27ac alone and also when S28 is phosphorylated.

Epicypher H3K4me1 (SKU:13-0040)
This antibody meets EpiCypher's "SNAP-ChIP® Certified" criteria for specificity and efficient target enrichment in a ChIP experiment (<20% cross-reactivity across the panel, >5% recovery of target input). This antibody binds to H3K4me1 and no significant cross reactivity with other lysine methylations in the EpiCypher SNAP-ChIP K-MetStat Panel (EpiCypher 19-1001) is detected.

Cell Signaling human CD9 antibody is noted as recognizing endogenous levels of total human CD9 protein and certified for western blot use by the manufacturer.

Santa Cruz Biotechnology Galectin 3 Binding Protein antibody is noted as being suitable for detection of Galecting-3BP/Mac-2BP of mouse, rat and human origin by western blot use by the manufacturer.

Abcam Fibronectin antibody is covered by the Abcam guarantee for use in Western blot and the manufacturer notes that it detects a band of approximately 285 kDa (predicted molecular weight: 262 kDa).

Cell Signaling β-actin antibody is marked for Western blot use, and the manufacturer notes that iut recognizes endogenous levels of total β-actin protein. Due to the high sequence identity between the cytoplasmic actin isoforms, β-actin and cytoplasmic γ-actin, this antibody may cross-react with cytoplasmic γ-actin. It does not cross-react with α-skeletal, α-cardiac, α-vascular smooth, or γ-enteric smooth muscle isoforms. Species Reactivity: Human, Mouse, Rat, Monkey, D. melanogaster, Zebrafish

