## [Peer Review File · Nature]

Manuscript Title: The space omics and medical atlas (SOMA) and international astronaut biobank

Reviewer Comments & Author Rebuttals

Reviewer Reports on the Initial Version:

Referee #1 (Remarks to the Author):

In “The Space Omics and Medical Atlas (SOMA): A comprehensive data resource and biobank for astronauts,” Overbey et al. present a multi-omics and multi-sample resource that describes molecular phenotypes obtained from astronauts from a short-term high-elevation space mission. Data from different types of specimens collected before, during, and after the mission were analyzed, intending to reveal transient and persistent alterations across other classes of molecules. An app was developed to summarize the data and allow users to browse the different data types.

This study is undoubtedly a unique and rich resource of general interest. Studies with space travel participants will inform about the molecular burdens of short and presumably longer-term trips into space. I applaud the authors for their achievement, and I thank the editors for the possibility of reviewing this work.

Nonetheless, and despite the exclusivity of the study, there are some concerns to be addressed:

Statistically, the study presents a longitudinal multi-omics analysis of N=4 (2 females/2 males). Thus, it is highly skewed toward a high number of data points on very few samples. Multiple testing corrections will present a high bar for many of the findings to be deemed “significant.” Even if findings pass this threshold, the observations might be difficult to replicate or generalize. It would be appreciated if the results could be benchmarked on existing knowledge about causal effects and consequences for the highlighted molecules. Nonetheless, the presented study offers a unique opportunity and starting point to develop N=1 statistics and apply these accordingly (see comment about the app).

Physiologically, the authors see the main reasons for altered molecular profiles are microgravity and radiation (line 138). There, however, is no evidence that changes were not induced due to the lack of movement during space travel, stress, differences in air supply, and nutrition intake. Ideally, any space travel study should be designed with a second (or the same) group remaining on earth and experiencing similar conditions (e.g., in a simulator or airplane). This would reduce the number of differences and uncertainties and ultimately allow better reference to any changes. Similar questions arise about the time of space exposure. Is such data available, or can data from other space travel studies be used to support some of the observations?

Molecularly, the authors should aim to build their statement of short, medium, and long-term consequences of space travel on the functional categories of the molecular data. It remains unclear

which organs would be affected by space travel and why. Not all molecules will be equally “reactive” to the change in environment. Metabolites and lipids may be highly influenced by nutrition, and blood protein levels may be affected by liver and kidney function, and blood cell counts. In addition, a lack of exercise might alter muscle mass and protein turnover. These connections must be clarified.

Further, the observed brain-specific proteins could result from changes in the liquor during gravity, but no CSF samples were available to support such data. At last, molecules like IL-6 or IL-10 have been reported for various (inflammatory) conditions. Hence, space travel alone may not be causal but induce micro-inflammations. If so, can this be explained by RNAseq data of the blood cells? It might be helpful to focus on blood because many molecular measurements were obtained from this sample type (on earth). Could continuing micro-inflammations lead to changes (or even exhaust) in immune functions?

A complete sampling of the astronauts was only possible on earth, and it remains unclear why. A compendium paper (Overbey et al. Collection Protocols Paper [Nature Methods]) that describes the protocols was unavailable. Hence, assessing the impact of sampling on the molecular data was impossible. Still, some questions emerged. One relates to the skin microbiome: What is the effect of hygiene for space travel on these measurements? Were the samples collected from different areas or adjacent spots? Exposure to clothing or air, temperature, and light might influence this microbiome. The other question relates to the influence of the sample quality to separate the “sampling effect” from a “physiological effect.” Data on precision and variance would help to understand this.

Inter-individual differences between the astronauts must be assumed. Hence their molecular profiles at baseline are very likely to be unique. In fact, any data should be referenced to the clinical traits (e.g., BMI, ALT, CRP...) and considered variable when judging the longitudinal trajectories. For example, is there a co-variation of weight with any variables (note: Extended Data Figure 4 was not convenient to extract meaningful information from)? Which fraction of the data fluctuated between the three phases (pre, recover, long)? Instead of looking at changes in absolute values, the heterogeneity of variance in the molecular data can be another informative measure because it is less limited to expecting the same changes in all subjects at the exact sampling and allows revealing/shortlisting alterations that occur more generally. The mentioned analyses and connections should be computed and implemented into the app.

In addition, it would be more relevant to assume that each astronaut responds differently to the space trip. The authors should attempt to identify different response phenotypes. This could assist future studies in testing ways to improve the preparation and recovery of space travel.

The app is an excellent tool but should host all of the molecular data in the manuscript. Simple longitudinal plots will help the audience to understand alterations on an inter-individual level. It would also be helpful to enable the clustering of all the different data types using PCA and UMAP. Please allow users to select features of interest. Also, there is a need to generate qq-plots for each data type and implement a statistics and QC section into the app. It will be essential to judge the findings on accuracy and precision. The study has otherwise used established methods, and inviting

more methods to be used in the future will improve the validity of the data.

Instead of sharing data, is it possible to submit analysis codes to the project and present such on the app? A “call for codes” could be an attractive initiative to raise interest in the data as an N=4 challenge (see Kaggle).

Large-scale efforts often struggle with finding a core observation to present, tend to describe the data from many different angles consecutively, and choose to present observations rather than the biology behind these. This makes reviewing every detail of such papers difficult. The authors should reconsider the flow of the manuscript and find better connectors between the different data types. It is not one data type that will explain all the effects and consequences on human health, but it is the connections between them that will assist us in learning more about the system biology of space travel.

The authors have cited relevant literature but could add more recent multi-omic studies that have aimed to integrate longitudinal profiles in health and diseases. This will help to demonstrate the possibilities and limitations. The manuscript is otherwise clear and well-written.

Any sample collection tube should be listed with the product and LOT number.

Which sample volumes are available, and which MTAs apply when accessing the material?

Referee #2 (Remarks to the Author):

With an increasing number of commercial spaceflights and a growing interest in long-duration spaceflight projects, understanding the molecular impact on astronauts is becoming highly relevant. In this study, Overby et al. describe a comprehensive resource of high throughput (omics) data covering the key events of space flight and extended follow up from the SpaceX Inspiration4 mission. The authors confirm previously described findings such as telomere elongation and elevation of cytokine levels, illustrating that commercial spaceflight missions provide a useful source for such data. The manuscript goes beyond existing work by providing several novel, high-dimensional data types, including spatial transcriptomics, RNA modifications, single cell RNA-Seq and ATAC-Seq, obtaining new insights in particular into the long term molecular impact of space flight.

One of the key novelties of the manuscript is the breadth of technologies used for profiling the samples. This design allows the authors to quantify the impact of space flight on different aspects such as the skin, microbiome, or gene expression over the different time points. Partially due to this breadth of data, the level of detail for each data type is limited, sometimes focusing on more descriptive findings regarding the global impact of space flight on each molecular data type. As described by the authors, additional data points from longer follow up or alternative space flight missions will be needed to confirm these observations and allow for more detailed analysis and interpretation. The manuscript and data resource are a foundation for collection of such data in the future, and as such data are likely to become more frequent, it provides a timely description, analysis, and resource to build on. The manuscript is well written and motivated, and (with a few exceptions outlined below) accessible to a broad audience. Due to the breadth of assays and the general interest in this topic, I expect this work to appeal to scientists from diverse disciplines.

Major comments:

1) Statistical tests, data description, and methodology are not always sufficiently clear

The manuscript presents a large data set covering different time points, different samples, and different data types. While the data overview and analysis overview is well presented and summarised, I found the details to be partially insufficient:

- The number of samples per data type and time point is not clearly described. Generally it seems that 4 astronauts were profiled for all data types and time points, and in many figures as well as in the supplementary tables the data and QC metrics from each sample are shown. However, when reading the text I was searching for this information from the beginning. I would suggest to add that description somewhere clearly in the text (and if possible add to Figure 1). I assume that all data was generated for all 4 astronauts so this is probably a trivial addition, but it is important to understand how the analysis was done.

- For some analyses and statistical tests it is unclear how samples were combined or how the tests were performed. This was particularly difficult to assess for the single cell data, where the authors seem to pool all the samples (without describing sample numbers). However, this was also unclear for other analysis and statistical tests in the manuscript (for example: the BCR mutation analysis,

how was this tested for significance? Is this based on the mean of the samples? Was the paired structure in the data considered for this test?; line 291 "GESECA padj..." how was this tested? How is the p-value related to each term that is mentioned?; Line 220/Extended Data Fig. 5c, The data suggests a very strong increase that should be statistically significant across all samples, but this was tested per astronaut, which then was not significant. I could not find out how the significance estimate was calculated per astronaut here (and why not across all astronauts);...).

- The authors summarise the overall analysis strategy in the main text for each analysis, which I very much appreciate and which I think helps to understand the data processing and analysis sufficiently for most readers. However, the methods section and supplementary text do not provide much more information beyond this general overview, they are often not sufficient to reproduce the results. Furthermore, the method section combines experimental methods with computational methods description, which makes it very difficult to find the relevant details. I would recommend to separate experimental and computational methods using different sub-headers. The parameters, software version, and any thresholds or filter that were applied should also be described. Ideally, in addition to those changes, code is made available by the authors that could generate results and figures from the publicly available, processed data, but given the breadth of data sets and analyses I understand that this might not be feasible.

2) Analysis-specific comments

2.1) single cell analysis

- This analysis was the only analysis where the individual samples were not clearly shown or described. For single cell data, a higher level of variation might be expected across astronauts, but it would still be helpful to investigate this. Are the observations and results described by the authors consistent across all astronauts?
- The differential expression analysis using the single cell data was not clearly described. Was the paired structure of samples (same astronaut at different time point) considered when analysing differential gene expression? Was this done by pooling all samples from the same time point? Did the authors consider the astronaut as a covariate in the differential expression analysis?
- the authors say that the number of DEGs returns to baseline levels. However, it is not clear what baseline levels are, and the figure that is referenced does not show any baseline differential expression numbers either.
- what are the genes that change expression? Is there any enrichment of specific categories?
- Regarding the illustration of differentially expressed genes (Figure 2d): (a) the authors provide names for the comparison, but as these names are not intuitive one has to jump between figure 1 and figure 2 to understand this analysis. Maybe a more descriptive title could help, or an additional illustration in Figure 2d. The data does not show the expected differences, which made it particularly difficult to understand (see point below)
- Regarding the differential expression analysis (Figure 2d): R1 and R82 seem more similar than R45 in the longitudinal profile. The longitudinal heatmap looks very similar to the flight profile heatmap, where the R+1 and R+82 are also more similar. Why is that the case? It's not clear whether this reflects properties of the data related to the samples, errors in the analysis, or possibly QC issues, and if that impacts the claims that can be made from these data. Showing consistency across samples could be one way to validate these results.

- In the TFBS enrichment analysis, many transcription factors from the same family are highlighted. Many motifs from such transcription factors might be very similar, and the precise transcription factor might not be identifiable from the sequence. I would suggest to show these individual motifs in the supplement, and if they are very similar, they could be combined into motif groups rather than listing all transcription factor names (eg the CEBPX family might be very similar in terms of motif)
- some QC plots or analysis for the snATAC-seq data would be helpful. Does the data cluster by cells as expected? Is there an agreement between expression and chromatin accessibility? Do the samples from different astronauts show similar results?

2.2) gene expression analysis

- The details in the RNA modification analysis are not sufficient to fully reproduce the results (similar to other sections, see point above). For example, which sites are selected to be considered in the comparative analysis? What about sites which were not detected in a samples due to lower read numbers, how was that considered in this analysis?
- The supplementary table shows a substantial variation in read length, with 3 out of 4 R+1 samples having much longer reads. Could this impact the analysis?
- The authors perform a gene set enrichment analysis for gene expression, but there is only limited information provided. How was this analysis done? The full results could be shown as a supplementary figure. Is there any enrichment of genes that show differences in modifications?

2.3) spatial transcriptomics

- The different regions are introduced and Fig 5b is referenced, but the relation between Figure 5b and these regions is not clear. Maybe additional labels in Figure 5b might help, or at the minimum an explanation in the legend. For someone who is not familiar with skin it is unclear how these figures show specific regions.
- The association of spatial transcriptomics with microbiome changes was difficult to follow. First the authors describe the spatial transcriptomics data, then the microbiome association is described, and then the authors go back to describing the results from the spatial transcriptomics data. The authors describe the analysis for each data type in detail, however, the microbiome data was never described when it is introduced at this point. The analysis itself is a good example of how the different data sets could be combined (see comment below about data integration/comparison). However I found this part difficult to follow, this could be restructured for more clarity
- I could not match the text (lines 423-429) to the figure (the names do not match). This part either needs clearer labels or additional information.

2.4) Microbiome analysis

- The microbiome section in the manuscript describes the data processing and analysis in details, but the findings and the interpretation of these findings are not very clear (partially due to missing figure references). For example, the authors describe their approach to build a non-redundant protein catalog, but it's not clear how this is used, or what the conclusions are. The paragraph that follows describes some findings on variation at the species level as a function of flight (referencing figure 6b and Figure 6). It's not clear how these statements are shown on these figures. It's possible that the authors wanted to refer to Figures 6e and f, which show the global overview. However these figures do not quantify any observed variation in relation to sample time/flight.
- The authors quantify changes in microbiome using linear mixed models in the next paragraph. Which organisms contribute to these changes? The authors write that this indicates a rapid microbiome transfer, can this be shown with the data? This appears to be a consequence of contact between astronauts, not a consequence of space flight.

3) Data Quality

The authors have performed detailed Quality Control (QC) and made QC metrics for many data sets available, which is very helpful for a data resource that includes such a diverse set of data types. However, the supplementary tables currently only provide this information for some of the experiments. The QC metrics and criteria for sample exclusion (if any) could be described in more detail in the methods section for each data type, and they could be made available for each sample in a supplementary table, similar to the existing supplementary tables.

4) Multi-Omics integration/comparison

- The data resource describes many different data types, but the authors largely analysed each data type separately (often arriving at similar conclusions related to the impact on space flight on molecular signatures). A comparison of these results across data sets could help to provide a global overview, and possibly strengthen support for some claims on the long term impact on molecular signatures. Figure 1b shows that many comparative analyses are available for many different data sets, do these show the same patterns?
- The authors describe that some data types show an impact of space flight which returns quickly to pre-flight levels, whereas other data types show more long term impact. These statements could be quantified for each data type, and presented in a global overview that assesses the impact on molecular changes of space flight over time across the different data modalities.

5) online data visualisation

The online data visualisation tool is not sufficiently documented, or possibly not fully functional: Following the links in the manuscript I land at a page that says "Hello" (wrong link) or "I4_Multiome" (correct link). The portal then allows me to look at PBMC data, but not all data sets as described in the text (line 197-199). For example, direct RNA-Seq, spatial transcriptomics, or microbiome data is not included it seems. Furthermore, the different tabs (eg gene expression, TFBS, ...) led to the same figures based on gene expression with exactly the same description. Either the other analyses are not properly linked, or they are not properly described.

Minor comments:

- figure 2c: The title says "DAP counts", should this be DAR counts?
- figure 4h is referenced as 4g in the text, Figure 4g seems to be referenced as 4c
- line 423 the figure reference seems a bit misplaced here (Fig 1b).
- "Skin-microbiota interaction changes were also observed resulting from spaceflight exposure" I am not sure if the authors can claim that this results from space flight based on the data, I would recommend to reformulate this, eg "were observed after spaceflight exposure"
- figure 6, the y axis labels are missing
- figure 6 e and f are not referenced in the text
- citations are missing in discussion when previous findings are referenced (first paragraph)
- the authors could explore differential isoform usage, splicing, or polyA tail length with the long read direct RNA-Seq data. Isoforms are even mentioned in the introduction.

Referee #3 (Remarks to the Author):

The manuscript “The Space Omics and Medical Atlas (SOMA): A comprehensive data resource and biobank for astronauts” provides a dataset with a large number of clinical features and multi-omics of 4 astronauts before and after a 3-day spaceflight. This dataset also covers time series spanning 92 days before to 82 days after spaceflight, which provides a strong foundation for deep analysis of short- and long-term effects of spaceflight. The authors use an array of almost all known omics technology to almost maximize the amount of information that could be gained from this study.

This manuscript functions as an inventory of available data and follow up studies will also increase its impact. Some additional analysis could improve the manuscript.

(1) In the section “Proteome and Metabolome of Blood Plasma”, the authors provide heatmaps of plasma proteomics, metabolomics and Extracellular Vesicle and Particle (EVP) proteomics in Figure 4. The pattern of the heatmaps give the appearance of some structure, however there is no further discussion in the main text. The authors provide pathway enrichment information of the most variable metabolites in Extended Data Figure 6, but these results are not connected to the heatmap. The authors could add pathway information alongside the heatmaps to more clearly illustrate how those pathways change during spaceflight.

Similarly, for the EVP proteomics, the authors provide liver and brain enrichment information, but this information is not connected to the information in heatmap. Furthermore, there lacks pathway enrichment in plasma proteome analysis. Similar analysis could be considered.

2) Ease of accessibility of the data is the key for the impact of this study as it is expected these data will be used in follow up studies. The authors should state clearly in the manuscript how they intend to make the data available (preferably an easy to use website)

Minor:

(1) Some figures in the manuscript have a low resolution, making them difficult to interpret, (e.g., Fig 2D, 2E, 6B, 6E, 6F).

Referee #4 (Remarks to the Author):

This reviewer has been impressed when reading this manuscript of Dr. Overbey and colleagues on „The Space Omics and Medical Atlas (SOMA): A comprehensive data resource and biobank for astronauts“.

This valuable approach of a high resolution investigation taken up by this consortium is unprecedented and will allow also many follow up experiments and analyses.

This very holistic investigation on molecular and cellular changes as a function of spaceflight as performed in a crew of four (2 female, 2 male) will direct new research approaches and will deliver added values well beyond the initial investigation as presented. The multi-omics- approach linked to some clinical profiles pre-flight, in-flight, recovery, and longitudinal responses is in its limitations well designed. The i4 mission crew was in space for about three days and some signatures of a longer duration spaceflight mission could be seen here as well. The follow up was as interesting, since many effects went back to normal while other features (gene expressions, also proteins) seem to persist longer post-mission. This multiomics approach and application of the „full battery“ of modern analytical tools aimed to provide a new and innovative biological analysis approach especially also when compared in the future also with data from other conditions.

This reviewer was delighted to read, that all datasets can be mined through an interactive data portal, maintaining the spirit of research and especially space research to have not limits. Also in the light of patient care, drug effect studies and as for the success of upcoming lunar and exploration-class missions to Mars these studies can be helpful.

Detailed comments:

While the manuscript has – to the best of this reviewer (though in some areas limited knowledge of this reviewer, as indicated to the editor) no flaws which should prohibit its publication per se some major critic has to be raised since the significance as by the conclusions drawn, however, have induced some major concerns that are explained here.

The abstract is well written as is the text and supplementary material. It is a huge piece of data sets that are included and explained and therefore suffers from more integrative views on the data especially including a more critical view and discussion of it. This reviewer well acknowledges the challenges to extract all the information and to well balance their impact in one document. However, some key elements of a scientific evaluation however went therefore underrepresented.

- 1.) It lacks information about the subjects investigated and the referenced paper on the methods is not accessible. They shall be included here irrespectively of other publications.
- 2.) it is still a very small cohort and of two biological sexes, mean values are shown and sex differences not adressed.
- 3.) barely significant data are highlighted (as for telemore) and very little discussion and efforts are made to explain this moderat though surprising effect of a 3 days flight. This interesting topic shall be specifically adressed though also its likeliness to be affected so early, as well if that differs on the physical state / regular excercise, biological sex and so forth. Especially since exercise can reduce oxidative stress and preservation of DNA and telomeres as know for a bit (e.g. see Song et al Aging Cell. 2010; 9:607–615). The individual data points cannot be discriminated in Ext.Fig 6 C
- 4.) the proteo-metabolomics are so dependent on the individual conditions across several independent studies, key proteins and metabolites which are involved can be reguled by identified food compounds affecting key metabolites <https://doi.org/10.1159/000381349>
- 5.) The more and more established concept of targeting individual needs and health risk and therapies as termed „personalized medicine“ or „precision medicine“, respectively, can rely also on global and such integrative approaches for a single subject / patient / crew member. Especially this short flight can be an interesting model for personalized medicine because the deregulated pathways underlying these conditions. It is recommended to include this in the discussion. This is one of the major values of this study and the follow-up and shall deserve more space (now only mentioned once) p7/line126.
- 6.) also that the exposure to space for such a short time can barely be distinguished from an semi-acute stress model and here respective controls are warranted to make this distinction.
- 7.) The habits and way of life of this non-professional crew before and after mission, their experiences with more stressfull conditions of life, their motivations and other „soft“ factors shall at least be discussed in the context of data interpretations and conclusion. The stress hormonal impact can be huge and may be differential also on the preparatory phase as well as in the flight.
- 8.) Statistics with an n=4 are limited in their value given the high interindividual differences. In the light of the importance of this report to foster the tools to be applied in personalized medicine (and crew care), the individual and sex differencs over time and parameter should be always displayed and discussed, given the approval to hereby identify the individual by a chance of 50%.
- 9.) Control groups are announced but not further explained and a limitation section shall be included

Minor:

Figure resolution and readability is very limited (likely a matter of .pdf conversion)

Referee #5 (Remarks to the Author):

Overbey et al. profile multiple data sources from astronauts on the Inspiration4 mission, including data collected longitudinally pre-, mid-, and post-flight. From biomarker panels, the authors find an increase in inflammatory cytokines, in agreement with studies on longer duration flights. The authors also use single-cell expression and accessibility measurements to show long-duration changes in the epigenomes of specifically monocytes, including an increase in accessibility at sites containing binding sites for general stress and immune response TFs. Proteomics shows a dysregulation of some brain-related proteins, while spatial transcriptomics showed signatures of inflammation and KRAS signaling in the skin. Microbiome analyses showed transfer between astronauts in the exposed areas, with relatively minor domain-level changes. Most of the observed changes stabilized to baseline after re-acclimation to earth. The authors make these data available via a biobank/web portal, and are accumulating more data from future missions.

The authors have collected a large amount of diverse data on these astronauts, and in principle, multiomic analysis may have the potential to both understand the systems-level changes to humans as they are exposed to new environments and to develop plans to mitigate the risks for future missions. However, the lack of a truly integrative analysis of these data, the relatively limited amount of time of exposure, the relative lack of statistical power that would allow for concrete conclusions substantially limits our enthusiasm for this work.

Major Critiques

Lack of significant findings and integrated analyses - The authors have collected a large amount of data, but these measurements are not really compared or integrated in a meaningful way (i.e. analysis of each data type is on an island of its own). The work would be substantially improved if these data types could “talk to one another” and potentially support the conclusions and findings that are made with different data types separately. Ideally, some connection to the molecular changes observed might be discussed in the context of the physiological changes that are already known as side effects of space travel. As the manuscript stands, there are many individual datasets presented with very little explanation for why those -omes were collected, or how the different -ome datasets relate to one another, even at a very basic level (e.g. do changes in regulatory sequence accessibility in monocytes correspond to changes in regulated gene RNA abundance?). Furthermore, there appear to be very few major changes in any data type and no (or very few) significant findings reported. The many -omes are not synthesized together to create a comprehensive picture of what happens to molecular human physiology in space flight, which is, in the end, what an astronaut -ome might be expected to provide.

Limited experimental conditions - To be able to make claims about how human physiology changes in space flight, one might be especially interested in molecular changes that occur on timescales longer than 3 days. Furthermore, it is unclear what changes are occurring because of being in confined quarters and eating the same food, vs. being in a microgravity etc. These limitations to the data corpus substantially limit its utility.

Lack of motivation and functional interpretation of results - It is not clear why the authors collected the data they did, or what the few changes they observe in the data mean. Additionally, the analysis of these data is relatively superficial, and does not integrate independent external data sets that might provide a more concrete biological interpretation. For example, the authors claim that there are some differential accessibility peaks in monocytes and that TF motifs are enriched in the motifs. Are the changes similar to other documented monocyte activation? Do the motif enrichments make sense with monocyte physiology or correspond to changes in monocyte function? Is it happening in all monocytes? Is this the expected immune response to people living in close quarters? Similarly, analysis of m6A abundance is likewise quite superficial. Statements such as 'some transcripts change in m6a abundance' do not provide any biological insight. What genes? Are these genes related, or are there commonalities in the m6A targets? Does this correspond to an increase or decrease in protein levels? Similar questions could be asked for all datasets, and if support for observations in one data set is not supported by other data sets that might be expected to provide support, this casts doubt on the reproducibility of the findings in individual molecular modalities.

Statistical analysis. Given the relatively limited number of individuals profiled (in contrast to the immense diversity of the data types), it seems that permutation testing would be a wise way to ensure that the differential elements that are reported are truly significant. We suggest permuting labels for all tests and building an empirical null distribution from these permuted data sets to provide a more stringent assessment of the number of significantly differential measurements actually observed.

m6A data: The authors present main text data in which individual samples show extremely strong differentially methylation signals (Fig 3C). Are these large scale changes for single individuals believable? Are there controls that convince the authors these observations are not technical artifacts? Is this a new finding in biology, or are these large % changes not significant? It is highly challenging to wade through all these data sets to obtain some sort of biological narrative the way it is currently presented.

Statistical significance of the telomere lengthening results - The authors emphasize in the abstract and discussion that they observe lengthening of the telomeres of the astronauts, in agreement with studies of astronauts exposed to longer duration space flight. However, if these results do not pass statistical significance in 3 of the 4 astronauts, and the data are not presented in any main figure, the authors should remove the centrality of these claims.

Presentation of figures - Many of the figures are hard to read and low resolution, making detailed examination of the data presented difficult, if not impossible. The authors should embed high-resolution panels (as well as fix some of the axis labels and styling that hamper interpretation, e.g. 2b/e).

Referee #6 (Remarks to the Author):

Overbey and colleagues present a dataset of 2,686 multi-omic samples collected from four crew members during the short low-orbital mission Inspiration4. Together, these samples span numerous omics strategies (e.g., genomics, RNA-seq, snATAC-seq, cfDNA-seq, epitranscriptomics, proteomics, metabolomics, metagenomics, metatranscriptomics) and contribute to the newly established Human Space Omics and Medical Atlas (SOMA), the largest molecular dataset to study the impact of spaceflight on the human body. After describing the dataset, the authors analyze its different omics dimensions and first recapitulate previously observed consequences of long-term spaceflight. This comparison to previous results, although not providing original conclusions, is welcomed as it validates the newly established atlas and demonstrates the relevance of studying short duration spaceflight (three days). In addition, the authors analyzed of the Inspiration4 to reveal new features they attribute to spaceflight, including increase in circulating brain-derived proteins, rapid microbiome transfer and KRAS signaling in the skin, as well as identified changes (e.g. in histone H2A levels) that persisted after long after the short spaceflight and thus warrant continued monitoring.

The dataset presented by the authors (n=4) covers many more omics dimensions compared to currently available resources (e.g., n=11 for telomere length and n=18 for blood plasma protein expression) and increases by four-fold the previous multi-omics study of one astronaut (the NASA twins study). Overall, there is a lot to like about the dataset itself, its originality and the impressive number of dimensions studied, which will likely make it of interest across the fields of human genomics, physiology, biomedicine and microbiology.

However, the presentation of the data and the interpretation and discussion of the results need to be substantially revised. In particular, a large part of the main text is dedicated to content akin to material and methods, while too little focuses on interpreting and discussing the results, both based on the analyzed data and in the context of prior literature. More importantly, the conclusions often lack sufficient statistical support, a limitation that is not appropriately discussed.

Major concern 1: limits of the study design

Although the complexity of the dataset is well introduced through a dedicated section as well as Figure 1, the authors do not mention or describe the cohort in the main text (crew of n=4). This information is crucial for the reader to properly place this work in the context of previous studies and evaluate the robustness of its conclusions.

In particular, the limited cohort size has important consequences for statistical significance. Specifically, statistically significant results are a valuable contribution over the NASA Twins Study but the absence of statistical significance should not be interpreted as an absence of difference, as it may rather result from a lack of statistical power. This point impacts the reporting and interpretation of results across the manuscript and should be addressed throughout (see detailed examples below).

The authors motivate the importance of the dataset by introducing the need to better understand the impact of spaceflight on the human race (line 105) and mention the role of inter-individual differences (line 574). As such, the authors should discuss the diversity and representativeness of the crew to better place the contribution of this work with regard to these aspects.

Although the longitudinal design and logistical constraints of the study are well understood, the lack of control (i.e., individuals that would go through the same treatment as the real participants except for the flight) prevents untangling the impact of confounding factors from spaceflight itself (e.g., similar environment, proximity or diet change). The authors do mention that the data is compared to other “control sets” (line 139), but do not mention those further. The authors should expand on whether these sets can help address potential confounding factors and should discuss any potential confounding factors when interpreting their results. In addition, the observed variations should be put in context of or controlled for “normal” (i.e. non spaceflight related) variation, where available, for interpretation.

Major concern 2: statistical support and validity of conclusions

The authors make strong statements with important implications (e.g., lines 225, 239-241, 253-254, 261-262, 730) based on absence of significance between recovery time points and pre-flight time points. With a cohort size of $n=4$, interpreting absence of significance as absence of difference is a strong assumption that warrants additional discussion before reaching such a conclusion. Especially when the conclusion is that changes revert back or return to baseline and therefore that, for a given parameter, spaceflight has limited consequences (e.g., “These data suggest that even short duration spaceflight [...] poses no obvious risk of mutagenesis and instability.” line 225).

In addition, the lack of detailed information about the statistical procedure used (Which test is used? is it paired? What exactly is being compared to what? Does the distribution of the data meet the assumption of the test?) hampers proper assessment of the reported results. Please add detailed statistical information wherever reporting p-values (e.g., lines 284, 318, 737, 805, 1258, 1282).

The authors report that “the telomere length of all Inspiration4 crew members showed telomere elongation during spaceflight (17-22% longer), though this trend was statistically significant ($p=0.029$, Student’s T-test) only in C003 and (Extended Data Fig. 5c)”, which contradicts the caption of the Extended Data Figure 5c “Although not statistically significant, telomere lengthening during spaceflight can be seen in the aggregate in-flight measurements and R+1”. Please clarify why these statements are contradicting each other, how significance is tested on a per-astronaut basis, and whether a paired Student’s T-test or a paired Wilcoxon test (which appear more appropriate based on the data in Extended Data Figure 5c) between pre-flight and in-flight time points would be significant. Furthermore, although there is a clear trend in shortening of the telomere during recovery, this can hardly be interpreted as mostly returning to baseline (line 503) as the last measurements show a ~10% increase in length.

The authors state that “The number of differentially expressed genes (DEGs) and differentially accessible regions (DARs) showed the largest change at the immediate post-flight time point, R+1, compared to the pre-flight time points, and then mostly returned to baseline levels during the recovery time points (Fig 2c).” (Lines 239-241). Please clarify how this statement is reached based on the data presented in Figure 2? My understanding is that hundreds of genes are differentially expressed across all cell types and hundreds (?) of regions were differentially accessible (at least for dendritic cells) between the recovery time points and the pre-flight time points. These values are also reported in the text (lines 241-243) and represent between 30 to 50% of the in-flight values, which hardly seems like a return to baseline.

In the section “Metagenomes and Metatranscriptomic Profiles of the Skin and Dragon Capsule”, the authors describe and analyze a set of metagenomes and metatranscriptomes and conclude that “Despite the overall stability of microbial domains, variation on the species-level was readily apparent, indicating the presence of community-level, within-domain shifts specifically as a function of flight (Fig 6b).”. This is however difficult to link to the data presented (Figures 6b,e,f) and should be supported by beta-diversity metrics and their visualization rather than the raw abundance profiles (which need to be visually compared). Interestingly, the authors report taxa that are differentially abundant or expressed pre- or post-flight (line 482, Figure 6 g-h). However, the representation of the data or discussion of those results does not mention what taxa are impacted.

In addition, the authors interpret the different responses of microbiome composition across body sites as a rapid microbiome transfer with the capsule and other crew members (line 494). However, the changes observed could also be explained by changes in species-level microbiome composition in response to the same environmental drivers. To support the claim that transfer is occurring, the authors should go beyond species-level profiling and conduct strain-level analyses (see <https://www.nature.com/articles/s41586-022-05620-1>).

Major concern 3: contextualization with existing literature

The authors place the Inspiration4 dataset and its analyses in the context of previous studies (the NASA twins study and the JAXA CFE, lines 118-120) but it is not immediately clear what is similar and what differs between those studies. Specifically, what are the different durations and altitudes studied? I note that the apogee altitude is later mentioned in the results for Inspiration4 and the ISS, but the link between the ISS and the previous studies is not explicit. In addition, additional studies with larger cohorts (e.g., n=11 for telomere length and n=18 for blood plasma protein expression) should be mentioned to properly contextualize the data presented.

Multiple parts of the manuscript lack contextualization, interpretation and/or discussion. However, I note that sections reporting on (epi)transcriptome of whole blood, proteome and metabolome of blood plasma and EVPs provide a better level of description, interpretation and contextualisation (e.g., line 291 “To construct the first epitranscriptome map...”, line 321 “Consistent with previous studies, we observe dysregulation...”), although the introduction to these sections could be better motivated (why are these parameters studied in the first place?) and significance of the results could be better communicated (in these sections or in summary). Overall, for the reader to better

understand the advance over the art would require an improved, harmonized contextualization of the results with existing literature throughout the manuscript sections (e.g., “Single-Nuclei Gene Expression, Chromatin Accessibility, and Immune Profiling of PBMCs”, “Metagenomes and Metatranscriptomic Profiles of the Skin and Dragon Capsule”).

This current limitation impacts, for instance, the significance of the analyses of microbiome data (of note, the four citations used to provide context in this section are self-citations, although the human microbiome has been a vibrant field of research for over a decade). In particular, the authors should address the following questions when introducing this section or discussing its results:

- What is the expected variability of the human microbiome composition at the taxonomic levels studied here? What is the rationale behind testing the domain-level stability of the microbiome? Are changes expected?
- How does the set of recovered genomes compare to the current set of genomes available for the human microbiome (e.g., <https://doi.org/10.1016/j.cell.2019.01.001>, <https://www.nature.com/articles/s41564-021-01011-w>)?
- How do the observed transfers relate to the expected transfer landscape of the human microbiome (<https://www.nature.com/articles/s41586-022-05620-1>)? How is the transfer different from living together in a household?
- Since several of the previous markers studied are indicative of inflammation, what could the impact of inflammation be on the microbiome composition?

Beyond taxonomic profiles, the authors built and briefly described a gene catalog. Although investigating the functional shifts in the microbiome as a function of flight is arguably of high interest, this catalog is not used in the presented analyses.

The authors also mention that “some of these genes correlated with the known, skin disease-related microbiome changes” (line 420). Please add a reference to support this statement. In addition, the statement is followed by the mention of family-level changes in microbial composition associating to the expression of specific genes. This association needs to be refined since microbial families encompass a large diversity, only a fraction of which would be associated with a disease phenotype. Note those correlations between the skin expression profiles and the microbiome composition (line 419) before introducing the microbiome data and its analysis (starting 433).

Figures and data presentation

Generally, the figures are very inconsistent, including the visualization, title and caption. Some of the visuals are excellent or have a lot of potential (e.g. Figure 1, 3, 4 and 5) while others are really difficult to extract information from (Figure 2 and 6). In addition, some titles only describe the approach while others convey a result (which is preferable in my opinion as it guides the reader). The reporting of p-values and/or lack of significance is often inconsistent and should always mention the statistical test used and what is being compared to what. More specifically:

Figure 2:

- In the caption of panel b, please specify what data is being projected.
- In panel c, my understanding is that each bar is an average of multiple time points, each of which is an average of four individuals. This should be reflected by the presence of error bars to capture the variability. Should it be DAP or DAR for the bottom subpanel? Please correct and adjust the caption accordingly. The y-axis scale of the bottom subpanel does not allow for proper evaluation of the data. Please switch to a log-scale or add a break.
- Panel d is too small to extract data from and the color codes are not explained in the legend nor caption. Maybe harmonize the visuals with figure 3b/c.
- Panel e is also difficult to read. Note that humans are not good at comparing area sizes and thus encoding a quantitative variable that way is not desirable. An alternative could be to use the split rectangle approach of figure 3 with two different color codes.
- In the captions of panels g and h, please specify (1) the statistical test being used and (2) what is being tested.

Figure 3:

- In panel c the triangle symbols are too small, please the same size as in panel b.
- What is the significance threshold and what is the statistical test used?

Figure 4:

- Consider replacing some of the color coding by textual information (akin to Figure 3).
- Please specify the statistical tests used.

Figure 5:

- The figure caption needs more details.
- Line 789: "Most of the ROIs showed good clustering around ROI types in both time points". Please provide statistical support to this claim (e.g. with a PERMANOVA).
- In panel c, my understanding is that both time points are projected in the same embedding and split for visualization purposes. If this is correct, please add the other time point as gray dots in the background of each panel to allow for robust comparisons between time points.
- In the caption of panel d, please clarify what is being correlated to what. Is it the expression of human genes involved in skin diseases? What measure of microbiome composition is used? Is it relative abundance? Why are genera and species levels mixed?
- Legends for panels e and f are cropped.

Figure 6:

- Figure 6c: In the current presentation, the colors (taxonomy) are difficult to read and make sense of.
- Figure 6e-f: These panels are difficult to read and interpret. I suggest considering dimension reduction approaches if the point is to communicate the differences in structures.
- Figure 6g-h: These results are particularly interesting but their interpretation is limited by the lack of annotations of the individual points (e.g. taxonomy). Please clarify what each point corresponds to (e.g. a species based on the kraken profiles?). Please specify the statistical tests used. Please provide details of the statistical methods used in the different analyses, either in a dedicated section or with each analyses (Some are available in supplementaries but should be more readily accessible).

Minor comments

Data sharing:

The authors have shared the data through a dedicated repository (OSDR) hosted at NASA. However, the identifiers provided in the table of the “Data Availability” section do not appear to be live at the moment. This data is indeed a valuable resource for the research community and should be accessible to a wide audience, which could be improved by sharing data through standard repositories. In particular, it would be valuable if non-sensitive sequencing data were available through INSDC compatible repositories (e.g., NCBI/SRA, EBI/ENA, DDBJ).

In addition, the authors provide a web application for users to explore and mine the processed datasets. Although I commend their effort to share the data in a meaningful way, it is currently difficult to use in particular due to a lack of documentation. Some questions that may help guide this include: what are the four different drop down menus at the top? Why are the categories all the same for the different types of data? In addition, from the manuscript, it reads as if all the data is available through the web portal, but it currently appears to be restricted to PMBCs.

Methodology:

The metagenomic and metatranscriptomic analyses presented in this work employ various bioinformatic pipelines that reflect a set of state-of-the-art and rigorous approaches. Having said that, I do have a few minor comments:

- Was the binning of metagenomic contigs done single-sample or cross-sample abundance correlation? (although the latter would be preferable it would at least be good to specify the approach taken)
- Although I commend the authors for the utilization of multiple approaches for taxonomic profiling, I note that they are all based on strategies comparing kmers to whole genomes. Thus, the value of this multi-approach taxonomic profiling would largely benefit from the inclusion of approaches relying on different strategies (e.g. marker gene based), which were also shown to perform best in a recent community-driven benchmark study (<https://www.nature.com/articles/s41592-022-01431-4>).
- Regarding the ad hoc approach used to assign viral taxonomy, how does it compare to previously published software? (e.g., vConTACT v.2.0; <https://www.nature.com/articles/s41587-019-0100-8>) Note that p-values are not consistently reported throughout the manuscript (p=XX, p-value = XX, p-value=XX, p value = XX, p.adj <XX).

Line by line:

Line 160: The acronym PBMC is used but not defined. There are other such examples in the manuscript (e.g. MYC, SNCA, KRAS).

Line 219: ‘C003’, which refers to the crew member, is not defined at that point in the text. I suggest to couple that with the introduction of the crew size.

Lines 219, 221-224: A sentence ends with “and” and in another VAF is both subject and object. Please check the writing here and throughout.

Line 477: Should the reference to figure 6 refer specifically to panels 6e-h?

Lines 512/524: Should “first-ever” have a different connotation than just “first”?

Line 1214: Authors contributions are very minimal and the contributions of many authors are not specified.

Author Rebuttals to Initial Comments:

We thank the reviewers for their thoughtful comments, and we have worked to address each specific question raised during review. We were glad to see that the reviewers were generally satisfied, and we agree with the reviewers that this paper will be of great utility to the research community. Our responses to each point are in blue, as well as the original comments from the reviewers.

Thank you.

Referees' comments:

Referee #1 (Remarks to the Author):

In “The Space Omics and Medical Atlas (SOMA): A comprehensive data resource and biobank for astronauts,” Overbey et al. present a multi-omics and multi-sample resource that describes molecular phenotypes obtained from astronauts from a short-term high-elevation space mission. Data from different types of specimens collected before, during, and after the mission were analyzed, intending to reveal transient and persistent alterations across other classes of molecules. An app was developed to summarize the data and allow users to browse the different data types.

This study is undoubtedly a unique and rich resource of general interest. Studies with space travel participants will inform about the molecular burdens of short and presumable longer-term trips into space. I applaud the authors for their achievement, and I thank the editors for the possibility of reviewing this work.

Nonetheless, and despite the exclusivity of the study, there are some concerns to be addressed:

Statistically, the study presents a longitudinal multi-omics analysis of N=4 (2 females/2 males). Thus, it is highly skewed toward a high number of data points on very few samples. Multiple testing corrections will present a high bar for many of the findings to be deemed “significant.” Even if findings pass this threshold, the observations might be difficult to replicate or generalize. It would be appreciated if the results could be benchmarked on existing knowledge about causal effects and consequences for the highlighted molecules. Nonetheless, the presented study offers a unique opportunity and starting point to develop N=1 statistics and apply these accordingly (see comment about the app).

While there is very little omics data available from astronauts, we have worked to include all publicly available data in this revision. Specifically, we have created a new Figure 2 that highlights comparisons to the NASA Twins Study and other known chemistry and biomarkers from 6- and 12-month missions from the Johnson Space Center Biochemistry lab (n=64 astronauts), and we have included Dr. Scott Smith now as a co-author, who provided these data.

Since this work is fundamentally building from the NASA Twin Study, for this revision we have included more benchmarking against this paper. This includes comparisons of the differentially expressed genes (DEGs) and cytokine changes from the Twins Study to those observed in the snRNA-seq data from I4 mission. This includes data in the new Figure 2 and several new Extended Data Tables.

As the reviewer notes, we required all reported results to pass significance based on corrections for multiple testing (using FDR-corrected p-values at 0.05 or 0.01), and also refer to other spaceflight studies whenever possible to contextualize our data. We have also updated the online data portal to include data from the NASA Twins study and JAXA cRNA (see below screenshots and link). When a user enters a gene, the data portal shows the JAXA data, I4 data, and the Twins Study Data results, which can then be compared to the data from this mission. These plotted data represent the vast majority of human omics that are possible to query, which we think will aid researchers in getting a context and broader benchmarking of changes in the i4 mission relative to other missions, for any gene, protein, or metabolite.

Reviewer Figure 1. Cross-study plotting functions of the SOMA browser. These screenshots show the ability to enter a given gene and visualize the expression of the Inspiration4 data relative to other human spaceflight data (tabs for each study on top, plotted for a user-selected gene, left). Taken from: <https://soma.weill.cornell.edu/>.

Physiologically, the authors see the main reasons for altered molecular profiles are microgravity and radiation (line 138). There, however, is no evidence that changes were not induced due to the lack of movement during space travel, stress, differences in air supply, and nutrition intake. Ideally, any space travel study should be designed with a second (or the same) group remaining on earth and experiencing similar conditions (e.g., in a simulator or airplane). This would reduce the number of differences and uncertainties and ultimately allow better reference to any changes. Similar questions

arise about the time of space exposure. Is such data available, or can data from other space travel studies be used to support some of the observations?

Microgravity and radiation are the two factors that will be a universal hazard to spaceflight. Factors such as stress, air supply, and nutritional intake are variables that will be much easier to control during space missions. This is the reasoning behind why radiation and microgravity were highlighted, as they will ultimately be the factors that may be easier to mitigate with medication and medical intervention. However, the reviewer is correct that, when analyzing these datasets, stress, air supply, lack of movement, and nutritional intake will also be factors influencing genomic profiles, and this is the main objective for using intra-personal controls and models that build on the pre-flight and post-flight collections. We've more thoroughly discussed and cited work that emphasizes these factors so that readers may understand the various influences that contribute to in-flight space omics profiles.

Molecularly, the authors should aim to build their statement of short, medium, and long-term consequences of space travel on the functional categories of the molecular data. It remains unclear which organs would be affected by space travel and why. Not all molecules will be equally "reactive" to the change in environment. Metabolites and lipids may be highly influenced by nutrition, and blood protein levels may be affected by liver and kidney function, and blood cell counts. In addition, a lack of exercise might alter muscle mass and protein turnover. These connections must be clarified.

At a broad level, the short/medium/long-term effects of spaceflight are summarized in Extended Data Figure 2, and we have added new data from cell-free RNA (cfRNA) assays that can measure the degree of stress and lysis across the entire body. In our revision, we have included cfRNA data from the I4 crew as well as the JAXA CFE study. We also submitted high quality versions of all figures so they are easier to review. In this figure, we break down the number of statistically significant features (i.e. differentially expressed genes, differentially abundant proteins, etc) from each of the datasets in each of the different flight, recovery, and longitudinal profiles. For specific differences in molecular profiles, we have a broader compendium of papers under review at various *Nature* publications that highlight these differences in greater depth than we could do in this paper.

Further, the observed brain-specific proteins could result from changes in the liquor during gravity, but no CSF samples were available to support such data. At last, molecules like IL-6 or IL-10 have been reported for various (inflammatory) conditions. Hence, space travel alone may not be causal but induce micro-inflammations. If so, can this be explained by RNAseq data of the blood cells? It might be helpful to focus on blood because many molecular measurements were obtained from this sample type (on earth). Could continuing micro-inflammations lead to changes (or even exhaust) in immune functions?

Thanks for the excellent question. We have generated new data to profile cell-free RNA in blood plasma from the astronauts, and also added deep RNA-seq (>350M reads/sample) to the Resource paper and web

site. We now include a figure showing the interesting results of an increase in hematopoietic stem cells (Figure 3e).

A complete sampling of the astronauts was only possible on earth, and it remains unclear why. A compendium paper (Overbey et al. Collection Protocols Paper [Nature Communications]) that describes the protocols was unavailable. Hence, assessing the impact of sampling on the molecular data was impossible. Still, some questions emerged. One relates to the skin microbiome: What is the effect of hygiene for space travel on these measurements? Were the samples collected from different areas or adjacent spots? Exposure to clothing or air, temperature, and light might influence this microbiome. The other question relates to the influence of the sample quality to separate the “sampling effect” from a “physiological effect.” Data on precision and variance would help to understand this.

A full paper detailing the collection protocols is now in press at *Nature Communications*, and we have also posted the pre-print of the paper on BioRxiv (doi: [10.1101/2023.05.02.539108](https://doi.org/10.1101/2023.05.02.539108)). Also, the revised manuscript has also been submitted as supplementary information to the reviewers with this submission. The details in Figure 8 of this paper should clarify details about skin sampling.

Reasons for the incomplete sampling during flight compared to Earth were due to cargo limits and lack of sample preservation options. The entire mission was performed in a Dragon capsule and space for scientific components of the mission was limited. The limited space available was shared with the other experiments in the mission. Furthermore, no cold stowage was available for use, so venipuncture tubes would have been subject to timepoint-specific degradation. Additionally, experiments on the ISS have centrifuges available to separate plasma, serum, and PBMCs before freezing, which was also not available on Dragon. The skin swab and capillary blood collected in flight were stable at room temperature due to the addition of DNA/RNA shield (swabs) and the dried blood spot card material (capillary blood). (More information in the collection protocols paper: doi: [10.1101/2023.05.02.539108](https://doi.org/10.1101/2023.05.02.539108))

Inter-individual differences between the astronauts must be assumed. Hence their molecular profiles at baseline are very likely to be unique. In fact, any data should be referenced to the clinical traits (e.g., BMI, ALT, CRP...) and considered variable when judging the longitudinal trajectories. For example, is there a co-variation of weight with any variables (note: Extended Data Figure 4 was not convenient to extract meaningful information from)? Which fraction of the data fluctuated between the three phases (pre, recover, long)? Instead of looking at changes in absolute values, the heterogeneity of variance in the molecular data can be another informative measure because it is less limited to expecting the same changes in all subjects at the exact sampling and allows revealing/shortlisting alterations that occur more generally. The mentioned analyses and connections should be computed and implemented into the app.

We have uploaded the data from our original Extended Data Figure 4 to the NASA GeneLab repository. The complete blood count is at OSD-569 (whole blood data repo) and the comprehensive metabolic panel is in OSD-575 (blood serum repo).

In addition, it would be more relevant to assume that each astronaut responds differently to the space trip. The authors should attempt to identify different response phenotypes. This could assist future studies in testing ways to improve the preparation and recovery of space travel.

We have updated the manuscript and included an entirely new Figure 5 to address this important question. We first focused our characterization of the high coefficient of variation (CV) for the various omics measurements, since these will be of great interest to personalized medicine uses for spaceflight medicine. We indeed found differences by each omics type, which are now included in the updated manuscript, and represent the priority list for countermeasure development and personalization.

The app is an excellent tool but should host all of the molecular data in the manuscript. Simple longitudinal plots will help the audience to understand alterations on an inter-individual level. It would also be helpful to enable the clustering of all the different data types using PCA and UMAP. Please allow users to select features of interest. Also, there is a need to generate qq-plots for each data type and implement a statistics and QC section into the app. It will be essential to judge the findings on accuracy and precision. The study has otherwise used established methods, and inviting more methods to be used in the future will improve the validity of the data.

We have now added another tool that enables user to create UMAP/PCA and other dimensionality reduction plots, for both the single-cell RNA-seq data and the ATAC-seq data, as well as volcano plots and measures of variance, which can be found here:

https://soma.weill.cornell.edu/apps/I4_Multiome/
(main landing page: <https://soma.weill.cornell.edu/>)

Moreover, we have added a section on the data portal that provides more details on QC metrics and enables a deeper analysis of the distributions of the data.

Also, the raw FASTQ files and the fastQC files are all available on GeneLab, to help users gauge the quality of specific samples they might want to analyze, and which can help with data re-analysis.

Finally, we have included two sections under each omics assay that details a sortable table of the p-values and corrected q-values, as well as the log fold change, and a plot of the range of values observed between the comparisons (e.g. pre-flight vs. post-flight), and a detail on the model used, when applicable (e.g. Lasso).

Instead of sharing data, is it possible to submit analysis codes to the project and present such on the app? A “call for codes” could be an attractive initiative to raise interest in the data as an N=4 challenge (see Kaggle).

This is an interesting idea. The democratization and accessibility of spaceflight omics data, specifically getting this data in the hands of as many people as possible to accelerate spaceflight discoveries, is our ultimate goal. The IRB designed for this study was designed with this goal in mind. Currently we are uploading data to GeneLab so that it can be tackled within the GeneLab working groups and integrated with analyses from other organisms. GeneLab has working groups in the following areas which analyze spaceflight omics data: Animal, Plant, Microbes, Multi-Omics, Ames Life Science Data Archive (ALSDA), Machine Learning/Artificial Intelligence.

Large-scale efforts often struggle with finding a core observation to present, tend to describe the data from many different angles consecutively, and choose to present observations rather than the biology behind these. This makes reviewing every detail of such papers difficult. The authors should reconsider the flow of the manuscript and find better connectors between the different data types. It is not one data type that will explain all the effects and consequences on human health, but it is the connections between them that will assist us in learning more about the system biology of space travel.

We agree with the reviewer on the importance of these findings . To help with this goal, we have provided both a descriptive overview of the datasets for the scientific community (who will want to analyze this raw and processed data), but we also demonstrate the biological narratives that we can uncover with these data. To improve on this, we have strengthened our narrative via comparisons with the NASA Twin Study, especially in Figure 2 and the new Extended Data Tables. Also, we have added a paragraph to the introduction section that points people to relevant analyses in the breakout papers which identify biological findings key to different datasets and find supporting evidence throughout the breadth of multi-omic data collected.

The authors have cited relevant literature but could add more recent multi-omic studies that have aimed to integrate longitudinal profiles in health and diseases. This will help to demonstrate the possibilities and limitations. The manuscript is otherwise clear and well-written.

We have added >6 more papers that reference related work in the field, as well as other multi-omics cohorts of patients (e.g. Zhou et al, 2019).

Any sample collection tube should be listed with the product and LOT number.

These are now all annotated with the collections and protocols paper, which can be found here: <https://pubmed.ncbi.nlm.nih.gov/37205403/>.

Which sample volumes are available, and which MTAs apply when accessing the material?

We have now launched a specific Biobank portal that enables users to request samples, at various volumes and amounts, and the MTA from Cornell to specific users will be agreed upon through that web site: (<https://cambank.weill.cornell.edu/>).

Sample list for biobank:

- CPT blood plasma
- cfDNA BCT blood plasma
- SST Serum
- CPT PBMCs
- PAXgene blood RNA tube - purified total RNA
- Vacutainer red blood cell pellets
- Crude urine (no preservative)
- Urine with zymo conditioning buffer
- DNA extracted from stool
- RNA extracted from stool
- Saliva DNA
- Saliva RNA
- Oral/Nasal/Skin swab DNA
- Oral/Nasal/Skin swab RNA

Referee #2 (Remarks to the Author):

With an increasing number of commercial spaceflights and a growing interest in long-duration spaceflight projects, understanding the molecular impact on astronauts is becoming highly relevant. In this study, Overby et al. describe a comprehensive resource of high throughput (omics) data covering the key events of space flight and extended follow up from the SpaceX Inspiration4 mission. The authors confirm previously described findings such as telomere elongation and elevation of cytokine levels, illustrating that commercial spaceflight missions provide a useful source for such data. The manuscript goes beyond existing work by providing several novel, high-dimensional data types, including spatial transcriptomics, RNA modifications, single cell RNA-Seq and ATAC-Seq, obtaining new insights in particular into the long term molecular impact of space flight.

One of the key novelties of the manuscript is the breadth of technologies used for profiling the samples. This design allows the authors to quantify the impact of space flight on different aspects such as the skin, microbiome, or gene expression over the different time points. Partially due to this breadth of data, the level of detail for each data type is limited, sometimes focusing on more descriptive findings regarding the global impact of space flight on each molecular data type. As described by the authors, additional data points from longer follow up or alternative space flight missions will be needed to confirm these observations and allow for more detailed analysis and interpretation. The manuscript and data resource are a foundation for collection of such data in the future, and as such data are likely to become more frequent, it provides a timely description, analysis, and resource to build on. The manuscript is well written and motivated, and (with a few exceptions outlined below) accessible to a broad audience. Due to the breadth of assays and the general interest in this topic, I expect this work to appeal to scientists from diverse disciplines.

Major comments:

1) Statistical tests, data description, and methodology are not always sufficiently clear

The manuscript presents a large data set covering different time points, different samples, and different data types. While the data overview and analysis overview is well presented and summarized, I found the details to be partially insufficient:

- The number of samples per data type and time point is not clearly described. Generally it seems that 4 astronauts were profiled for all data types and time points, and in many figures as well as in the supplementary tables the data and QC metrics from each sample are shown. However, when reading the text I was searching for this information from the beginning. I would suggest to add that description somewhere clearly in the text (and if possible add to Figure 1). I assume that all data was generated for all 4 astronauts so this is probably a trivial addition, but it is important to understand how the analysis was done.

We have clarified in figure 1 that all assays were performed on all four crew members and denoted any exceptions. Additionally, we have now included a supplemental table (Extended Data Table 1: Sample Information) that is easily searchable by sample type to see specifically which crew members are available. We hope that this will assist researchers in planning what analyzes they can perform from our datasets.

- For some analyses and statistical tests it is unclear how samples were combined or how the tests were performed. This was particularly difficult to assess for the single cell data, where the authors seem to pool all the samples (without describing sample numbers). However, this was also unclear for other analysis and statistical tests in the manuscript (for example: the BCR mutation analysis, how was this tested for significance? Is this based on the mean of the samples? Was the paired structure in the data considered for this test?; line 291 "GESECA padj..." how was this tested? How is the p-value related to each term that is mentioned?; Line 220/Extended Data Fig. 5c, The data suggests a very strong increase that should be statistically significant across all samples, but this was tested per astronaut, which then was not significant. I could not find out how the significance estimate was calculated per astronaut here (and why not across all astronauts);...).

We thank the reviewer for these suggestions, and we have updated the text and methods to make the testing and statistical assumptions more clear. Also, there's a more in depth single-cell paper for particular analyses beyond this paper (Kim J, Tierney B, et al., 2024).

- The authors summarise the overall analysis strategy in the main text for each analysis, which I very much appreciate and which I think helps to understand the data processing and analysis sufficiently for most readers. However, the methods section and supplementary text do not provide much more information beyond this general overview, they are often not sufficient to reproduce the results. Furthermore, the method section combines experimental methods with computational methods

description, which makes it very difficult to find the relevant details. I would recommend to separate experimental and computational methods using different sub-headers. The parameters, software version, and any thresholds or filter that were applied should also be described. Ideally, in addition to those changes, code is made available by the authors that could generate results and figures from the publicly available, processed data, but given the breadth of data sets and analyses I understand that this might not be feasible.

The methods section has been reorganized, separating wetlab and drylab methodology. Also, a table of contents has also been added with section headers for different components. Furthermore, the parameters, software version, and any thresholds for the code that generated the results for the paper (e.g. differentially expressed genes, differentially abundant proteins) has been uploaded to a centralized Github repository: <https://github.com/eliah-o/inspiration4-omics>

2) Analysis-specific comments

2.1) single cell analysis

- This analysis was the only analysis where the individual samples were not clearly shown or described. For single cell data, a higher level of variation might be expected across astronauts, but it would still be helpful to investigate this. Are the observations and results described by the authors consistent across all astronauts?

We agree with the need to visualize data on an individual/crew member basis. We've included data viewing options in the SOMA browser to view individual crew members VST normalized gene expression at the single cell resolution: https://soma.weill.cornell.edu/apps/SOMA_Browser/
Lefthand option: "Human Gene Expression"
Tab: I4 Multiome RNA

If readers are interested in specific DEGs, they can cross reference from our processed datasets with the SOMA browser to gauge consistency between individuals.

- The differential expression analysis using the single cell data was not clearly described. Was the paired structure of samples (same astronaut at different time point) considered when analysing differential gene expression? Was this done by pooling all samples from the same time point? Did the authors consider the astronaut as a covariate in the differential expression analysis?

Whenever possible, we used paired tests, but some methods required other techniques. For example, the single-cell multi-omic analysis was done by pooling all the cells from 4 crew members into a single group by timepoint. Each group has a different number of cells per group and a different number of cells per cell type, and thus we used non-parametric tests whenever possible, so as not to assume normality. For the new figures we've added (ex: Figure 4), we've included specific references to the flight (FP), recovery (RP), and longitudinal profiles (LP) described in figure 1 to make the specific timepoint comparisons clear.

The authors say that the number of DEGs returns to baseline levels. However, it is not clear what baseline levels are, and the figure that is referenced does not show any baseline differential expression numbers either. What are the genes that change expression? Is there any enrichment of specific categories?

We have fully updated this section and created an updated Figure 4 to make this more clear. Based on our subject design, the baseline is the pre-flight timepoint and the number of DEGs of the pre-flight vs pre-flight is 0. We have also included the pathways that are altered in the main and supplemental figures.

- Regarding the illustration of differentially expressed genes (Figure 2d): (a) the authors provide names for the comparison, but as these names are not intuitive one has to jump between figure 1 and figure 2 to understand this analysis. Maybe a more descriptive title could help, or an additional illustration in Figure 2d. The data does not show the expected differences, which made it particularly difficult to understand (see point below).

We have updated and created a new version of Figure 4, as well as the tables from the paper, to make this more clear.

- Regarding the differential expression analysis (Figure 2d): R1 and R82 seem more similar than R45 in the longitudinal profile. The longitudinal heatmap looks very similar to the flight profile heatmap, where the R+1 and R+82 are also more similar. Why is that the case? It's not clear whether this reflects properties of the data related to the samples, errors in the analysis, or possibly QC issues, and if that impacts the claims that can be made from these data. Showing consistency across samples could be one way to validate these results.

We have now included calculations of variation across all time points for crew members, as well as included other control data in the supplement, but there is no obvious reason that R+45 is sometimes an outlier for specific crew members. The same training, tube types, tube batch, and protocols were applied to all timepoints, but we have added a discussion of the need for larger comparison cohorts into the discussion.

- In the TFBS enrichment analysis, many transcription factors from the same family are highlighted. Many motifs from such transcription factors might be very similar, and the precise transcription factor might not be identifiable from the sequence. I would suggest to show these individual motifs in the supplement, and if they are very similar, they could be combined into motif groups rather than listing all transcription factor names (eg the CEBPX family might be very similar in terms of motif). Some QC plots or analysis for the snATAC-seq data would be helpful. Does the data cluster by cells as expected? Is there an agreement between expression and chromatin accessibility? Do the samples from different astronauts show similar results?

We have added extensive details to the Supplemental Information file for the paper for sample and data processing steps. The TFBS enrichment section has been removed from the paper and integrated into a different paper from the I4/JAXA package, since it had related immunological data.

2.2) gene expression analysis

- The details in the RNA modification analysis are not sufficient to fully reproduce the results (similar to other sections, see point above). For example, which sites are selected to be considered in the comparative analysis? What about sites which were not detected in a samples due to lower read numbers, how was that considered in this analysis?

We relied on established methodologies and metrics to perform RNA modification analysis. m6anet, which was recently found to be the most robust m6a caller (10.1038/s41467-023-37596-5), requires at least twenty reads per site in order to generate a m6a methylation probability. For our comparative analysis, we excluded sites that lacked probabilities for all samples. Within our comparisons, we generally observed that we could only confidently call differential modification at sites which were covered by the majority of samples. We have edited both our methodology and figure legend to better reflect these considerations, and added a lot more details to the Supplemental Information file for the paper.

Direct RNA-seq

Oxford Nanopore Technologies' direct RNA library prep protocol for PromethION was used with 500ng of total RNA. One PromethION flow cell (FLO-PRO002) was used per library (i.e per sample), and libraries were sequenced on PromethION 48 beta for 72 hours using the MinKNOW software. Fast5 files were basecalled with Guppy version 6.2.1. The Oxford Nanopore Technologies pipeline, pipeline-transcriptome-de6, was used to quantify transcripts and to perform differential expression analysis. Differentially expressed genes were also calculated using minimap27, featureCounts8, DESeq29, and the SARTools10 wrapper package. Quality assessment was performed with pycoQC4, samtools [excluding multi-mapping reads], and MultiQC5. m6Anet11, which requires a minimum coverage of 20 reads to generate predictions, was used to detect m6A modifications. methylKit was used to identify differentially methylated sites, excluding samples within site comparisons lacking predictions.

The updated figure caption is as follows:

Direct RNA-seq Gene Expression and RNA m6A Modifications Across 13 Comparative Profiles. (a) The data flow in the analysis of direct-RNA data. (b) Patterns of gene expression across seven time points and 13 comparisons. Left: z-scored log-transformed normalized gene counts obtained from salmon (bottom left of each cell) and featureCounts (top right of each cell). Right: $\log_2(\text{FoldChange})$ values obtained from edgeR (bottom left of each cell) and from DESeq2 (top right of each cell). The genes are clustered by z-scored log-transformed normalized counts using the correlation distance metric. (c) Patterns of base-level m6A modifications across seven time points and 13 comparisons. Left: z-scored log-transformed positional methylation probabilities obtained from m6anet. NA denotes sites lacking necessary coverage to obtain a methylation probability for the

charted sample. Right: percentage of change in methylation between the conditions in each comparison, obtained from methylKit. The sites are clustered by the pattern of differential methylation a cross all comparisons using the correlation distance metric. On both panes (b) and (c), only the genes and sites with significant differences in expression ($p\text{-adj.} < .05$, *edgeR* or *DESeq2*) and/or methylation ($q < .01$, *methylKit*) in at least one comparison are plotted; the significance of individual comparisons is annotated with up and down arrows.

- The supplementary table shows a substantial variation in read length, with 3 out of 4 R+1 samples having much longer reads. Could this impact the analysis?

While the base-called lengths exhibit this variation, the alignment lengths for the R+1 samples are in line with the remaining samples. We provide this data below, generated with the *samtools stats* command, and have reported it publicly Extended Data Table 2: Direct RNA-seq Stats. We have updated the methods to reflect this addition.

Time Point	Crew ID	Read Length - Aligned (bp)
L-92	C001	582
L-92	C002	560
L-92	C003	575
L-92	C004	590
L-44	C001	620
L-44	C002	601
L-44	C003	609
L-44	C004	607
L-3	C001	552
L-3	C002	520
L-3	C003	632
L-3	C004	617
R+1	C001	632
R+1	C002	602
R+1	C003	614
R+1	C004	671
R+45	C001	612
R+45	C002	562

R+45	C003	312
R+45	C004	477
R+82	C001	555
R+82	C002	572
R+82	C003	556
R+82	C004	563
R+194	C001	613
R+194	C002	603
R+194	C003	599
R+194	C004	617

- The authors perform a gene set enrichment analysis for gene expression, but there is only limited information provided. How was this analysis done? The full results could be shown as a supplementary figure. Is there any enrichment of genes that show differences in modifications?

We have devoted an entire companion manuscript, entitled *Direct RNA sequencing of astronaut blood reveals spaceflight-associated m6A increases and hematopoietic transcriptional responses* [in review: Nature Communications], to exploring the biological implications through m6a-DGE integrative GESECA analysis, allowing for a richer presentation. We certainly address the reviewer comments above comprehensively with this manuscript. Ultimately, the focus within the resource paper is on demonstrating the creation of transcriptomic gene sets associated with astronaut spaceflight. Therefore, we have removed the following line from the manuscript: “The largest set of changes occurred at R+1 and included genes enriched for MYC targets, oxidative phosphorylation, and interferon response (GESECA, $\text{padj} < 2.3 \times 10^{-37}$).”

2.3) spatial transcriptomics

- The different regions are introduced and Fig 5b is referenced, but the relation between Figure 5b and these regions is not clear. Maybe additional labels in Figure 5b might help, or at the minimum an explanation in the legend. For someone who is not familiar with skin it is unclear how these figures show specific regions.

- The association of spatial transcriptomics with microbiome changes was difficult to follow. First the authors describe the spatial transcriptomics data, then the microbiome association is described, and then the authors go back to describing the results from the spatial transcriptomics data. The authors describe the analysis for each data type in detail, however, the microbiome data was never described when it is introduced at this point. The analysis itself is a good example of how the different data sets could be combined (see comment below about data integration/comparison). However I found this part difficult to follow, this could be restructured for more clarity

- I could not match the text (lines 423-429) to the figure (the names do not match). This part either needs clearer labels or additional information.

Thank you for the careful reading. The spatial transcriptome section has been revised and updated, and more details are now also in the Supplemental Info file and the figures.

2.4) Microbiome analysis

- The microbiome section in the manuscript describes the data processing and analysis in details, but the findings and the interpretation of these findings are not very clear (partially due to missing figure references). For example, the authors describe their approach to build a non-redundant protein catalog, but it's not clear how this is used, or what the conclusions are. The paragraph that follows describes some findings on variation at the species level as a function of flight (referencing figure 6b and Figure 6). It's not clear how these statements are shown on these figures. It's possible that the authors wanted to refer to Figures 6e and f, which show the global overview. However these figures do not quantify any observed variation in relation to sample time/flight.

- The authors quantify changes in microbiome using linear mixed models in the next paragraph. Which organisms contribute to these changes? The authors write that this indicates a rapid microbiome transfer, can this be shown with the data? This appears to be a consequence of contact between astronauts, not a consequence of space flight.

We have updated the CV methods and microbiome protocols used for the paper, and also added extensive details to the supplemental materials section. We have also updated the plotting functions on the SOMA data browser, enabling two means by which to map and plot the data, which also link to the species annotated in the Extended Data Tables.

3) Data Quality

The authors have performed detailed Quality Control (QC) and made QC metrics for many data sets available, which is very helpful for a data resource that includes such a diverse set of data types. However, the supplementary tables currently only provide this information for some of the experiments. The QC metrics and criteria for sample exclusion (if any) could be described in more detail in the methods section for each data type, and they could be made available for each sample in a supplementary table, similar to the existing supplementary tables.

We have now created a dedicated supplemental methods section, and also submitted a focused protocols paper that will be published in parallel with this paper in *Nature Communications*. See here for more details:

<https://pubmed.ncbi.nlm.nih.gov/37205403/>

4) Multi-Omics integration/comparison

- The data resource describes many different data types, but the authors largely analysed each data type separately (often arriving at similar conclusions related to the impact on space flight on molecular signatures). A comparison of these results across data sets could help to provide a global overview, and possibly strengthen support for some claims on the long term impact on molecular

signatures. Figure 1b shows that many comparative analyses are available for many different data sets, do these show the same patterns?

We now highlight the places where chromatin and gene expression are both changed, which hadn't been integrated before, and we note that direct RNA-seq data enabled us to detect both gene expression and m6A level changes, which we also linked to cytokine data in the text. We also show the variation across time for each other assays, and show where they differ in terms of inter-individual variation (Figures 4, 5).

- The authors describe that some data types show an impact of space flight which returns quickly to pre-flight levels, whereas other data types show more long term impact. These statements could be quantified for each data type, and presented in a global overview that assesses the impact on molecular changes of space flight over time across the different data modalities.

We have updated the tables and online resources in NASA's GeneLab repository to highlight the specific changes for each modality of biology that was assayed, which can help researchers examine the changes specific to spaceflight, pre-flight, and afterward. Namely, each dataset has an in-depth metadata sheet as part of the Open Science Data Repository (e.g. OSD-569), and we have updated the Figures to highlight differences.

5) online data visualization

The online data visualization tool is not sufficiently documented, or possibly not fully functional: Following the links in the manuscript I land at a page that says "Hello" (wrong link) or "I4_Multiome" (correct link). The portal then allows me to look at PBMC data, but not all data sets as described in the text (line 197-199). For example, direct RNA-Seq, spatial transcriptomics, or microbiome data is not included it seems. Furthermore, the different tabs (eg gene expression, TFBS, ...) led to the same figures based on gene expression with exactly the same description. Either the other analyses are not properly linked, or they are not properly described.

We have now fully updated the data portal and included the other data types mentioned in the text, and also added more tools for the comparison to other spaceflight data (e.g. NASA Twins Study, JAVXA CFE mission).

Minor comments:

- figure 2c: The title says "DAP counts", should this be DAR counts?
- figure 4h is referenced as 4g in the text, Figure 4g seems to be referenced as 4c
- line 423 the figure reference seems a bit misplaced here (Fig 1b).
- "Skin-microbiota interaction changes were also observed resulting from spaceflight exposure" I am not sure if the authors can claim that this results from space flight based on the data, I would recommend to reformulate this, eg "were observed after spaceflight exposure"
- figure 6, the y axis labels are missing
- figure 6 e and f are not referenced in the text
- citations are missing in discussion when previous findings are referenced (first paragraph)

- the authors could explore differential isoform usage, splicing, or polyA tail length with the long read direct RNA-Seq data. Isoforms are even mentioned in the introduction.

We appreciated the detailed review and have now fully updated our figures and also added more references. For the RNA biology questions, we are performing additional sequencing with an orthogonal method (cDNA sequencing with Ultima), and this work is now the basis for an in-depth paper on the transcriptome.

Referee #3 (Remarks to the Author):

The manuscript “The Space Omics and Medical Atlas (SOMA): A comprehensive data resource and biobank for astronauts” provides a dataset with a large number of clinical features and multi-omics of 4 astronauts before and after a 3-day spaceflight. This dataset also covers time series spanning 92 days before to 82 days after spaceflight, which provides a strong foundation for deep analysis of short- and long-term effects of spaceflight. The authors use an array of almost all known omics technology to almost maximize the amount of information that could be gained from this study.

This manuscript functions as an inventory of available data and follow up studies will also increase its impact.

We appreciate the positive comments about our manuscript and thank the reviewer.

Some additional analysis could improve the manuscript.

(1) In the section “Proteome and Metabolome of Blood Plasma”, the authors provide heatmaps of plasma proteomics, metabolomics and Extracellular Vesicle and Particle (EVP) proteomics in Figure 4. The pattern of the heatmaps give the appearance of some structure, however there is no further discussion in the main text. The authors provide pathway enrichment information of the most variable metabolites in Extended Data Figure 6, but these results are not connected to the heatmap. The authors could add pathway information alongside the heatmaps to more clearly illustrate how those pathways change during spaceflight.

Similarly, for the EVP proteomics, the authors provide liver and brain enrichment information, but this information is not connected to the information in heatmap. Furthermore, there lacks pathway enrichment in plasma proteome analysis. Similar analysis could be considered.

We have updated this section in terms of the methods and QC. Given the reviewers’ and editors comments, we have also now expanded the EVP results into a specific, distinct manuscript, and kept the data anchor in this paper, to give more detail on the data was generated, processed, and used for inter-individual comparisons of the variation in response to spaceflight, especially for the new Figure 5.

2) Ease of accessibility of the data is the key for the impact of this study as it is expected these data will be used in follow up studies. The authors should state clearly in the manuscript how they intend to make the data available (preferably an easy to use website)

The data sections and tables have been updated with the latest GeneLab OSDR links, and all the data will be fully available upon publication. We have now updated the data portal to include easy-to-use plotting and data visualization tools, including volcano plots, DEG lists, and statistical significance for various comparisons between missions and time points, as well as compared to other studies (e.g. NASA Twins Study, JAXA). Please see our updated data browser here: https://soma.weill.cornell.edu/apps/SOMA_Browser/

Minor:

(1) Some figures in the manuscript have a low resolution, making them difficult to interpret, (e.g., Fig 2D, 2E, 6B, 6E, 6F).

High resolution images have been submitted with our revised version of the manuscript and all figures have been updated.

Referee #4 (Remarks to the Author):

This reviewer has been impressed when reading this manuscript of Dr. Overbey and colleagues on „The Space Omics and Medical Atlas (SOMA): A comprehensive data resource and biobank for astronauts“.

This valuable approach of a high resolution investigation taken up by this consortium is unprecedented and will allow also many follow up experiments and analyses.

We are very grateful for the positive comments about our manuscript and thank the reviewer.

This very holistic investigation on molecular and cellular changes as a function of spaceflight as performed in a crew of four (2 female, 2 male) will direct new research approaches and will deliver added values well beyond the initial investigation as presented. The multi-omics- approach linked to some clinical profiles pre-flight, in-flight, recovery, and longitudinal responses is in its limitations well designed. The i4 mission crew was in space for about three days and some signatures of a longer duration spaceflight mission could be seen here as well. The follow up was as interesting, since many effects went back to normal while other features (gene expressions, also proteins) seem to persist longer post-mission. This multiomics approach and application of the „full battery“ of modern analytical tools aimed to provide a new and innovative biological analysis approach especially also when compared in the future also with data from other conditions.

We thank the reviewer for the appreciation of the work and utility of these data for upcoming missions and spaceflight studies.

This reviewer was delighted to read, that all datasets can be mined through an interactive data portal, maintaining the spirit of research and especially space research to have not limits. Also in the light of patient care, drug effect studies and as for the success of upcoming lunar and exploration-class missions to Mars these studies can be helpful.

We agree with the reviewer in the utility of the data portal, and we think the reviewer will find the updated portal even more compelling. We have added new data from the JAXA cell-free RNA study, new cfRNA data from the I4 crew to enable comparisons to the JAXA study, and also created a browsable volcano plot function that enables comparisons between timepoints, crews, and assays. Finally, we re-consented the Kelly Twins from the NASA Twins Study to enable us to share their data (expression, proteomics, and metabolomics) under our new IRB protocol, and on the same portal.

Detailed comments:

While the manuscript has – to the best of this reviewer (though in some areas limited knowledge of this reviewer, as indicated to the editor) no flaws which should prohibit its publication per se, some major critique has to be raised since the significance as by the conclusions drawn, however, have induced some major concerns that are explained here.

The abstract is well written as is the text and supplementary material. It is a huge piece of data sets that are included and explained and therefore suffers from more integrative views on the data especially including a more critical view and discussion of it. This reviewer well acknowledges the challenges to extract all the information and to well balance their impact in one document. However, some key elements of a scientific evaluation however went therefore underrepresented.

1.) It lacks information about the subjects investigated and the referenced paper on the methods is not accessible. They shall be included here irrespectively of other publications.

We have updated the paper to clarify the crew and mission details, and we have also now completed and submitted an in-depth protocols and methods paper, which is in press at *Nature Communications* and also now available on bioRxiv:

<https://www.biorxiv.org/content/10.1101/2023.05.02.539108v1>

2.) it is still a very small cohort and of two biological sexes, mean values are shown and sex differences not addressed.

This is a very good question, and give the space limitations, we have now created an in-depth analysis of sex differences in a parallel paper about the immune system changes (Kim et al) and we have shared this with the editors.

3.) barely significant data are highlighted (as for telemore) and very little discussion and efforts are made to explain this moderat though surprising effect of a 3 days flight. This interesting topic shall

be specifically addressed though also its likeliness to be affected so early, as well if that differs on the physical state / regular exercise, biological sex and so forth. Especially since exercise can reduce oxidative stress and preservation of DNA and telomeres as know for a bit (e.g. see Song et al Aging Cell. 2010; 9:607–615). The individual data points cannot be discriminated in Ext.Fig 5C.

We have added a citation to the paper above, which is very helpful for context. For Figure 5C, the individual points are available, but did not appear in the resolution of the submitted manuscript. We are submitting high-resolution images this time so these points should be clear.

4.) the proteo-metabolomics are so dependent on the individual conditions across several independent studies, key proteins and metabolites which are involved can be regulated by identified food compounds affecting key metabolites <https://doi.org/10.1159/000381349>

This is a key consideration, and we agree with the reviewer. We have added a reference to the above paper, and mentioned the need for contextualization for these kinds of normal biological changes in the introduction and the discussion.

5.) The more and more established concept of targeting individual needs and health risk and therapies as termed “personalized medicine“ or “precision medicine”, respectively, can rely also on global and such integrative approaches for a single subject / patient / crew member. Especially this short flight can be an interesting model for personalized medicine because the deregulated pathways underlying these conditions. It is recommended to include this in the discussion. This is one of the major values of this study and the follow-up and shall deserve more space (now only mentioned once) p7/line126.

We agree with the reviewer on the importance of this work to help guide future precision medicine in spaceflight, and we have updated the manuscript to discuss more of the utility for future missions and also now included a dedicated Figure 5 about the inter-individual variation. We have also discussed the new opportunities for precision medicine and individualized risk assessments.

6.) also that the exposure to space for such a short time can barely be distinguished from an semi-acute stress model and here respective controls are warranted to make this distinction.

We have now included additional controls from varying-length missions in the data portal, which helps to validate hypotheses and provide context for the consistent features from spaceflight.

7.) The habits and way of life of this non-professional crew before and after mission, their experiences with more stressful conditions of life, their motivations and other “soft” factors shall at least be discussed in the context of data interpretations and conclusion. The stress hormonal impact can be huge and may be differential also on the preparatory phase as well as in the flight.

We agree with the reviewer, and we have now updated the text to highlight these inter-individual differences. For example, L-3 looks different from the L-44, and L-44 is also distinct at certain points due to training. Moreover, since Jared Isaacman is flying again over multiple missions, we will have an unprecedented ability to address the various phases and the impact on the biomedical measures.

8.) Statistics with an $n=4$ are limited in their value given the high interindividual differences. In the light of the importance of this report to foster the tools to be applied in personalized medicine (and crew care), the individual and sex differences over time and parameter should be always displayed and discussed, given the approval to hereby identify the individual by a chance of 50%.

The methods we deployed for the paper have been well-characterized and published, and notably most methods use a more conservative correction for multiple testing (e.g. Bonferroni) than permutation testing. However, to further help with validation, we utilized multiple analytical methods whenever possible and the most conservative corrections whenever possible. For example, in the microbiome analysis, we have built a new browser and data portal that shows the results across seven different algorithmic approaches (Xtree, MetaPhlan4/StrainPhlan4, Phanta, Kraken2) and five different databases (GTDB, GenBank Viruses, Refseq, Phanta Default, and MetaPhlan4) to classify reads into different taxonomic categories, and we also did a de novo assembly analysis to identify the abundance of non-redundant genes/functions. The updated data portal can be seen here:

https://soma.weill.cornell.edu/apps/I4_Microbiome/

9.) Control groups are announced but not further explained and a limitation section shall be included. We have added more specifics about the control cohorts and the Biobank, and also highlighted the limitations of the study in the discussion, as well as added data from the NASA Twins Study and the JAXA data.

Minor:

Figure resolution and readability is very limited (likely a matter of .pdf conversion)

We have now uploaded high-resolution figures for the revision. Thank you.

Referee #5 (Remarks to the Author):

Overbey et al. profile multiple data sources from astronauts on the Inspiration4 mission, including data collected longitudinally pre-, mid-, and post-flight. From biomarker panels, the authors find an increase in inflammatory cytokines, in agreement with studies on longer duration flights. The authors also use single-cell expression and accessibility measurements to show long-duration changes in the epigenomes of specifically monocytes, including an increase in accessibility at sites containing binding sites for general stress and immune response TFs. Proteomics shows a dysregulation of some brain-related proteins, while spatial transcriptomics showed signatures of inflammation and KRAS signaling in the skin. Microbiome analyses showed transfer between astronauts in the exposed areas, with relatively minor domain-level changes. Most of the observed changes stabilized to baseline after re-acclimation to earth. The authors make these data available via a biobank/web portal, and are accumulating more data from future missions.

The authors have collected a large amount of diverse data on these astronauts, and in principle, multiomic analysis may have the potential to both understand the systems-level changes to humans as they are exposed to new environments and to develop plans to mitigate the risks for future

missions. However, the lack of a truly integrative analysis of these data, the relatively limited amount of time of exposure, the relative lack of statistical power that would allow for concrete conclusions substantially limits our enthusiasm for this work.

Major Critiques

Lack of significant findings and integrated analyses - The authors have collected a large amount of data, but these measurements are not really compared or integrated in a meaningful way (i.e. analysis of each data type is on an island of its own). The work would be substantially improved if these data types could “talk to one another” and potentially support the conclusions and findings that are made with different data types separately. Ideally, some connection to the molecular changes observed might be discussed in the context of the physiological changes that are already known as side effects of space travel. As the manuscript stands, there are many individual datasets presented with very little explanation for why those -omes were collected, or how the different -ome datasets relate to one another, even at a very basic level (e.g. do changes in regulatory sequence accessibility in monocytes correspond to changes in regulated gene RNA abundance?).

We focused on these specific assays for the mission in order to replicate and expand upon the NASA Twins Study, but also to add newer assays and data types that emerged since the original 2014-6 study, and we have now highlighted the novel assays and findings from this study, as well as those which recapitulated the prior work.

Also, we have significantly updated the SOMA Data portal (see Reviewer Figure 1), to enable a single location for all the data sets (pre-flight, mid-flight, post-flight) from this mission, which enables an easy means to plot comparisons (line plots, volcano plots, violin plots) between a given gene, chromatin state, protein, metabolite, exosome marker, and microbiome association, and then be able to plot these changes relative to other human spaceflight data. To our knowledge, the comparisons in our paper that merge epigenetic and transcriptional data, and also point to novel microbiome data, are the first-ever combination of such data sets, and the first-ever portal with such integrated astronaut biomedical and omics data. Moreover, with this data resource and linked samples becoming public, additional value is gained by the broader research community.

Furthermore, there appear to be very few major changes in any data type and no (or very few) significant findings reported. The many -omes are not synthesized together to create a comprehensive picture of what happens to molecular human physiology in space flight, which is, in the end, what an astronaut -ome might be expected to provide.

We have now updated the manuscript to focus on the distinct and overlapping pathways and biological functions across the omics results. Specifically, we have created a new figure with an overrepresentation analysis (using the exact same processed data files that are uploaded to GeneLab), which now shows the overlapping enrichments across the gene expression landscape and specific proteomic changes.

Limited experimental conditions - To be able to make claims about how human physiology changes in space flight, one might be especially interested in molecular changes that occur on timescales longer than 3 days. Furthermore, it is unclear what changes are occurring because of being in confined quarters and eating the same food, vs. being in a microgravity etc. These limitations to the data corpus substantially limit its utility.

We have now updated the manuscript to include comparisons to the NASA Twins Study and other biomarkers from 6- and 12-month missions from the Johnson Space Center Biochemistry lab (n=64 astronauts), which give context and more utility to the data, enabled a comparison of the short vs. long duration missions.

Lack of motivation and functional interpretation of results - It is not clear why the authors collected the data they did, or what the few changes they observe in the data mean. Additionally, the analysis of these data is relatively superficial, and does not integrate independent external data sets that might provide a more concrete biological interpretation. For example, the authors claim that there are some differential accessibility peaks in monocytes and that TF motifs are enriched in the motifs. Are the changes similar to other documented monocyte activation? Do the motif enrichments make sense with monocyte physiology or correspond to changes in monocyte function? Is it happening in all monocytes? Is this the expected immune response to people living in close quarters?

These are the only samples available from this mission, but we collected as many timepoints and assays as we could. We used the pre-flight baseline data for all crew members as the best controls, we have also included additional data in the revision to validate some of observed changes. Moreover, as described above, the JAXA and Twins Data are now integrated into the web portal, which we have now highlighted in the revision.

Similarly, analysis of m6A abundance is likewise quite superficial. Statements such as ‘some transcripts change in m6a abundance’ do not provide any biological insight. What genes? Are these genes related, or are there commonalities in the m6A targets? Does this correspond to an increase or decrease in protein levels? Similar questions could be asked for all datasets, and if support for observations in one data set is not supported by other data sets that might be expected to provide support, this casts doubt on the reproducibility of the findings in individual molecular modalities. We have updated the results section to make it more clear and to highlight the impactful changes that propagate across the different omics, and there is now an in-depth analysis of the transcriptome in a companion paper (Grigorev et al).

Also, we have now included a comparison of the protein levels and the transcripts in another companion paper (Houerbi and Kim et al).

For the methylation methods, the m6aNet tool was recently benchmarked and found to be the most accurate m6A caller, and thus we believe these are the best results possible with this current nanopore data and base caller. Specifically, we have added this to the references:

<https://www.nature.com/articles/s41592-022-01666-1>

“Detection of m6A from direct RNA sequencing using a multiple instance learning framework.”

Moreover, we found that the m6A changes are still well beyond the normal reference range found in whole blood direct RNA-sequencing (Reviewer Table 1), where the standard deviation of methylation changes is normally quite low, in the range of 0.05-0.06, and the methylation changes observed for the significant differences were at least 0.21. As such, any changes observed in the crew were generally >3 standard deviations from any background measurement noise, and also were modeled, measured, and corrected for multiple testing with DESeq2, to further ensure their statistical rigor.

Reviewer Table 1 - Standard deviation of RNA methylation changes from the direct RNA-seq data.

L-92	L-44	L-3	R+1	R+45	R+82	R+194
0.0601	0.0590	0.0591	0.0567	0.0582	0.0551	0.0600

Statistical analysis. Given the relatively limited number of individuals profiled (in contrast to the immense diversity of the data types), it seems that permutation testing would be a wise way to ensure that the differential elements that are reported are truly significant. We suggest permuting labels for all tests and building an empirical null distribution from these permuted data sets to provide a more stringent assessment of the number of significantly differential measurements actually observed.

We note that the analytical methods deployed for the paper have been well-characterized and published, and notably most methods use a more conservative correction for multiple testing (e.g. Bonferroni). Moreover, for the microbiome results, we have used seven different algorithmic approaches (Xtree, MetaPhlan4/StrainPhlan4, Phanta, Kraken2) and five different databases (GTDB, GenBank Viruses, Refseq, Phanta Default, and MetaPhlan4) to classify reads into different taxonomic categories, and we also did a *de novo* assembly analysis to identify the abundance of non-redundant genes/functions, all of which confirmed our primary findings.

m6A data: The authors present main text data in which individual samples show extremely strong differentially methylation signals (Fig 3C). Are these large scale changes for single individuals believable? Are there controls that convince the authors these observations are not technical artifacts? Is this a new finding in biology, or are these large % changes not significant? It is highly challenging to wade through all these data sets to obtain some sort of biological narrative the way it is currently presented.

The application of high-throughput sequencing to produce m6a epi-transcriptomic maps is quite novel. It is an open question whether gene sets based on differential expression are comparable with gene sets based on differential RNA methylation. Our statements were meant to highlight the applicability of our methods to generating phenotype-associated methylation sets, in our case space-flight associated m6a methylation.

Comparing the 193 gene loci which contained m6a motifs with differentially methylated sites (q-value < 0.01, *methylKit*) with the set of 440 differentially expressed gene loci (FDR-adjusted p-value < 0.05, either *salmon* or *SARTools*), we find a modest overlap of 53 genes. We have devoted an entire companion manuscript, entitled *Direct RNA sequencing of astronaut blood reveals spaceflight-associated m6A increases and hematopoietic transcriptional responses* [in review: *Nature Communications*], to exploring the biological implications through m6a-DGE integrative GESECA analysis, allowing for a richer presentation.

We agree that convergent observations would strengthen the reproducibility of our results. With regard to further integrative analysis with the datasets presented within this manuscript, we examined RNA within whole blood, whereas the proteomic profile sampled plasma. With this caveat, we have now performed a brief m6a-proteomic gene set comparison based on the RP1 time-point comparison, for which processed gene profiles were publicly deposited in the NASA GeneLab for both data types. Of the 116 gene loci which contained differentially-methylated m6a motifs (q-value < 0.01, *methylKit*), there were a total of five gene loci which overlapped with the differentially-translated plasma proteomics set, consisting of 59 proteins (q-value < .05). Of these, two genes, *TGFBI* and *PRDX6*, demonstrated sites with hypermethylation, whereas one gene, *RPL27*, demonstrated decreased methylation. Two genes, *SRGN* and *MNDA*, contained multiple sites, some of which exhibited increased methylation, whereas others demonstrated decreased methylation. The role of m6a in mRNA decay, which has been demonstrated but not fully mechanistically explained, may play a role in these observed differences within the proteome.

Statistical significance of the telomere lengthening results - The authors emphasize in the abstract and discussion that they observe lengthening of the telomeres of the astronauts, in agreement with studies of astronauts exposed to longer duration space flight. However, if these results do not pass statistical significance in 3 of the 4 astronauts, and the data are not presented in any main figure, the authors should remove the centrality of these claims.

We have updated the Figures to now include the updated results, and to highlight the differences between trends and statistical differences.

Presentation of figures - Many of the figures are hard to read and low resolution, making detailed examination of the data presented difficult, if not impossible. The authors should embed high-resolution panels (as well as fix some of the axis labels and styling that hamper interpretation, e.g. 2b/e).

We have now uploaded high-resolution figures for the revision. Thank you.

Referee #6 (Remarks to the Author):

Overbey and colleagues present a dataset of 2,686 multi-omic samples collected from four crew members during the short low-orbital mission Inspiration4. Together, these samples span numerous omics strategies (e.g., genomics, RNA-seq, snATAC-seq, cfDNA-seq, epitranscriptomics, proteomics, metabolomics, metagenomics, metatranscriptomics) and contribute to the newly established Human Space Omics and Medical Atlas (SOMA), the largest molecular dataset to study the impact of spaceflight on the human body. After describing the dataset, the authors analyze its different omics dimensions and first recapitulate previously observed consequences of long-term spaceflight. This comparison to previous results, although not providing original conclusions, is welcomed as it validates the newly established atlas and demonstrates the relevance of studying short duration spaceflight (three days). In addition, the authors analyzed of the Inspiration4 to reveal new features they attribute to spaceflight, including increase in circulating brain-derived proteins, rapid microbiome transfer and KRAS signaling in the skin, as well as identified changes (e.g. in histone H2A levels) that persisted after long after the short spaceflight and thus warrant continued monitoring.

The dataset presented by the authors (n=4) covers many more omics dimensions compared to currently available resources (e.g., n=11 for telomere length and n=18 for blood plasma protein expression) and increases by four-fold the previous multi-omics study of one astronaut (the NASA twins study). Overall, there is a lot to like about the dataset itself, its originality and the impressive number of dimensions studied, which will likely make it of interest across the fields of human genomics, physiology, biomedicine and microbiology.

However, the presentation of the data and the interpretation and discussion of the results need to be substantially revised. In particular, a large part of the main text is dedicated to content akin to material and methods, while too little focuses on interpreting and discussing the results, both based on the analyzed data and in the context of prior literature. More importantly, the conclusions often lack sufficient statistical support, a limitation that is not appropriately discussed.

We appreciate the reviewer's insights, which have helped to reorganize the paper and online data portal. Also, we have now annotated and added more samples to the Biobank (n=2,911), and included more methods about their protocols in a dedicated Protocols paper and also online. This paper can be found here:

<https://pubmed.ncbi.nlm.nih.gov/37205403/>

Moreover, we have moved a large part of the materials and methods into the supplement, and we have updated the Figures to highlight the results, rather than the processing tools utilized to create the figures. While these methods are still important, we now highlight them in the supplement and also on the NASA GeneLab repository.

Major concern 1: limits of the study design

Although the complexity of the dataset is well introduced through a dedicated section as well as Figure 1, the authors do not mention or describe the cohort in the main text (crew of n=4). This information is crucial for the reader to properly place this work in the context of previous studies and evaluate the robustness of its conclusions. In particular, the limited cohort size has important consequences for statistical significance. Specifically, statistically significant results are a valuable contribution over the NASA Twins Study but the absence of statistical significance should not be interpreted as an absence of difference, as it may rather result from a lack of statistical power. This point impacts the reporting and interpretation of results across the manuscript and should be addressed throughout (see detailed examples below).

We have updated the main text to clarify the details about the crew, including their age range and background. Moreover, we have enabled direct comparisons to the NASA Twins Study and JAXA study on the data portal.

The authors motivate the importance of the dataset by introducing the need to better understand the impact of spaceflight on the human race (line 105) and mention the role of inter-individual differences (line 574). As such, the authors should discuss the diversity and representativeness of the crew to better place the contribution of this work with regard to these aspects.

We agree with the reviewer, and we have highlighted this broader biomedical range of the crew in the revision, and their ancestry is self-reported Caucasian (3) and African American (1). We have also created a dedicated new analysis and Figure 5 that characterizes the most variable analytes in response to spaceflight.

Although the longitudinal design and logistical constraints of the study are well understood, the lack of control (i.e., individuals that would go through the same treatment as the real participants except for the flight) prevents untangling the impact of confounding factors from spaceflight itself (e.g., similar environment, proximity or diet change). The authors do mention that the data is compared to other “control sets” (line 139), but do not mention those further. The authors should expand on whether these sets can help address potential confounding factors and should discuss any potential confounding factors when interpreting their results. In addition, the observed variations should be put in context of or controlled for “normal” (i.e. non spaceflight related) variation, where available, for interpretation.

We have updated the paper to show some of the data sets from the other missions, and we have also included more details about the possible confounding variables from the collection and processing of the samples. Also, we have included several statistical models in the revision for the associations reported in the paper.

Major concern 2: statistical support and validity of conclusions

The authors make strong statements with important implications (e.g., lines 225, 239-241, 253-254, 261-262, 730) based on absence of significance between recovery time points and pre-flight time points. With a cohort size of n=4, interpreting absence of significance as absence of difference is a

strong assumption that warrants additional discussion before reaching such a conclusion. Especially when the conclusion is that changes revert back or return to baseline and therefore that, for a given parameter, spaceflight has limited consequences (e.g., “These data suggest that even short duration spaceflight [...] poses no obvious risk of mutagenesis and instability.” line 225).

We have now updated the text, and especially the discussion, to highlight the limitations of the study and the statistical power.

In addition, the lack of detailed information about the statistical procedure used (Which test is used? is it paired? What exactly is being compared to what? Does the distribution of the data meet the assumption of the test?) hampers proper assessment of the reported results. Please add detailed statistical information wherever reporting p-values (e.g., lines 284, 318, 737, 805, 1258, 1282).

We have greatly expanded the materials and method section, and we also now post the raw code and methods used for the statistical tests for all analyses, which can be found here:

<https://github.com/eliyah-o/inspiration4-omics>

The authors report that “the telomere length of all Inspiration4 crew members showed telomere elongation during spaceflight (17-22% longer), though this trend was statistically significant ($p=0.029$, Student’s T-test) only in C003 and (Extended Data Fig. 5c)”, which contradicts the caption of the Extended Data Figure 5c “Although not statistically significant, telomere lengthening during spaceflight can be seen in the aggregate in-flight measurements and R+1”. Please clarify why these statements are contradicting each other, how significance is tested on a per-astronaut basis, and whether a paired Student’s T-test or a paired Wilcoxon test (which appear more appropriate based on the data in Extended Data Figure 5c) between pre-flight and in-flight time points would be significant. Furthermore, although there is a clear trend in shortening of the telomere during recovery, this can hardly be interpreted as mostly returning to baseline (line 503) as the last measurements show a ~10% increase in length.

This is an important point, and we have updated the caption for the figures to match, and also clarified the tests used in the manuscript.

The authors state that “The number of differentially expressed genes (DEGs) and differentially accessible regions (DARs) showed the largest change at the immediate post-flight time point, R+1, compared to the pre-flight time points, and then mostly returned to baseline levels during the recovery time points (Fig 2c).” (Lines 239-241). Please clarify how this statement is reached based on the data presented in Figure 2? My understanding is that hundreds of genes are differentially expressed across all cell types and hundreds (?) of regions were differentially accessible (at least for dendritic cells) between the recovery time points and the pre-flight time points. These values are also reported in the text (lines 241-243) and represent between 30 to 50% of the in-flight values, which hardly seems like a return to baseline.

We have updated the text to clarify that this is a trend towards returning to baseline levels, but that the recovery is not complete. We have also included a breakdown of each cell types DEG and DAR change, and clarified how these were processed in the discussion, and we have updated the analysis into a new Figure 4 that is more clear for readers.

In the section “Metagenomes and Metatranscriptomic Profiles of the Skin and Dragon Capsule”, the authors describe and analyze a set of metagenomes and metatranscriptomes and conclude that “Despite the overall stability of microbial domains, variation on the species-level was readily apparent, indicating the presence of community-level, within-domain shifts specifically as a function of flight (Fig 6b).”. This is however difficult to link to the data presented (Figures 6b,e,f) and should be supported by beta-diversity metrics and their visualization rather than the raw abundance profiles (which need to be visually compared). Interestingly, the authors report taxa that are differentially abundant or expressed pre- or post-flight (line 482, Figure 6 g-h). However, the representation of the data or discussion of those results does not mention what taxa are impacted.

In addition, the authors interpret the different responses of microbiome composition across body sites as a rapid microbiome transfer with the capsule and other crew members (line 494). However, the changes observed could also be explained by changes in species-level microbiome composition in response to the same environmental drivers. To support the claim that transfer is occurring, the authors should go beyond species-level profiling and conduct strain-level analyses (see <https://www.nature.com/articles/s41586-022-05620-1>).

Major concern 3: contextualization with existing literature

The authors place the Inspiration4 dataset and its analyses in the context of previous studies (the NASA twins study and the JAXA CFE, lines 118-120) but it is not immediately clear what is similar and what differs between those studies. Specifically, what are the different durations and altitudes studied? I note that the apogee altitude is later mentioned in the results for Inspiration4 and the ISS, but the link between the ISS and the previous studies is not explicit. In addition, additional studies with larger cohorts (e.g., n=11 for telomere length and n=18 for blood plasma protein expression) should be mentioned to properly contextualize the data presented.

We have now included details about the other missions, which were all at the ISS (400km), notably lower than the 590km of the I4 mission. We also now include a means by which to plot the relevant data sets and values from other studies (NASA, JAXA) and to compare them, to the i4 Resource page.

Multiple parts of the manuscript lack contextualization, interpretation and/or discussion. However, I note that sections reporting on (epi)transcriptome of whole blood, proteome and metabolome of blood plasma and EVPs provide a better level of description, interpretation and contextualisation (e.g., line 291 “To construct the first epitranscriptome map...”, line 321 “Consistent with previous studies, we observe dysregulation...”), although the introduction to these sections could be better motivated (why are these parameters studied in the first place?) and significance of the results could be better communicated (in these sections or in summary). Overall, for the reader to better understand the advance over the art would require an improved, harmonized contextualization of the

results with existing literature throughout the manuscript sections (e.g., “Single-Nuclei Gene Expression, Chromatin Accessibility, and Immune Profiling of PBMCs”, “Metagenomes and Metatranscriptomic Profiles of the Skin and Dragon Capsule”).

We agree with the reviewer on the importance of context, and we have updated the text to include more references to prior work, and noted the cases when our paper is the first to profile some aspects of the crew (e.g. direct RNA, spatial transcriptomics). As suggested by the reviewers and the editors, we have focused this paper on the data and biobank resources, the difference and similarities to other missions, and the inter-astronaut variation measures.

This current limitation impacts, for instance, the significance of the analyses of microbiome data (of note, the four citations used to provide context in this section are self-citations, although the human microbiome has been a vibrant field of research for over a decade). In particular, the authors should address the following questions when introducing this section or discussing its results:

- What is the expected variability of the human microbiome composition at the taxonomic levels studied here? What is the rationale behind testing the domain-level stability of the microbiome? Are changes expected? How does the set of recovered genomes compare to the current set of genomes available for the human microbiome (e.g., <https://doi.org/10.1016/j.cell.2019.01.001>, <https://www.nature.com/articles/s41564-021-01011-w>)?
- How do the observed transfers relate to the expected transfer landscape of the human microbiome (<https://www.nature.com/articles/s41586-022-05620-1>)? How is the transfer different from living together in a household?
- Since several of the previous markers studied are indicative of inflammation, what could the impact of inflammation be on the microbiome composition?

Beyond taxonomic profiles, the authors built and briefly described a gene catalog. Although investigating the functional shifts in the microbiome as a function of flight is arguably of high interest, this catalog is not used in the presented analyses.

The authors also mention that “some of these genes correlated with the known, skin disease-related microbiome changes” (line 420). Please add a reference to support this statement. In addition, the statement is followed by the mention of family-level changes in microbial composition associating to the expression of specific genes. This association needs to be refined since microbial families encompass a large diversity, only a fraction of which would be associated with a disease phenotype. Note those correlations between the skin expression profiles and the microbiome composition (line 419) before introducing the microbiome data and its analysis (starting 433).

Given the complexity and importance of these microbiome and analyses, we have responded to these points by the reviewer and expanded them into a dedicated, microbiome-host interaction manuscript. We have included this paper with the updated manuscript, which includes more of the above references and details, but we have focuses on the species that are most variable in this paper, and also on the data resources that are now online as part of the data portal.

Figures and data presentation

Generally, the figures are very inconsistent, including the visualization, title and caption. Some of the visuals are excellent or have a lot of potential (e.g. Figure 1, 3, 4 and 5) while others are really difficult to extract information from (Figure 2 and 6). In addition, some titles only describe the approach while others convey a result (which is preferable in my opinion as it guides the reader). The reporting of p-values and/or lack of significance is often inconsistent and should always mention the statistical test used and what is being compared to what. More specifically:

We have updated the figures to improve readability and coordination across the figures.

Figure 2:

- In the caption of panel b, please specify what data is being projected.
- In panel c, my understanding is that each bar is an average of multiple time points, each of which is an average of four individuals. This should be reflected by the presence of error bars to capture the variability. Should it be DAP or DAR for the bottom subpanel? Please correct and adjust the caption accordingly. The y-axis scale of the bottom subpanel does not allow for proper evaluation of the data. Please switch to a log-scale or add a break.
- Panel d is too small to extract data from and the color codes are not explained in the legend nor caption. Maybe harmonize the visuals with figure 3b/c.
- Panel e is also difficult to read. Note that humans are not good at comparing area sizes and thus encoding a quantitative variable that way is not desirable. An alternative could be to use the split rectangle approach of figure 3 with two different color codes.
- In the captions of panels g and h, please specify (1) the statistical test being used and (2) what is being tested.

We have updated the Figure now to improve the clarity and details on the methods used in the caption, and also moved some methods in to a dedicated Protocols paper here:

<https://pubmed.ncbi.nlm.nih.gov/37205403/>

Figure 3:

- In panel c the triangle symbols are too small, please the same size as in panel b.
- What is the significance threshold and what is the statistical test used?

We have modified Figure 3, panel c, to make the triangle symbols the same size as in panel b. Please find the revised figure in the Supplement, and we have now included the significance threshold and statistical program within the Figure Legend. In terms of underlying statistical methods, *edgeR* utilizes a statistical test analogous to Fisher's exact test, but adapted by the authors to fit overdispersed data. *methyKit* likewise utilizes Fisher's exact test. *DESeq2* employs the Wald test.

Figure 4:

- Consider replacing some of the color coding by textual information (akin to Figure 3).
- Please specify the statistical tests used.

Thank you - this figure has been updated and now focuses on the changes over time across the crew.

Figure 5:

- The figure caption needs more details.
- Line 789: “Most of the ROIs showed good clustering around ROI types in both time points”. Please provide statistical support to this claim (e.g. with a PERMANOVA).
- In panel c, my understanding is that both time points are projected in the same embedding and split for visualization purposes. If this is correct, please add the other time point as gray dots in the background of each panel to allow for robust comparisons between time points.
- In the caption of panel d, please clarify what is being correlated to what. Is it the expression of human genes involved in skin diseases? What measure of microbiome composition is used? Is it relative abundance? Why are genera and species levels mixed?
- Legends for panels e and f are cropped.

This figure has been updated. Thank you.

Figure 6:

- Figure 6c: In the current presentation, the colors (taxonomy) are difficult to read and make sense of.
- Figure 6e-f: These panels are difficult to read and interpret. I suggest considering dimension reduction approaches if the point is to communicate the differences in structures.
- Figure 6g-h: These results are particularly interesting but their interpretation is limited by the lack of annotations of the individual points (e.g. taxonomy). Please clarify what each point corresponds to (e.g. a species based on the kraken profiles?). Please specify the statistical tests used.

Please provide details of the statistical methods used in the different analyses, either in a dedicated section or with each analyses (Some are available in supplementaries but should be more readily accessible).

We have now expanded several pages of additional details in the materials and methods section, and we have also submitted a dedication protocols paper to provide access to the full, in-depth protocols and processing for the mission.

The preprint can be found on BioRxiv, and we have this paper also currently in review.

<https://www.biorxiv.org/content/10.1101/2023.05.02.539108v1>

Minor comments

Data sharing:

The authors have shared the data through a dedicated repository (OSDR) hosted at NASA. However, the identifiers provided in the table of the “Data Availability” section do not appear to be live at the moment. This data is indeed a valuable resource for the research community and should be accessible to a wide audience, which could be improved by sharing data through standard repositories. In particular, it would be valuable if non-sensitive sequencing data were available through INSDC compatible repositories (e.g., NCBI/SRA, EBI/ENA, DDBJ).

The data is now live in GeneLab at the following links:

OSD Number Private preview link

OSD-569 (whole blood) <https://osdr.nasa.gov/bio/repo/data/studies/OSD-569/preview/SQ3QgEzNMH5q1X19jkWJxVLF5Lx5RQtp>

OSD-570 (PBMC) <https://osdr.nasa.gov/bio/repo/data/studies/OSD-570/preview/y0lXYJACBT2kLVOnsLESaPa1NrI3lKyh>

OSD-571 (blood plasma) <https://osdr.nasa.gov/bio/repo/data/studies/OSD-571/preview/Wr-IXUj2sAQ4dOp4UcAuajlGfMcZAX0E>

OSD-572 (skin, oral and nasal swab) <https://osdr.nasa.gov/bio/repo/data/studies/OSD-572/preview/CuhxWdYAPRX4QJRLd74yzdXjon2OxrXm>

OSD-573 (dragon capsule) <https://osdr.nasa.gov/bio/repo/data/studies/OSD-573/preview/Ze2eyzuoTvJ7CI0887nweUeBe9GqYTZN>

OSD-574 (skin biopsy/swab) <https://osdr.nasa.gov/bio/repo/data/studies/OSD-574/preview/cRBHSdY8b-ekuGYCFZd8bexhWfGIdM9i>

OSD-575 (blood serum) <https://osdr.nasa.gov/bio/repo/data/studies/OSD-575/preview/bIm-fESutlV3vdJofJPygFXGW6omqd8e>

OSD-630 (stool) <https://osdr.nasa.gov/bio/repo/data/studies/OSD-630/preview/BilzXsJxraKrE9sYDCsOh-MLL3yPppSC>

OSD-656 (urine) <https://osdr.nasa.gov/bio/repo/data/studies/OSD-656/preview/TO0a-2qmUg4xGuGnPidm40r4yt1HYIGq>

In addition, the authors provide a web application for users to explore and mine the processed datasets. Although I commend their effort to share the data in a meaningful way, it is currently difficult to use in particular due to a lack of documentation. Some questions that may help guide this include: what are the four different drop down menus at the top? Why are the categories all the same for the different types of data? In addition, from the manuscript, it reads as if all the data is available through the web portal, but it currently appears to be restricted to PMBCs.

We have updated the site to give more context about how the data was generated and the links to the papers from which the data is plotted and analyzed. We have also updated the discussion section to point to additional papers and resources.

Methodology:

The metagenomic and metatranscriptomic analyses presented in this work employ various bioinformatic pipelines that reflect a set of state-of-the-art and rigorous approaches. Having said that, I do have a few minor comments:

- Was the binning of metagenomic contigs done single-sample or cross-sample abundance correlation? (although the latter would be preferable it would at least be good to specify the approach taken)
- Although I commend the authors for the utilization of multiple approaches for taxonomic profiling, I note that they are all based on strategies comparing kmers to whole genomes. Thus, the value of this multi-approach taxonomic profiling would largely benefit from the inclusion of approaches

relying on different strategies (e.g. marker gene based), which were also shown to perform best in a recent community-driven benchmark study (<https://www.nature.com/articles/s41592-022-01431-4>).
- Regarding the ad hoc approach used to assign viral taxonomy, how does it compare to previously published software? (e.g., vConTACT v.2.0; <https://www.nature.com/articles/s41587-019-0100-8>)

Note that p-values are not consistently reported throughout the manuscript (p=XX, p-value = XX, p-value=XX, p value = XX, p.adj <XX).

Line by line:

Line 160: The acronym PBMC is used but not defined. There are other such examples in the manuscript (e.g. MYC, SNCA, KRAS).

Line 219: 'C003', which refers to the crew member, is not defined at that point in the text. I suggest to couple that with the introduction of the crew size.

Lines 219, 221-224: A sentence ends with "and" and in another VAF is both subject and object. Please check the writing here and throughout.

Line 477: Should the reference to figure 6 refer specifically to panels 6e-h?

Lines 512/524: Should "first-ever" have a different connotation than just "first"?

Line 1214: Authors contributions are very minimal and the contributions of many authors are not specified.

Thank you - the main paper here has been updated with the above comments, and the now separate, dedicated microbiome paper has responded to the above comments as well.

We greatly appreciate the responses and guidance by the reviewers and thank them for their time.

Reviewer Reports on the First Revision:

Referee #1 (Remarks to the Author):

The authors have considered my suggestions and answered all remaining questions. I am content with the revision and have no further comment on the scientific aspects. The manuscript provides a useful description of the resource and relevant links to the other studies that focus on data interpretation.

A final cosmetic note relates to the font sizes used in the multi-panel figures. Besides Fig 4, I found these hard to read when printed, which compromised my experience with the shown information. The legends could also be improved in how they describe the shown content.

Referee #2 (Remarks to the Author):

In the revised manuscript, the authors made substantial changes to the organisation and presentation of results. In contrast to the original manuscript, which presented detailed results from all data types, the revised manuscript emphasises specific aspects of the data and their novelty over previous data sets. To achieve this, some results and overview figures have been moved to the supplement, and more details and results are presented in additional manuscripts. While these changes reduce the scientific content of this manuscript, they have greatly improved the readability and accessibility to a broader audience, thereby providing a comprehensive overview that more clearly highlights the novelty of this space omics resource.

The authors have replied to the original comments with additional information and details that address these concerns. Some comments related to analyses or figures that have since been entirely replaced, or moved to dedicated manuscripts which are separately reviewed. The data description and methods sections has been greatly improved, the analysis code has been made available, and the presentation was improved for clarity.

Minor comments

-Figure 2b, gene expression: It is not clear what the x and y axes are, which one is Inspiration4? I would also recommend to add consistent labels (e.g. " $\log_2(\text{fold change})$ ") The different colours are very difficult to find in the figure, so the results don't intuitively match the claim ("The overall trends of DEGs matched across both cohorts"). The agreement appears to be quite clear for CD4+ T cells and CD8+ T cells, but the colour description does not match.

- Analysis of coefficient of variation: This analysis captures the variation across individuals for different omics data and presents a new perspective across data modalities. This analysis could benefit from improved figure labels, for example the x axis for Figure 5a, how are these sorted? The y-axis of Figure 5c is missing (this should be the CV). How is the x-axis sorted?

- I commend the authors for making the code available through GitHub. Given the novelty of these data, this will be a great resource for the community. While many code files are uploaded directly, some are only uploaded as HTML files, which are not rendered on GitHub, and therefore not readable, unless they are downloaded and opened with a browser. It seems that the authors have used a markdown system for these analysis, so the original code file could be uploaded to make this repository more easy to browse through GitHub.

Referee #3 (Remarks to the Author):

My concerns have been addressed. This will be a highly impactful paper.

Referee #4 (Remarks to the Author):

Thank you for your work in the revision process, I have no other comment to add

Referee #5 (Remarks to the Author):

We have now read the updated manuscript from Overbey and colleagues. While authors have made substantial revisions to the manuscript (in particular the figures), essentially very few of our conceptual comments (nor many of the other reviewers') were more than superficially addressed, and thus even in this highly modified manuscript, most of our substantial conceptual and data quality concerns remain. In short, it remains challenging to understand what can be said about the molecular phenotype of space flight in this paper, both because of the limited number of individuals profiled, and because there has been no attempt to isolate the effects of confining individuals in a small space for days from actual effects of the space environment, but also because of a lack of clear demonstration that these data are of high quality and are devoid of large batch effects that would confound the modest biological differences observed.

Our concerns about impact are evident from the abstract:

“Spaceflight induces drastic molecular, cellular, and physiological shifts in astronauts and poses myriad biomedical challenges to the human body, which are becoming increasingly critical as more humans venture into space. Yet, current frameworks for aerospace medicine are nascent and lag far behind advancements in precision medicine on Earth, underscoring the need for rapid development of space medicine databases, tools, and protocols. Here, we present the Space Omics and Medical Atlas (SOMA), an integrated data and sample repository for clinical, cellular, and multi-omic research profiles from a diverse range of missions, including the SpaceX Inspiration4, Polaris Dawn, Axiom-2, NASA Twins Study, and JAXA missions. The SOMA resource, available through several interactive online portals, represents a >10-fold increase in total publicly available space omics data, with matched samples available from the Cornell Aerospace Medicine Biobank. The Atlas includes extensive molecular and physiological profiles encompassing genomics, epigenomics, transcriptomics, proteomics, metabolomics, and microbiome data sets, which reveal some consistent features across missions, including cytokine shifts, telomere elongation, and gene expression changes, as well as mission-specific molecular responses. Leveraging the datasets, tools, and resources in SOMA can help accelerate precision aerospace medicine, bringing needed health monitoring, risk mitigation, and personalized countermeasures for pending lunar, Mars, and exploration-class missions.”

Apart from the announcement that data has been generated, the findings of this paper are communicated in this half-sentence: “... which reveal some consistent features across missions, including cytokine shifts, telomere elongation, and gene expression changes, as well as mission-specific molecular responses.” It appears that no cytokine shifts are statistically consistent with other data sets, telomere elongation is significant only for one individual, and gene expression analysis compared to other data sets is extremely challenging to understand (see below), and it is hard to understand what “mission-specific molecular responses” means. In short, the scientific contribution of this manuscript appears to be in the generation of data sets for others to investigate, it would appear.

Our initial critique questioned what could be learned from data integration. In short, the authors appear to rebut this question by stating these types of data have not been collected before, which does not really wrestle with the fundamental challenge of articulating what biologically we learned from these data and how certain we can be that these claims are actually not confounded by other effects (i.e. how certain we can be that they are reproducible observations of real molecular changes). The authors also seem to have substantially shifted the focus of this manuscript to away from some data types and onto others. Indeed it seems as though many of the data modalities that were initially highlighted in this manuscript have also been removed to different stand-alone papers. While we had hoped that the findings in these distinct molecular methods might be used to reinforce one another and build towards a coherent biological story, it appears that they have largely been simply removed, which does not address our overall critique.

We were initially also extremely concerned that these data were not placed in context of data generated over decades of spaceflight. The authors have now compared some of their gene expression data to the NASA Twins Study, but this comparison honestly creates far more questions than it answers. Figure 2B is the central figure describing this comparison, and, quite frankly, it is nearly incomprehensible. The quality of this figure (from the data presentation to the legend and axis labels to the description of what is being displayed) is simply not up to the standards of a Nature paper. We are unable to interpret what it is meant to convey, and the rather haphazard description of the very small number of common differentially expressed genes in the text (without any biological “through line”) makes it impossible to really understand the nature of the similarities and differences between these data sets. There appear to also be large systematic differences between the differential expression results from the two studies (seen as vertical offsets of points in Fig 2B, as far as I can understand it). The authors also do not clearly compare the overlap of these data sets. Both expression datasets have hundreds of differential genes observed, and it would appear that the overlap is only 8 genes? Is this correct? We had similar confusion in the cytokine data – it appears there is simply no agreement with respect to significant changes in cytokine expression in this separate cohort. Is this not concerning? Is this evidence for “mission-specific molecular responses” or just for batch effects? We also were concerned that batch and small sample sizes might inflate significance using parametric assumptions and suggested more extensive analysis of power including permutation testing – this critique seems to have been ignored.

The cfRNA analysis is interesting, but likewise challenging to interpret. It appears that some of the largest changes observed were in L-3 (pre-flight), which makes it challenging to understand what standing variation exists across batches and whether this might be a result of common travel (?) experiences or simply batch (technical) variation. Also, while deconvolution is a method that others have developed and deployed, the statistical rigor of these analyses are challenging to infer. What is the limit of detection or the limit of quantification of these deconvolution methods? Is seeing 0 vs 0.02 fraction melanocytes something we should believe? How do we know? Also, it appears that these significance values are not controlled for multiple hypothesis testing, as far as I can tell. The authors attempt to link this finding to other data sets with “Notably, the melanocyte cell proportions demonstrate significant changes in different skin layers during flight (Fig 3f) that were also found in the spatial skin transcriptomics data, which provides additional evidence of adaptive skin responses to the space

environment.” However they do not cite exactly what data they are attempting to link to, so this attempt at coming up with an integrative observation (having at least some data sets pointing to consistent findings) is not possible to really interpret.

After a few attempts at understanding Figure 4, we remain almost confused as to what is attempting to be demonstrated, or the ways that the data has been analyzed. The second half of this figure also seems to fundamentally confuse statistical power and magnitude of effect. No sense of the strength of the pathway enrichments is given, nor any sense of if the uniqueness of the pathways in the cell types might be driven by changes to the power of measuring different differentials (which of course scales with the number of cells etc.). The use of the number of differential genes from the RNA-seq data as a metric is also fraught, as the power to call differentials is clearly affected by sample to sample variability (in collection quality, cell number, sequencing depth, etc.). If the authors want to use the absolute number of differentials to quantify the severity of a response and its return (or not) to baseline, they first need to convincingly demonstrate that these power considerations are not an issue. For example, if you took two multiome samples from the same astronaut on the ground and processed them on back to back days, would you see the same number of differentials? Fewer? The analysis of these data sets is not well described or controlled with null distributions.

Thus the conclusion from analyzing longitudinal snRNA-seq across multiple individuals and conditions appears to be: there are differentially expressed genes. There seems to be no attempt to actually describe the molecular basis for the physiological changes resulting from spaceflight. Furthermore, a similar concern applies to the analysis of the cell free nucleic acids (is it biologically relevant that snoRNAs decrease in abundance in cfRNA post-flight?), m6a (what is the significance of the trajectories of methylated sites, as in, are there meaningful biological differences between Set M-I through M-III?), and other modalities. Overall, this paper continues to read as a laundry list of assays that generates troves of data that remain analyzed in a way that is very hard to interpret or understand in the context of 1) reproducibility and rigor or 2) biological mechanism.

Figure 5 is likewise confusing. It would appear that the message from the metagenomics is that “things are variable.” Then the figure analyzes variability in all data sets, and sees disparate results. Given the immense number of hypotheses tested, it appears that there is not a uniform means to control hypothesis testing across data sets. When just reporting “significant” findings by testing many data sets comprehensively for all comparisons, the right way to control would really be to have a uniform correction for all tests performed for all assays (otherwise p-values will remain inflated). It is simply unclear what associations here are to be believed given this fundamental issue (quite apart from the QC issues that remain unaddressed).

The authors make highly speculative statements in the discussion, including: “Hence, the systematic analysis of the molecular and cellular changes observed post-flight afford us with a unique opportunity to capture naturally occurring health repairing mechanisms which can be used for therapeutic target discovery.” How might this happen?

In the discussion, the authors state:

“Of note, these data are the first-ever joint single nuclei chromatin profiles (snATAC-seq) for astronauts, and they also leverage new methods that can track gene expression and epigenetic changes within the same cells. This single-cell, dual-measurement assay provides mechanistic insights into the molecular changes and regulatory response to spaceflight, and the data revealed distinct levels of stress and adaptation by each cell type.”

However, snATAC-seq data is basically not discussed in this manuscript (other than to mention it was generated). Using these data to validate some of the gene expression changes observed (via concomitant changes in local chromatin accessibility) would be welcome, but it would appear that these data sets have effectively been eliminated from the main flow of the paper. It is also entirely unclear how “this single-cell, dual-measurement assay provides mechanistic insights into the molecular changes and regulatory response to spaceflight, and the data revealed distinct levels of stress and adaptation by each cell type.” This is not consistent with the commonly-understood definition of “mechanism.”

Globally, there is a uniform lack of 1) communicating quality control measurements of any of the data sets, and 2) any discussion or analysis relating to attempts to understand or use internal controls for batch correction – crucial concerns for a large-scale data set with data potentially collected in very different conditions. Furthermore, it seems that the authors claim multiple times that changes are to be believed because “... QC metrics for the samples were the same” or “However, they increased in B cells, CD4+ T cells, CD8+ T cells, and CD14+ monocytes (Fig 4a), even though QC metrics were similar across timepoints.” However no citations appear to any figures or supplements or anything to support these claims that the data is high quality and uniform and devoid of concerns regarding batch effects. We believe this lack of support for crucial data quality claims does not rise to the level of rigor required for a Nature paper.

Finally, nearly every reviewer raised the point that this study made no attempt to control for the non-spaceflight-intrinsic stressors of this trip (e.g. confinement and diet, compared to microgravity and radiation exposure). At the very least, the authors should tamp down their attribution of these changes to specific causes in light of this crucial missing control. However, this lack of control makes it effectively impossible to really understand what these observations mean in the context of spaceflight specifically.

In short, the paper continues to lack coherent biological insight or observations of physiological changes that are convincingly supported by multiple data modalities or in other data sets. The authors make many dramatic claims about the need for multiomic profiling of astronauts (e.g. “single-cell sequencing must be implemented more regularly in spaceflight”) and their “unprecedented view of the multi-system omics changes before, during, and after spaceflight.” Yet we are left wondering about what, precisely, has been learned about the physiological response to spaceflight from the authors’ 10s of billions of sequencing reads, and are also left without a clear picture of the quality of these data sets that are, it would appear, the main deliverable of this work.

Referee #6 (Remarks to the Author):

I appreciate the authors' efforts to respond to my and other referees' comments and to update their manuscript accordingly. I believe the revisions have largely and appropriately addressed my concerns and have improved the clarity and quality of the manuscript. Although I commend the authors for their impressive work, I would also like to raise a few remaining, albeit small, points:

Major points:

- I was not able to access the data deposited at OSDR. For instance "<https://osdr.nasa.gov/bio/repo/data/studies/OSD-569>" showed the message "The study you requested is not available. Please contact the OSDR staff if you have further questions". This occurred across sample types and remained a problem despite multiple attempts on different dates.
- I commend the authors for the improved manuscript and report of statistical significance (or lack thereof). I did however note a few instances (lines 272, 343, 468) that may benefit from some additional nuancing, for instance by mentioning that the absence of significance "suggests" a return to baseline. Although my concerns regarding statistical reporting have been addressed throughout the main text and thanks to additional supplementary methods, the inclusion of a short paragraph to discuss the interpretation of lack of significance based on a small cohort (n=4) would be welcome; in particular to mention that lack of significance can suggest a return to baseline, but that impacts below a given effect-size cannot be detected with the current dataset. The minimal detectable effect size could even be modeled for some of the variables tested under a hypothesized probability distribution based on sample size and variance.

Minor points:

- The web portal for the available biobanked samples (<https://cambank.weill.cornell.edu/>) is a valuable addition. However, the web interface could include some additional documentation along with a possibility to browse the list of samples.
- Similarly, the addition of the SOMA (https://soma.weill.cornell.edu/apps/SOMA_Browser/) and I4 Microbiome (https://soma.weill.cornell.edu/apps/I4_Microbiome/) browsers in addition to the initial I4 Multi-ome browser (https://soma.weill.cornell.edu/apps/I4_Multiome/) provide interactive access to important information. However, inclusion of additional documentation on those web portals (in particular for the multi-ome) would be needed to fully enable the community to benefit from those resources.

Author Rebuttals to First Revision:

We have responded to each point below, indicating our responses **in blue** below each reviewer comment. We have also included more details about the code and data accessibility, as well as a preview of the data portal and NASA GeneLab repositories, which also hold annotated versions of the raw data, computational code, and processed data. We are grateful for your time, and we believe the reviewers will find the updated manuscript much improved and hopefully suitable for publication.

Thank you.

Reviewer Comments:

Referee #1:

Remarks to the Author:

The authors have considered my suggestions and answered all remaining questions. I am content with the revision and have no further comment on the scientific aspects. The manuscript provides a useful description of the resource and relevant links to the other studies that focus on data interpretation.

A final cosmetic note relates to the font sizes used in the multi-panel figures. Besides Fig 4, I found these hard to read when printed, which compromised my experience with the shown information. The legends could also be improved in how they describe the shown content.

Thank you for the positive response, and we have updated the figures and legends to improve readability and clarity.

Referee #2:

Remarks to the Author:

In the revised manuscript, the authors made substantial changes to the organisation and presentation of results. In contrast to the original manuscript, which presented detailed results from all data types, the revised manuscript emphasises specific aspects of the data and their novelty over previous data sets. To achieve this, some results and overview figures have been moved to the supplement, and more details and results are presented in additional manuscripts. While these changes reduce the scientific content of this manuscript, they have greatly improved the readability and accessibility to a broader audience, thereby providing a comprehensive overview that more clearly highlights the novelty of this space omics resource.

The authors have replied to the original comments with additional information and details that address these concerns. Some comments related to analyses or figures that have since been entirely replaced, or moved to dedicated manuscripts which are separately reviewed. The data description and methods sections has been greatly improved, the analysis code has been made available, and the presentation was improved for clarity.

Thank you for the positive response and review, and we have updated the paper relative to the below comments.

Minor comments

-Figure 2b, gene expression: It is not clear what the x and y axes are, which one is Inspiration4? I would also recommend to add consistent labels (e.g. "log₂(fold change)") The different colours are very difficult to find in the figure, so the results don't intuitively match the claim ("The overall trends of DEGs matched across both cohorts"). The agreement appears to be quite clear for CD4+ T cells and CD8+ T cells, but the colour description does not match.

We have updated this plot to now show a violin plot with all DEGs as individual points, to highlight the comparisons between the missions, and we have also updated the caption description for the colors. The updated plot highlights how the T-cell populations across both missions share the same up- and down-regulated genes, across hundreds of DEGs between both types of sorted T-cells and single-cell annotated T-cells (CD4, 552 DEGs, and CD8, 608 DEGs, both $<2.2 \times 10^{-16}$). The similarity in violin plots for B-cells was expected from prior work in the Twins Study (Garrett-Bakelman et al. 2019), which showed B-cells as either not altered (telomere length) or less responsive (gene expression) to spaceflight.

- Analysis of coefficient of variation: This analysis captures the variation across individuals for different omics data and presents a new perspective across data modalities. This analysis could benefit from improved figure labels, for example the x axis for Figure 5a, how are these sorted? The y-axis of Figure 5c is missing (this should be the CV). How is the x-axis sorted?

We thank the reviewer for their constructive feedback on our analysis of the coefficient of variation across our different omics data, and we have re-worked this figure to make it more clear. Specifically, we updated the plots to show the six different molecule types that were measured (whole blood RNA-seq, direct RNA-seq, EVP proteomics, plasma proteomics, plasma metabolomics, and the NULISaseq cytokine data) and show the range of variation of CV across violin plots from pre-flight, post-flight, and recovery. We have also labeled the most differential molecules for each type of assay, and then compare the differentials between pre- and post-flight for the crew. We have also included statistical tests for each of the differences between the time points, and also added permutation ranges of the data for the lower panels. This figure should be much more clear to the reviewer, which highlights that some modalities of the biology are more responsive to spaceflight than others, and that there are differences between the mission stages as well.

- I commend the authors for making the code available through GitHub. Given the novelty of these data, this will be a great resource for the community. While many code files are uploaded directly, some are only uploaded as HTML files, which are not rendered on GitHub, and therefore not readable, unless they are downloaded and opened with a browser. It seems that the authors have used a markdown system for these analysis, so the original code file could be uploaded to make this repository more easy to browse through GitHub.

We strongly agree the code should be fully readable online. We have updated the code base on GitHub to include file types that can support this access. Thank you.

Referee #3:

Remarks to the Author:

My concerns have been addressed. This will be a highly impactful paper.

Thank you for the positive response, and we agree this will be a valuable resource for the field going forward.

Referee #4:

Remarks to the Author:

Thank you for your work in the revision process, I have no other comment to add

Thank you for the positive response and for your help in the revisions.

Referee #5:

Remarks to the Author:

We have now read the updated manuscript from Overbey and colleagues. While authors have made substantial revisions to the manuscript (in particular the figures), essentially very few of our conceptual comments (nor many of the other reviewers') were more than superficially addressed, and thus even in this highly modified manuscript, most of our substantial conceptual and data quality concerns remain. In short, it remains challenging to understand what can be said about the molecular phenotype of space flight in this paper, both because of the limited number of individuals profiled, and because there has been no attempt to isolate the effects of confining individuals in a small space for days from actual effects of the space environment, but also because of a lack of clear demonstration that these data are of high quality and are devoid of large batch effects that would confound the modest biological differences observed.

Our concerns about impact are evident from the abstract:

“Spaceflight induces drastic molecular, cellular, and physiological shifts in astronauts and poses myriad biomedical challenges to the human body, which are becoming increasingly critical as more humans venture into space. Yet, current frameworks for aerospace medicine are nascent and lag far behind advancements in precision medicine on Earth, underscoring the need for rapid development of space medicine databases, tools, and protocols. Here, we present the Space Omics and Medical Atlas (SOMA), an integrated data and sample repository for clinical, cellular, and multi-omic research profiles from a diverse range of missions, including the SpaceX Inspiration4, Polaris Dawn, Axiom-2, NASA Twins Study, and JAXA missions. The SOMA resource, available through several interactive online portals, represents a >10-fold increase in total publicly available space omics data, with matched samples available from the Cornell Aerospace Medicine Biobank. The Atlas includes extensive molecular and physiological profiles encompassing genomics, epigenomics, transcriptomics, proteomics, metabolomics, and microbiome data sets, which reveal some consistent features across missions, including cytokine shifts, telomere elongation, and gene expression changes, as well as mission-specific molecular responses. Leveraging the datasets, tools, and resources in SOMA can help accelerate precision aerospace medicine, bringing needed health monitoring, risk mitigation, and personalized countermeasures for pending lunar, Mars, and exploration-class missions.”

We have updated the abstract to highlight a few more of the specific changes, and improved clarity throughout the manuscript to help address these issues.

Apart from the announcement that data has been generated, the findings of this paper are communicated in this half-sentence: "... which reveal some consistent features across missions, including cytokine shifts, telomere elongation, and gene expression changes, as well as mission-specific molecular responses." It appears that no cytokine shifts are statistically consistent with other data sets, telomere elongation is significant only for one individual, and gene expression analysis compared to other data sets is extremely challenging to understand (see below), and it is hard to understand what "mission-specific molecular responses" means. In short, the scientific contribution of this manuscript appears to be in the generation of data sets for others to investigate, it would appear.

We now highlight the specific cytokines that are significant (IL-6, IL-10, CCL2), included data from 27 other NASA astronauts to show the validation of these markers in most other crew members (see updated Figure 2), and we have updated the statistics for the tests by grouping across the crew vs. individual, which confirmed the overall increase in telomeres by the most significant crew member (T-test) as well as a group difference (linear model).

Our initial critique questioned what could be learned from data integration. In short, the authors appear to rebut this question by stating these types of data have not been collected before, which does not really wrestle with the fundamental challenge of articulating what biologically we learned from these data and how certain we can be that these claims are actually not confounded by other effects (i.e. how certain we can be that they are reproducible observations of real molecular changes). The authors also seem to have substantially shifted the focus of this manuscript to away from some data types and onto others. Indeed, it seems as though many of the data modalities that were initially highlighted in this manuscript have also been removed to different stand-alone papers. While we had hoped that the findings in these distinct molecular methods might be used to reinforce one another and build towards a coherent biological story, it appears that they have largely been simply removed, which does not address our overall critique.

Thank you for the review, which guided new analyses and integration for our revision, and we have made clearer the links to other data sets and papers. We have now completed extensive comparisons to other missions, integrated these data sets into the SOMA data portal, and also integrated the expression and chromatin data, as well as the other data modalities in the updated paper.

We were initially also extremely concerned that these data were not placed in context of data generated over decades of spaceflight. The authors have now compared some of their gene expression data to the NASA Twins Study, but this comparison honestly creates far more questions than it answers. Figure 2B is the central figure describing this comparison, and, quite frankly, it is nearly incomprehensible. The quality of this figure (from the data presentation to the legend and axis labels to the description of what is being displayed) is simply not up to the standards of a Nature paper. We are unable to interpret what it is meant to convey, and the rather haphazard description of the very small number of common differentially expressed genes in the text (without any biological “through line”) makes it impossible to really understand the nature of the similarities and differences between these data sets. There appear to also be large systematic differences between the differential expression results from the two studies (seen as vertical offsets of points in Fig 2B, as far as I can understand it). The authors also do not clearly compare the overlap of these data sets. Both expression datasets have hundreds of differential genes observed, and it would appear that the overlap is only 8 genes? Is this correct? We had similar confusion in the cytokine data – it appears there is simply no agreement with respect to significant changes in cytokine expression in this separate cohort. Is this not concerning? Is this evidence for “mission-specific molecular responses” or just for batch effects? We also were concerned that batch and small sample sizes might inflate significance using parametric assumptions and suggested more extensive analysis of power including permutation testing – this critique seems to have been ignored.

The reviewer highlighted several important questions, and we have updated and clarified the figures. Since the NASA Twins study and I4 mission are the only two human spaceflight data sets with single-cell resolution of gene expression, we wanted to make a comparison between these missions. The 8 genes mentioned before were the ones most significant across all cell types and time points, with each study’s DEGs analyzed separately, and then compared. But, as the reviewer notes, it is important to compare the totality of data between the missions, and we have updated the analysis for DEGs by cell type and plotted in the new Figure 2.

Fortunately, the updated figure makes our results much more clear and underscores the cross-mission concordance, especially for T-cells. The new Figure 2b shows a detailed violin plot (with all data points shown) of the comparisons between the missions. The violin plot highlights the DEG correlation of the T-cell populations, which share mostly the same up- and down-regulated genes in terms of directionality (>500 DEGs each), between both types of sorted T-cells and single-cell annotated T-cells (CD4, 552 DEGs, and CD8, 608 DEGs, both $<2.2 \times 10^{-6}$, Wilcoxon rank-sum test). In addition, as shown in Garret-Bakelman *et al*, 2019, the sorted CD19 B-cell populations did not show a significant response in telomere changes after spaceflight, and thus were hypothesized to be a cell population less affected by spaceflight. Indeed, we observe the same trend in our single-cell expression data from the I4 mission, and now highlight this as well in the text. We have also added permutation testing for some analyses of the variation in the measurements (below Figure 5).

For the cytokines, we have updated the plot to show data across all other available cytokine profiles from NASA (data from 27 other astronauts for the same molecules). We now highlight the pre-flight and post-flight differences that are the most consistent across missions. We now also show permutation testing for the expression, proteomics, and cytokine data (see Figure 5 details, below). For the cytokine data, we corrected for the limitations of the Twins subjects by using inflight samples and the longitudinal data of the ground control twin to get an estimate of variance within each cytokine, and calculated significance based on this estimate. Additionally, across all datasets, we have corrected for batch and baseline effects by normalizing each subject and dataset to their pre-flight value means, and doing a multivariate linear analysis with standard error estimates correcting for subject and time.

The data from this current I4 paper measured >200 cytokines/molecules from plasma, but the default NASA panel is smaller, and thus only 19 analytes are covered by both the Twins and I4 assays. We then checked the cytokines that spiked upon landing from the Garret-Bakelman et al, 2019 paper (CCL2, IL6, IL10, CRP, IL1ra). Of these five, CRP was not characterized in the I4 assays, leaving 4 cytokines. Of these 4 cytokines from the Twins Study, 3 out of 4 were significantly altered in the I4 mission (CCL2, IL6, IL10), and are now plotted in **Figure 2c**. In addition, we have also now added cytokine data relative to all other known human spaceflight data from NASA across varying mission lengths (6 months, 1-year). We have also now explained this in more detail in the text, and enabled the plotting of all these data (including NASA data for groups of astronauts) in the SOMA data portal for researchers.

The cfRNA analysis is interesting, but likewise challenging to interpret. It appears that some of the largest changes observed were in L-3 (pre-flight), which makes it challenging to understand what standing variation exists across batches and whether this might be a result of common travel (?) experiences or simply batch (technical) variation. Also, while deconvolution is a method that others have developed and deployed, the statistical rigor of these analyses are challenging to infer. What is the limit of detection or the limit of quantification of these deconvolution methods? Is seeing 0 vs 0.02 fraction melanocytes something we should believe? How do we know? Also, it appears that these significance values are not controlled for multiple hypothesis testing, as far as I can tell. The authors attempt to link this finding to other data sets with “Notably, the melanocyte cell proportions demonstrate significant changes in different skin layers during flight (Fig 3f) that were also found in the spatial skin transcriptomics data, which provides additional evidence of adaptive skin responses to the space environment.” However they do not cite exactly what data they are attempting to link to, so this attempt at coming up with an integrative observation (having at least some data sets pointing to consistent findings) is not possible to really interpret.

As the reviewer notes, more controls can help give context for our data, and we now included 35 additional control subjects in panel 3d, for both pediatric (n=17) and adult (n=18) cfRNA profiles from plasma, and which were processed in the same manner, by the same person, and in the same lab as the I4 crew. These new data highlight the similarity of the crew’s earlier pre-flight status with controls, and the subsequent changes immediately before and after spaceflight. The L-3 (pre-flight) timepoint samples indeed showed some changes in cfRNA levels for the

crew, which may be related to pre-flight activities (flight testing, quarantine, and pre-flight stress), but we wanted to keep these data since they may represent pre-flight stress. An advantage of longitudinally sampled data is that we can observe trends over time, allowing us to better determine persistent changes, and to also focus on the significantly-changed cfRNA in all post-flight samples, which is the updated focus of this part of the paper.

The four cell types presented (hepatocyte, kidney, hematopoietic stem cells, and melanocytes) are statistically different between pre- and post- time point samples (Wilcoxon rank sum test, Benjamini-Hochberg adjusted q-value < 0.05). Additionally, these cell types are elevated in all post- time point samples, and in some cases, such melanocytes, follow a longitudinally-defined trend. Also, the cell types we are presenting are not only elevated by a single time point, but rather by all time points post-flight. Furthermore, the cfRNA trends we observe post-flight are distinct from other sample sets our group has analyzed, including pediatric inflammatory conditions (Loy et al. 2023) and tuberculosis (Chang et al. 2023). For example, we observe elevated melanocyte and tendon cell type of origin fractions post-flight, two cell types that have not been seen before in previous studies, yet which we also observed in the spatial transcriptomics data. Meanwhile, the pre-flight samples are more similar to previous control samples we have studied (see updated Figure with the 35 new pediatric and adult cfRNA proportion controls, Figure 3d).

In terms of statistical rigor of the deconvolution algorithms, we point the reviewer toward the BayesPrism primary manuscript (Chu et al. 2022). The paper details their validation on both normal and cancer samples, and details how their algorithm computes the posterior sum of the fraction and expression of each cell type over multiple cell states within a given cell type, which they recommend as the most robust method for deconvolution. Additionally, a recent preprint was posted which reports a comparison of deconvolution methods and found that BayesPrism is the most robust algorithm (Hu and Chikina 2023).

On the question of magnitude - the deconvolution estimate for an individual cell type is dependent on the reference used for deconvolution. Thus, BayesPrism recommends that deconvolution results are compared across samples deconvolved using the same reference (for example, comparing melanocyte fraction in sample X vs sample Y), which is what we have done in the updated Figure 3. Also, while a deconvolution result of 0.02 may seem low, we assigned an average of 1.8 million reads per sample to GENCODE annotated genes. Thus, representing 0.02 of the assigned reads is the equivalent of ~36,000 reads for such an annotation, and this is the standard method of cell proportion reporting for the field.

Finally, we have added all these data to our SOMA data portal and online resource, enabling other researchers and clinicians to mine and characterize these data. Any given gene can be plotted by users, to visualize data from cfRNA, single-cell expression, bulk RNA expression, and other metadata (example in **Reviewer Figure 1**). We used data from the JAXA Cell-Free Epigenome (CFE) Study to examine some of our I4-enriched cell types in cfRNA, focusing on cell-type specific genes like *PMEL* (premelanosome protein specific to melanocytes).

Reviewer Figure 1: Plot of Cell-free RNA (cfRNA) data from the JAXA Study. Any gene can be plotted in the SOMA data portal, to enable easy comparison across marker genes or signatures. Shown here is the mean expression and the standard error of the mean (SEM, y-axis), as well as the variance across each timepoint (x-axis). Data from: https://soma.weill.cornell.edu/apps/SOMA_Browser/.

We observed a similar increase in the postflight samples for the CFE Study, which matches the melanocyte data from cfRNA and spatial transcriptomics data shown in the updated Figure 3. As we found consistent changes across multiple studies, missions, and data types, we are more confident to include these changes in the paper.

After a few attempts at understanding Figure 4, we remain almost confused as to what is attempting to be demonstrated, or the ways that the data has been analyzed. The second half of this figure also seems to fundamentally confuse statistical power and magnitude of effect. No sense of the strength of the pathway enrichments is given, nor any sense of if the uniqueness of the pathways in the cell types might be driven by changes to the power of measuring different differentials (which of course scales with the number of cells etc.). The use of the number of differential genes from the RNA-seq data as a metric is also fraught, as the power to call differentials is clearly affected by sample to sample variability (in collection quality, cell number, sequencing depth, etc.). If the authors want to use the absolute number of differentials to quantify the severity of a response and its return (or not) to baseline, they first need to convincingly demonstrate that these power considerations are not an issue. For example, if you took two multiome samples from the same astronaut on the ground and processed them on back to back days, would you see the same number of differentials? Fewer? The analysis of these data sets is not well described or controlled with null distributions. Thus the conclusion from analyzing longitudinal snRNA-seq across multiple individuals and conditions appears to be: there are differentially expressed genes. There seems to be no attempt to actually describe the molecular basis for the physiological changes resulting from spaceflight. Furthermore, a similar concern applies to the analysis of the cell free nucleic acids (is it biologically relevant that snoRNAs decrease in abundance in cfRNA post-flight?), m6a (what is the significance of the trajectories of methylated sites, as in, are there meaningful biological differences between Set M-I through M-III?), and other modalities. Overall, this paper continues to read as a laundry

list of assays that generates troves of data that remain analyzed in a way that is very hard to interpret or understand in the context of 1) reproducibility and rigor or 2) biological mechanism.

Thank you for the constructive comments, which guided new experiments for our revision. As requested by the reviewer, we now include a set of single-cell PBMC negative controls, wherein healthy PBMC donor samples were processed, in triplicate, across back to back days, using the same type of single-cell, multi-ome kits and protocol used for the I4 astronauts' samples. We then tested across all pairwise comparison groups (intra-subject variance, subject-to-subject comparison, variation within the technical replicates, random sampling), and individual-specific comparison to generate null distributions.

These negative control samples were acquired from cryovials of PBMCs from a healthy male donor (AllCells, age 22), and stored in vapor phase liquid nitrogen cryotanks. Cryovials were thawed and processed with the 10xGenomics multiome kits and sequenced on the Illumina NovaSeq6000. We down-sampled all the reads from the control cells to match the I4 samples, and used the same number of cells by type as observed for the I4 data set, to calculate differentially expressed genes (DEGs). We also worked with three scientists at 10X Genomics (Joe Shuga, Stephen Williams, Corey Nemec) to ensure best practices and methods for the single-cell multiome kits, and their team also reviewed the data for this paper.

Reassuringly, the number of DEGs in the single-cell I4 data is higher than the matched negative controls and all permutation groups (**new Extended Data Figure 5a-b**). The DEGs from the replicate PBMC donors ("negative controls") are different both in kind and degree, such that the genes showing differential show <1% overlap with the DEGs from the I4 comparisons and generally not as large of a magnitude. Specifically, the negative controls show a 0.08% overlap with any down-regulated DEG and 3.41% overlap with any up-regulated DEGs from any of the I4 crew DEGs from any comparison.

The new heatmap in **Extended Data Figure 5b** represents the clear separation of I4 DEGs with the negative control and I4 individual variation (subject-to-subject comparison per each timepoint) groups. In addition, DEGs are conserved across individual samples in FP1 and RP3, but not in the negative control and I4 inter-individual variation, which represent spaceflight signatures. Finally, we visualized the log₂FC of the I4 FP1 PBMC DEGs in all groups to show the DEG profiles are unique in FP1 and RP3 (**Extended Data Fig 5c**). Overall, with downsampling and negative control groups now added, we feel appropriate context is now presented in the single-cell DEG analysis of the I4 crew. We have also included the upper-bound of negative control DEGs, from any cell type, to highlight the expected noise in single-cell multiome sample processing in the updated main **Figure 4a**.

Figure 5 is likewise confusing. It would appear that the message from the metagenomics is that "things are variable." Then the figure analyzes variability in all data sets, and sees disparate results. Given the immense number of hypotheses tested, it appears that there is not a uniform

means to control hypothesis testing across data sets. When just reporting “significant” findings by testing many data sets comprehensively for all comparisons, the right way to control would really be to have a uniform correction for all tests performed for all assays (otherwise p-values will remain inflated). It is simply unclear what associations here are to be believed given this fundamental issue (quite apart from the QC issues that remain unaddressed).

According to the reviewer’s suggestions, we have made two main changes:

1- Instead of using simple coefficient of variation (CV) calculations, we are now reporting the abundance standardized CV scores (*Reviewer Figure 2*).

Reviewer Figure 2: Mean/CV relationships and abundance-standardized CVs. The regular coefficient of variation calculation (left) and abundance standardized CVs (right) are plotted (dots). Red lines are the best fit lines using loess regression approximation for the data sets.

Due to the nature of the sequencing methods, it is well-known that there is a mean-variance and mean-CV relationship, where the least abundant analytes have higher CV compared to higher abundance analytes. To correct for this we are comparing the CV of each analyzed point to similarly abundant other molecules in the assay and checking the deviation from this distribution. The reference sets are calculated across all samples per assay, and it is used to normalize the CVs for each time point separately. This allows us to compare the CV across analytes that have different baseline abundances and corrects for the mean-CV relationship.

Using these abundance standardized CV values, we also ran permutation tests (n=1,000 permutations) for the difference in the post-flight versus pre-flight CV shifts, to get a null distribution of the data, independent of any assumptions about the data distribution. These data are now plotted across all data modalities in the new main Figure 5. The focus on variability for the last figure is meant to highlight the diverse response patterns that can be observed, even under similar conditions between individuals, as well as to show that the CV range is distinct for each modality of biology (e.g. metabolomics vs. RNA-seq). This approach is particularly relevant in this early, exploratory phase of omics studies in spaceflight, where an understanding of the range of variation is mostly unknown, and these data can serve as a baseline for future crews and missions.

The authors make highly speculative statements in the discussion, including: “Hence, the systematic analysis of the molecular and cellular changes observed post-flight afford us with a unique opportunity to capture naturally occurring health repairing mechanisms which can be used for therapeutic target discovery.” How might this happen?

We have removed the more speculative and forward-looking statements in the Discussion, including this statement about target discovery.

In the discussion, the authors state:

“Of note, these data are the first-ever joint single nuclei chromatin profiles (snATAC-seq) for astronauts, and they also leverage new methods that can track gene expression and epigenetic changes within the same cells. This single-cell, dual-measurement assay provides mechanistic insights into the molecular changes and regulatory response to spaceflight, and the data revealed distinct levels of stress and adaptation by each cell type.” However, snATAC-seq data is basically not discussed in this manuscript (other than to mention it was generated). Using these data to validate some of the gene expression changes observed (via concomitant changes in local chromatin accessibility) would be welcome, but it would appear that these data sets have effectively been eliminated from the main flow of the paper. It is also entirely unclear how “this single-cell, dual-measurement assay provides mechanistic insights into the molecular changes and regulatory response to spaceflight, and the data revealed distinct levels of stress and adaptation by each cell type.” This is not consistent with the commonly-understood definition of “mechanism.”

We have updated the text and now include a detailed integration of the chromatin, expression, and TFBS analysis in the main text and in Extended Data Figure 6. We have also extended the access and visualization of these data relative to other missions and control data sets, which are now in the SOMA data portal (see below, Reviewer Figure 4).

Globally, there is a uniform lack of 1) communicating quality control measurements of any of the data sets, and 2) any discussion or analysis relating to attempts to understand or use internal controls for batch correction – crucial concerns for a large-scale data set with data potentially collected in very different conditions. Furthermore, it seems that the authors claim multiple times that changes are to be believed because “... QC metrics for the samples were the same” or “However, they increased in B cells, CD4+ T cells, CD8+ T cells, and CD14+ monocytes (Fig 4a), even though QC metrics were similar across timepoints.” However no citations appear to any figures or supplements or anything to support these claims that the data is high quality and uniform and devoid of concerns regarding batch effects. We believe this lack of support for crucial data quality claims does not rise to the level of rigor required for a Nature paper. Finally, nearly every reviewer raised the point that this study made no attempt to control for the non-spaceflight-intrinsic stressors of this trip (e.g. confinement and diet, compared to microgravity and radiation exposure). At the very least, the authors should tamp down their attribution

of these changes to specific causes in light of this crucial missing control. However, this lack of control makes it effectively impossible to really understand what these observations mean in the context of spaceflight specifically.

We acknowledge the reviewer's concern and the need for clarity on rigor for hypothesis testing. We have ensured all p-values are corrected for multiple hypothesis testing (whenever possible), performed additional single-cell experiments on controls, added permutation analyses, and include more details on the QC steps in a dedicated paper, now cited as Overbey et al, Collection of Biospecimens from the Inspiration4 Mission Establishes the Standards for the Space Omics and Medical Atlas (SOMA), Nature Communications, in press, Reference #12). We have revised our manuscript to make these points clearer, particularly emphasizing that our analysis was focused on consistent, cross-mission molecular differences.

In short, the paper continues to lack coherent biological insight or observations of physiological changes that are convincingly supported by multiple data modalities or in other data sets. The authors make many dramatic claims about the need for multiomic profiling of astronauts (e.g. “single-cell sequencing must be implemented more regularly in spaceflight”) and their “unprecedented view of the multi-system omics changes before, during, and after spaceflight.” Yet we are left wondering about what, precisely, has been learned about the physiological response to spaceflight from the authors’ 10s of billions of sequencing reads, and are also left without a clear picture of the quality of these data sets that are, it would appear, the main deliverable of this work.

We believe these clarifications, as well as updates to the SOMA data portal (below), will help mollify concerns regarding the interpretation of our findings and the methodological approach. Moreover, we have compiled all of the human missions’ data now from various missions and complemented this I4 resource with additional murine and human cell line data from NASA’s GeneLab into the SOMA data portal, which now spans >1,400 samples and >30 spaceflight missions.

We used the comments from Referee #5 and #6 to help us guide the updates to the SOMA data portal, including more missions, sample types, and data visualizations, which are extensive, and which we believe will be useful for researchers in the field. Please see our responses and updated previews of these resources and links in Reviewer Figures 3 and 4 (below), and also the updated data portals online.

Thank you.

Referee #6:

Remarks to the Author:

I appreciate the authors' efforts to respond to my and other referees' comments and to update their manuscript accordingly. I believe the revisions have largely and appropriately addressed my concerns and have improved the clarity and quality of the manuscript. Although I commend the authors for their impressive work, I would also like to raise a few remaining, albeit small, points:

Thank you for the positive response and appreciation of our work.

Major points:

- I was not able to access the data deposited at OSDR. For instance <https://osdr.nasa.gov/bio/repo/data/studies/OSD-569> showed the message "The study you requested is not available. Please contact the OSDR staff if you have further questions". This occurred across sample types and remained a problem despite multiple attempts on different dates.

Please note that the final NASA GeneLab links for the data will be live upon publication. However, NASA has provided a list of "reviewer preview links" for the data, code, and sample annotations, and we've provided them here for ease of access:

OSD Number Reviewer Private preview links:

1. OSD-569 (whole blood)
<https://osdr.nasa.gov/bio/repo/data/studies/OSD-569/preview/SQ3QgEzNMH5q1X19jKWJxVLF5Lx5RQtp>
2. OSD-570 (PBMC)
<https://osdr.nasa.gov/bio/repo/data/studies/OSD-570/preview/y0IXYJACBT2kLVOnsLESaPa1NrI3IKyh>
3. OSD-571 (blood plasma)
<https://osdr.nasa.gov/bio/repo/data/studies/OSD-571/preview/Wr-IXUj2sAQ4dOp4UcAuajlGfMcZAX0E>
4. OSD-572 (skin, oral and nasal swab)
<https://osdr.nasa.gov/bio/repo/data/studies/OSD-572/preview/CuhxWdYAPRX4QJRLd74yzdXjon2OxrXm>
5. OSD-573 (dragon capsule)

<https://osdr.nasa.gov/bio/repo/data/studies/OSD-573/preview/Ze2eyzuoTvJ7CI0887nweUeBe9GqYTZN>

6. [6OSD-574 \(skin biopsy/swab\)](https://osdr.nasa.gov/bio/repo/data/studies/OSD-574/preview/cRBHSdY8b-ekuGYCFZd8bexhWfGIdM9i)
<https://osdr.nasa.gov/bio/repo/data/studies/OSD-574/preview/cRBHSdY8b-ekuGYCFZd8bexhWfGIdM9i>
7. [OSD-575 \(blood serum\)](https://osdr.nasa.gov/bio/repo/data/studies/OSD-575/preview/blm-fESutIV3vdJofJPygFXGW6omqd8e)
<https://osdr.nasa.gov/bio/repo/data/studies/OSD-575/preview/blm-fESutIV3vdJofJPygFXGW6omqd8e>
8. [OSD-630 \(stool\)](https://osdr.nasa.gov/bio/repo/data/studies/OSD-630/preview/BilzXsJxraKrE9sYDCsOh-MLL3yPppSC)
<https://osdr.nasa.gov/bio/repo/data/studies/OSD-630/preview/BilzXsJxraKrE9sYDCsOh-MLL3yPppSC>
9. [OSD-656 \(urine\)](https://osdr.nasa.gov/bio/repo/data/studies/OSD-656/preview/TO0a-2qmUg4xGuGnPidm40r4yt1HYIGq)
<https://osdr.nasa.gov/bio/repo/data/studies/OSD-656/preview/TO0a-2qmUg4xGuGnPidm40r4yt1HYIGq>

- I commend the authors for the improved manuscript and report of statistical significance (or lack thereof). I did however note a few instances (lines 272, 343, 468) that may benefit from some additional nuancing, for instance by mentioning that the absence of significance “suggests” a return to baseline. Although my concerns regarding statistical reporting have been addressed throughout the main text and thanks to additional supplementary methods, the inclusion of a short paragraph to discuss the interpretation of lack of significance based on a small cohort (n=4) would be welcome; in particular to mention that lack of significance can suggest a return to baseline, but that impacts below a given effect-size cannot be detected with the current dataset. The minimal detectable effect size could even be modeled for some of the variables tested under a hypothesized probability distribution based on sample size and variance.

We have added several points in the Discussion and Future Directions sections about effect sizes and limitations of the study, as well as the need to leverage additional data sets and methods in future studies. Also, we now include replicated negative control single-cell data from donor PBMCs and permutation testing, to generate the null distributions for the DEG and DAR analyses (see comments to Reviewer #5, above), which helps to estimate noise of the DEGs. Fortunately, these results confirmed that our cellular changes associated with spaceflight are distinct both in their loci and magnitude.

Minor points:

- The web portal for the available biobanked samples (<https://cambank.weill.cornell.edu/>) is a valuable addition. However, the web interface could include some additional documentation along with a possibility to browse the list of samples.

We have updated the CAMbank portal to include more information about the samples and what is available. Currently, there are >3,000 samples annotated and available for query or request by other researchers. A preview of the web page is included here (**Reviewer Figure 3**):

Sample Type	Timepoint									
	Total	L-92	L-44	L-3	R+1	R+45	R+82	R+194	R+668	
CPT Plasma	194	50	32	7	8	35	46	16	0	
cfDNA BCT Plasma	73	7	15	5	16	17	3	10	0	
SST Serum	103	17	14	17	9	26	12	8	0	
Vacutainer RBC Pellet	425	0	109	77	116	48	56	0	19	
PAXgene RNA	507	65	58	71	67	101	61	84	0	
Saliva DNA	229	8	35	28	55	61	42	0	0	
Saliva RNA	327	37	30	64	84	59	53	0	0	
CPT PBMCs	218	0	44	38	44	24	44	12	12	
Crude Urine	533	97	79	79	73	95	110	0	0	
Conditioning Buffer Urine	424	30	28	80	77	100	109	0	0	
Streck Urine	250	0	0	0	0	0	0	0	250	
K2EDTA Plasma	30	0	0	0	0	0	0	0	30	
Stool DNA	167	49	54	0	0	33	31	0	0	
Stool RNA	105	20	32	0	0	29	24	0	0	
Oral/Nasal/Skin Swab DNA	280	40	40	40	40	40	40	40	0	
Oral/Nasal/Skin Swab RNA	280	40	40	40	40	40	40	40	0	

Reviewer Figure 3: CAMbank sample portal. The landing page for CAMbank shows all the sample types banked by timepoint (top) and sample type (left), including blood tubes, extracted nucleic acids, stool, PBMCs, swabs, and other aliquots from the mission. This includes the number of samples from each timepoint before launch (e.g. L-92, L-44, for 92 and 44 days before launch) as well as post return (e.g. R+1 and R+45).

- Similarly, the addition of the SOMA (https://soma.weill.cornell.edu/apps/SOMA_Browser/) and I4 Microbiome (https://soma.weill.cornell.edu/apps/I4_Microbiome/) browsers in addition to the initial I4 Multi-ome browser (https://soma.weill.cornell.edu/apps/I4_Multiome/) provide interactive access to important information. However, inclusion of additional documentation on those web portals (in particular for the multi-ome) would be needed to fully enable the community to benefit from those resources.

Thank you for the suggestions, and we have now added several features to the SOMA data portal. First, on the left side browser, we have posted additional code and methodological details on a tab labeled “Information/Code/Methods.” This includes links to the protocols and methods papers, the code base on GitHub, and instructions on the sources of the data and related papers. Second, we have added new plotting functions that highlight trends for analytes across missions, featured in the “Differential Expression Comparisons” tab. Specifically, this plotting function includes mapping across each of the missions on the SOMA portal (NASA Twins, NASA Cytokine, I4 multi-omics, bulk and single-cell data, etc.) as either dot plots or arrow plots to show statistical significance and directionality of genes’ expression differences (**Reviewer Figure 4**). Third, we greatly expanded the resource by leveraging additional studies from NASA and GeneLab, wherein we mapped the orthologous mouse genes to the human data, and added 46 murine spaceflight studies, spanning >1,400 samples and >300 comparisons from the NASA’s Open Science Data repository (OSDR) and GeneLab databases. Fourth, we also created a plot for the mouse gene expression data from each specific tissue (liver, heart, kidney, liver, lung, muscle, bone, spleen, skin, colon, and other tissues), plotted in conjunction with any human ortholog. Fifth, we have added an additional documentation page about the single cell Multiome data browser and linked to its publication to help users better navigate the data.

a Data Portals

SOMA Browser

- Gene Expression (bulk, cell-type specific, scRNA)
- Chromatin Accessibility (cell-type specific)
- Protein Abundance (plasma, EVPS)
- Metabolite Abundance (plasma)
- Cytokine/Immune Response
- Twins Study Data
- JAXA Data
- NASA OSDR/Genelab Data (resident, spaceflight, controls)

Single-Cell Browser

- Single-Cell Expression Clustering (UMAP)
- Single-Cell Chromatin Accessibility Clustering (UMAP)
- Transcription Factor Motif Activity (UMAP)

Microbial Browser

- Metagenomic Abundance (skin location, capsule)
- Metatranscriptomic Abundance (skin location, capsule)

b

c

d

e

f

Reviewer Figure 4. SOMA Data Portal (now Extended Data Figure 8). Three different web portals (a) were expanded and updated: SOMA Browser, Single-Cell Browser, and Microbial Browser. The SOMA Browser includes (b) gene expression and protein abundance measurements in line chart and volcano plot formats, (c) log fold change calculations, and (d) comparison of DEGs across different contrasts, assays, studies, and organisms. (e) The Single-Cell Browser enables visualization of cell type specific information, including gene co-expression and ATAC-seq region peak visualization. (f) The Microbial Browser includes microbial abundances across timepoints from different annotation databases for metagenomic and metatranscriptomic datasets.

These five large improvements to the SOMA data portal enable easy comparisons between SpaceX, NASA, and JAXA missions, multiple species (human and mouse), and data modalities (various omics methods and metrics). We have also added contact information on the site for anyone who wants to request improvements, corrections, links to other resources, or other additions. These are also detailed in the updated manuscript in Extended Data Figure 8, SOMA Web Portals.

To our knowledge, this updated SOMA data set and online resource represents the largest single collection of human and murine spaceflight data ever compiled. Such resources can help address the key issues of statistical power and enable cross-mission comparisons for spaceflight studies, as well as serve as a needed anchor for continued sharing of data and hypothesis generation in the field (see below).

We are grateful for the opportunity to clarify our methods and results, and we hope that these revisions, new experimental data, and updated analyses will sufficiently address reviewer concerns.

Reviewer Reports on the Second Revision:

Referee #2 (Remarks to the Author):

The authors have largely addressed my remaining comments, and the analysis is now more clearly presented. I have one comment related to a revised figure, and a comment related to significance testing.

- Figure 5/ CV analysis: The authors have included Ultima Genomics RNA-Seq in addition to direct RNA-Seq. However, the data does not seem to agree very well, why is that the case? This discrepancy is not explained in the text. One possible explanation is that the variation between replicates is very high, and that additional replicates are required to obtain results that are reproducible across technologies. An alternative explanation might be that differences in sample extraction, library preparation, or sequencing technology lead to these observed differences. Both explanations would still suggest limitations in interpreting these results in terms of general gene expression changes. I would suggest to add a statement about the observed differences and limitations in interpreting results to the corresponding results section.

- Significance tests and error bars are described in the figure legend and generally appropriate for the analysis. As some tests are performed on thousands of measurements (e.g. RNA-Seq expression estimates), these tests have very high sensitivity to detect even small differences with very low p-values. However, these tests do not reflect the limitations due to small numbers of replicates that are used to generate these measurements. This is particularly relevant for Figure 5, where many p-values are $<2.2 \times 10^{-16}$ even for small differences in population means. However, one outlier sample can already lead to such a p-value by increasing the observed variation for all genes. This limitation is not reflected in the significance tests. I would suggest to remove p-values for Figure 5 a-f to avoid misinterpreting the confidence for this experiment.

- The permutation test might possibly address the limitation described above. However, I currently find it difficult to understand the test or how it was performed. A permutation test suggests that a confidence level or p-value is estimated, but only a difference is presented in Figure 5g-l, therefore I also do not understand how this is a "test". I recommend to clarify this in the main text and methods, and possibly adjust the figures.

Referee #5 (Remarks to the Author):

We have now read the revised manuscript from Overbey et al. We thank the authors for taking some of our concerns seriously and for doing further analyses and experiments in order to address them. The manuscript has improved somewhat, in particular the writing is clearer and tighter, and the presentation of some of the figures (notably 2) is improved. However, we still have two largely independent concerns, the first related to lingering data quality and analysis issues, and the second related to interpretation.

Data quality/analysis:

Many of our data quality concerns have been addressed. The mixed effects model for the telomere length is convincing, and the fact that the cell-type proportions deconvolution analysis from the cfRNA data (in re: melanocytes changing) is supported by the spatial transcriptomics is reassuring. We appreciate the authors clarifying these and other points. However, a few concerns remain. For one, we thank the authors for doing multiple hypothesis correction on the cytokine panel. However, it appears that the 3 common cytokines from the earlier NASA study aren't actually significant post correction (and while this is mentioned in the text, it doesn't appear in the figure, which still lists the nominal p-values for the 14 astronauts). We recognize that it is hard to gather significance from low N data and that the comparison to the previous missions does lend Bayesian support to the fact that these signals are real, but it would seem that the figure is somewhat misleading given the text.

Secondly, we thank the authors for doing the permutation testing we asked for in figure 5. However, we are a little bit confused about how these results are presented in the new figure. Specifically, the red line(s) represent the standard error of the mean of the difference in CV pre- and post-flight from the bootstrapped replicates? Of course this is miniscule, since it's the standard deviation of the empirical null divided by the sqrt(number of genes) (which is huge). What is actually informative is the variance of the empirical null itself (which is, I believe, plotted in orange). However, looking at the 95% interval of the empirical null compared to the data distribution, we don't see a huge enrichment of genes whose CV differences exceed what would be expected by chance. It seems like there are ~5% of genes that fall above and below the orange lines, which is exactly the expectation under the null. The authors should include some additional sanity check that these genes are indeed unexpected, perhaps by plotting a Q-Q plot of the empirical vs expected p-values (for example). Otherwise, the results in 5g-l are unremarkable.

Finally, the replicate multiome experiments to provide the null expectation in figure 4a is a compelling demonstration of the sensitivity of the assay (although we are still a bit dubious of using the absolute number of DEGs as a metric for the severity of the response, as we have mentioned before). However, our concern about the conflation of statistical significance and effect size in the pathway analysis remains unanswered. Furthermore, we also asked about whether the pathway uniqueness analysis is confounded by the ability to call DEGs on a per-cell type basis. Almost certainly the cell number for each of the cell types affects the power to call DEGs on a per-cell type basis, and therefore the pathway uniqueness analysis (and also the UpSet plot in d) are probably confounded by this. We don't know how to interpret the number and uniqueness arguments across cell types without understanding how to control for the different ability to call DEGs between cell types (but still within one experiment).

Data interpretation:

Nearly all the concerns raised in our last review had to do simultaneously with data quality issues and data interpretation. While the authors addressed some of the former, they uniformly ignored the latter. Specifically, for this paper to transcend mere “data dumping”, the immense trove of data they have collected need to be mined beyond the most facile analyses. This applies to nearly every figure. For example, in both of our prior reviews, we wondered if there was biological significance to the fact that there are 3 profiles of m6A patterns (Sets M-I through M-III). The authors did not provide an answer. Similarly, the authors continue to talk about how snoRNAs were elevated in cfRNA pre-flight compared to post-flight without any possible indication of why this might be the case, or what its significance might be. (This is in comparison to “all protein-coding genes”, which seem to go up.) Furthermore, the immense enrichment of these short RNAs raises substantial alarm bells with respect to confounding of sample preparation or other technical artifacts, and a lack of any biological context regarding these observations underlines these concerns.

The snATAC-seq analysis is similarly limited. It is relatively unsurprising that the accessibility at genes that increase in expression will also concomitantly go up (e.g. Wu 1980). Does that imply that “single-cell sequencing must be performed on all future space flights”? And many of the motifs that change by chromVAR are in the Fos/Jun family, which nearly always change in response to stress (often caused by technical covariation and sample handling differences) and are therefore often ignored in downstream analyses.

Overall, the contextualization of the immediate observations (some cytokines go up, some cell type proportions change, gene expression generally returns to baseline after spaceflight, etc.) into biologically-actionable insights remains lacking. To our ear, there remains a disconnect between the lack of biological findings and interpretations with the breathless tone in which the resource is presented in the text.

Minor concerns:

There is a dangling clause at the end of the sentence that spans lines 197-202.

We don't think “Permutation Testing” is likely an appropriate header for the figures in 5G-L.

If the results from a number of analyses (e.g. figure 5g-l) all look similar, only one needs to go in the main text and the rest can be relegated to the supplement, unless there are crucial differences between the results from the different panels that merit their inclusion.

Referee #6 (Remarks to the Author):

I thank the authors for their additional clarifications and adjustments, which have appropriately addressed my remaining concerns, and congratulate them on this impressive piece of work.

Author Rebuttals to Second Revision:

We have responded to each reviewer point below, indicating our responses in blue below to each comment. We have further updated the SOMA data portal and give an example (below) for mapping various pathways, which underscores the utility of this paper to help contextualize and validate questions that might arise when analyzing spaceflight omics data.

We are very grateful for the reviewers' time and rapid reviews. We believe the reviewers will find the updated manuscript and resource much improved and hopefully suitable for publication, enabling others in the field to use these data, tools, and methods.

Referees' comments:

Referee #2 (Remarks to the Author):

The authors have largely addressed my remaining comments, and the analysis is now more clearly presented.

Thank you for the positive response.

I have one comment related to a revised figure, and a comment related to significance testing.

- Figure 5/ CV analysis: The authors have included Ultima Genomics RNA-Seq in addition to direct RNA-Seq. However, the data does not seem to agree very well, why is that the case? This discrepancy is not explained in the text. One possible explanation is that the variation between replicates is very high, and that additional replicates are required to obtain results that are reproducible across technologies. An alternative explanation might be that differences in sample extraction, library preparation, or sequencing technology lead to these observed differences. Both explanations would still suggest limitations in interpreting these results in terms of general gene expression changes. I would suggest to add a statement about the observed differences and limitations in interpreting results to the corresponding results section.

We have added a new section in the Discussion to highlight that differences that can arise from different NGS, processing, and profiling methods, and also we have ensured that the paper does not overstate the results, specifically in this new section:

“However, this study is not without limitations. Although cell-specific comparisons could be made between the I4 mission and prior studies, the assays and collection protocols were slightly different (e.g. column purification vs. droplet-sorted cells), and comparisons will be imperfect. Also, while the same tube types and methods were employed whenever possible, cross-mission

comparisons will inevitably include noise from various other types of technical variation, including batches of library preparation and extraction kits, slightly different collection intervals (L-3 vs. L-10), and/or different sequencing/profiling technologies (e.g. Illumina vs. Ultima or LC-MS/MS vs. NULISAseq).”

We also ran an analysis comparing the RNA-seq quantification metrics. The largest difference between the RNA-seq methods is sequencing depth, which was >400M reads for the Ultima data and a mean 1.9M long reads for ONT, and we now refer to this in the text. When comparing the nanopore gene expression profiles to the Ultima data, we observed an average Pearson correlation of 0.65 with the UG100 platform, and as expected, the higher expressing genes showed the largest correlation (**Reviewer Figure 1**) ($R^2 > 0.56-0.71$ across all samples).

Reviewer Figure 1. Cross-platform correlation between gene expression profiles. The RNA-seq from R+1 samples produced by either Nanopore direct-RNA Promethion or Ultima Genomics UG100 platform are plotted. Quantification of normalized DESeq2 data for each crew member is shown, with Ultima counts (x-axis) compared to the ONT data (y-axis) and the R^2 values shown at the top of each plot.

This figure is also now part of a dedicated full article about the long-read ONT data from the Inspiration4 mission (Grigorev *et al*, in press, 2024), which also includes more discussion about the differences between platforms and isoform detection.

- Significance tests and error bars are described in the figure legend and generally appropriate for the analysis. As some tests are performed on thousands of measurements (e.g. RNA-Seq expression estimates), these tests have very high sensitivity to detect even small differences with very low p-values. However, these tests do not reflect the limitations due to small numbers of replicates that are used to generate these measurements. This is particularly relevant for Figure 5, where many p-values are $<2.2e-16$ even for small differences in population means. However, one outlier sample can already lead to such a p-value by increasing the observed variation for all genes. This limitation is not reflected in the significance tests. I would suggest to remove p-values for Figure 5 a-f to avoid misinterpreting the confidence for this experiment.

As requested, we have now updated this figure and removed the p-values. We now refer to them in the legend for reader clarity.

- The permutation test might possibly address the limitation described above. However, I currently find it difficult to understand the test or how it was performed. A permutation test suggests that a confidence level or p-value is estimated, but only a difference is presented in Figure 5g-l, therefore I also do not understand how this is a "test". I recommend to clarify this in the main text and methods, and possibly adjust the figures.

We have now updated the permutation figure and description in the methods and text to clarify. Thank you.

Referee #5 (Remarks to the Author):

We have now read the revised manuscript from Overbey et al. We thank the authors for taking some of our concerns seriously and for doing further analyses and experiments in order to address them. The manuscript has improved somewhat, in particular the writing is clearer and tighter, and the presentation of some of the figures (notably 2) is improved. However, we still have two largely independent concerns, the first related to lingering data quality and analysis issues, and the second related to interpretation.

Data quality/analysis:

Many of our data quality concerns have been addressed. The mixed effects model for the telomere length is convincing, and the fact that the cell-type proportions deconvolution analysis

from the cfRNA data (in re: melanocytes changing) is supported by the spatial transcriptomics is reassuring. We appreciate the authors clarifying these and other points.

Thank you for the positive response.

However, a few concerns remain. For one, we thank the authors for doing multiple hypothesis correction on the cytokine panel. However, it appears that the 3 common cytokines from the earlier NASA study aren't actually significant post correction (and while this is mentioned in the text, it doesn't appear in the figure, which still lists the nominal p-values for the I4 astronauts). We recognize that it is hard to gather significance from low N data and that the comparison to the previous missions does lend Bayesian support to the fact that these signals are real, but it would seem that the figure is somewhat misleading given the text.

We have now put an asterisk next to the p-values, which indicates if the results are still significant with multiple-testing correction, and we added a note in the figure description. Since these cytokines were already identified in a previous study (NASA Twins), we had a prior hypothesis to test on these samples, and we were glad to see the trend was still present. Indeed, this would be significant by Bayesian methods, and we also note that the same directional response, yet with a smaller effect size, matches the shorter mission duration.

Secondly, we thank the authors for doing the permutation testing we asked for in figure 5. However, we are a little bit confused about how these results are presented in the new figure. Specifically, the red line(s) represent the standard error of the mean of the difference in CV pre- and post-flight from the bootstrapped replicates? Of course this is miniscule, since it's the standard deviation of the empirical null divided by the sqrt(number of genes) (which is huge). What is actually informative is the variance of the empirical null itself (which is, I believe, plotted in orange). However, looking at the 95% interval of the empirical null compared to the data distribution, we don't see a huge enrichment of genes whose CV differences exceed what would be expected by chance. It seems like there are ~5% of genes that fall above and below the orange lines, which is exactly the expectation under the null. The authors should include some additional sanity check that these genes are indeed unexpected, perhaps by plotting a Q-Q plot of the empirical vs expected p-values (for example). Otherwise, the results in 5g-1 are unremarkable.

We thank the reviewer for this point. To answer the significance question, we have increased the number of permutations to 10,000, and checked which genes/analytes in these permutations have a value as extreme as the observed value (either CV in a timepoint, or postflight versus preflight difference of CV after the permutations). For the whole blood RNAseq data, we have 34 genes in preflight, 4 genes in postflight, 5 genes in recovery period, and 7 genes in postflight vs preflight difference that are the most extreme values in the permutations (i.e. 0 random permutations that

reach the level with 10K permutations), and there are many more with $p=1e-4$ (1/10,000 permutations) or similarly low values.

These numbers exceed what would be expected by random 1/10,000 chance in a dataset of ~16K expressed genes (cumulative probability of ≥ 5 successes $P(X \geq 5)$ in a 0.0001 probability task repeated 16,000 times with a binomial distribution is 0.01764, $P(X \geq 4)$ is 0.078, $P(X \geq 34)$ is effectively zero). The primary aim of this figure was to characterize and demonstrate the inter-crew variability within our dataset. For this purpose, we also focused on whether the CV and changes in variability show interpretable patterns that may give more insight into the biology, and to this end we have added pathway analysis for these rankings. To address the null-distribution question raised by the reviewer, we are using the results from the permuted sample labels, and ranking the genes based on how much they deviate from the 10K random permutations across timepoints. In addition, the GSEA pathway enrichment adds additional permutation testing across the genes. Positively enriched pathways would mean that the genes in that pathway have higher CV/ Δ CV than the random distributions based on their ranking, and vice versa. These results are also added to the Extended Data Tables.

We believe the current panels add more context and detail to the measures of inter-crew variability, and the expanded tables also show the details for each gene/analyte for downstream researchers. Given the consistent results across all analytes, we have also updated the lower part of Figure 5 to highlight the most recurrent pathways across missions, cell types, and assay types, which we think is the most relevant result.

Finally, the replicate multiome experiments to provide the null expectation in figure 4a is a compelling demonstration of the sensitivity of the assay (although we are still a bit dubious of using the absolute number of DEGs as a metric for the severity of the response, as we have mentioned before).

Thank you, and we agree with the reviewer on the compelling (and reassuring) result. Also, we now also include a more detailed analysis of the pathways impacted across all cell types and include metrics on the cell counts for the Data Tables, and updated Figure 4d and Figure 5. Also, we re-analyzed the scRNA-seq data with another method (DESeq2 on pseudobulk cell populations), which is known to be more robust to intra-sample correlation and varied cell counts (**Extended Data Figure 5**). Indeed, this method was shown to be one of the best methods for detecting DEGs from scRNA-seq data (<https://www.nature.com/articles/nmeth.4612>) in a recent benchmarking paper (lower false positives). We have now added these new analyses into the updated Extended Data Figure 5 (specifically 5b), and updated Figure 4, which have confirmed the trend across cell types and mission timelines.

However, our concern about the conflation of statistical significance and effect size in the pathway analysis remains unanswered. Furthermore, we also asked about whether the pathway uniqueness analysis is confounded by the ability to call DEGs on a per-cell type basis. Almost certainly the cell number for each of the cell types affects the power to call DEGs on a per-cell type basis, and therefore the pathway uniqueness analysis (and also the UpSet plot in d) are probably confounded by this. We don't know how to interpret the number and uniqueness arguments across cell types without understanding how to control for the different ability to call DEGs between cell types (but still within one experiment).

Thank you for these questions, and we have now reconfigured the prior panels into a new figure (**Figure 4d**) to enable greater clarity, focusing on *both* the enrichment ratio as well as the statistical significance. We also point the reviewer to the updated methods section for the scRNA-seq analysis, which used the non-parametric Wilcoxon rank sum test for DEGs, wherein comparisons had an average of >500 cells per timepoint. This test uses the proportions of cells expressing a given gene, and can account for data sets with distinct numbers of observations. Moreover, some of the pathways (e.g. IL-17 signaling) showed convergence across all cell types, whereas others are very cell-type specific.

Data interpretation:

Nearly all the concerns raised in our last review had to do simultaneously with data quality issues and data interpretation. While the authors addressed some of the former, they uniformly ignored the latter. Specifically, for this paper to transcend mere “data dumping”, the immense trove of data they have collected need to be mined beyond the most facile analyses. This applies to nearly every figure. For example, in both of our prior reviews, we wondered if there was biological significance to the fact that there are 3 profiles of m6A patterns (Sets M-I through M-III). The authors did not provide an answer.

For the m⁶A profiles, we have added a new summary of the pathways that show changes based on differential methylation and DEGs, with a focus on those specific to the recovery period. Specifically, we now state:

“The common pathways in all three M-I/II/III sets showed evidence of radiation and telomere response, including “TSAI response to radiation therapy” and “Wiemann telomere shortening.” However, the set of downregulated pathways after landing (recovery) were distinct, and notably included genes associated with breathing regulation (e.g. CO₂ and O₂ takeup and release by erythrocytes), which matches those loci with more inter-crew variation (below). These pathways suggest a potential relationship between expression and methylation in regulating these critical hematological functions upon return to Earth.”

We also point the reader to a companion paper for the SOMA package, which provides an in-depth analysis of the splicing, intron retention, and isoform validation of these long-read data (Grigorev et al., 2024, Reference #22).

Similarly, the authors continue to talk about how snoRNAs were elevated in cfRNA pre-flight compared to post-flight without any possible indication of why this might be the case, or what its significance might be. (This is in comparison to “all protein-coding genes”, which seem to go up.) Furthermore, the immense enrichment of these short RNAs raises substantial alarm bells with respect to confounding of sample preparation or other technical artifacts, and a lack of any biological context regarding these observations underlines these concerns.

We agree with the reviewer in the necessity to ensure any changes are not likely from sample processing or lab-related artifacts. To limit technical artifacts, all samples (including pre-flight and post-flight samples) were processed at the same time, by the same technician, and in the same lab (De Vlammick). To examine other evidence of snoRNA regulation, we analyzed the snRNA-seq Gene Ontology Pathways for snoRNA localization and snoRNA binding, and compared them to mRNA processing and DNA repair (**Reviewer Figure 2**). This analysis revealed that the pathways for snoRNAs show significant enrichment, across multiple cell types (e.g. CD16 monocytes, NK cells, and T-cells), and these enrichments are different in both direction and orientation when compared to mRNA processing or DNA repair.

Reviewer Figure 2. Pathway analysis of the I4 scRNA-seq data for snoRNA processing. Analysis of the single-cell RNA-seq (scRNAseq) data from the Inspiration4 mission shows the pathways (left) and the cell types (top), as well as a dot for any significant enrichment (p-value

as a function of size of the circle). The enrichment score is shown by the range of blue to red (right).

In addition, we have updated the SOMA data portal to enable a similar analysis for any set of genes or pathways by any user, annotated as the “Pathway Enrichment Comparison” tool (https://soma.weill.cornell.edu/apps/SOMA_Browser/). This method is what we used for our analysis of the snoRNA pathways here (and also in Figure 4), and such tools can be very useful in situations when any user (ourselves included) wants to check for an enrichment of genes or pathways from the I4 mission data. Also, we link the same pathways to the orthologous pathways annotated in the mouse GO database, which enables rapid characterization of human and murine spaceflight data.

Finally, we have shortened the section on snoRNAs to simply refer to generally observed dynamics in the RNA types (**Fig 3c**). As the reviewer notes, we do not have a proposed mechanism or specific biological meaning for these shifts, and the only additional evidence of snoRNA changes are from the pathway enrichments based on the scRNA-seq data (**Reviewer Figure 2**). Thus, we have shortened this section based on the reviewer comments, and we now close the RNA results section with this statement:

“Although further studies are needed to validate and delineate the potential mechanisms of these RNA dynamics (cfRNA, spatial, and RNA modifications), these pathways suggest a potential relationship between expression and methylation in regulating hematological and dermatological functions upon return to Earth.”

The snATAC-seq analysis is similarly limited. It is relatively unsurprising that the accessibility at genes that increase in expression will also concomitantly go up (e.g. Wu 1980). Does that imply that “single-cell sequencing must be performed on all future space flights”?

We agree that this is an expected result, and this plot was requested in a prior review of this paper, and given the prior reviewer request to keep this figure, we thought this would be a good quality control analysis for the Extended Data Figures. We have also updated the text in the Discussion to read:

“The differences between PBMC sub-populations also suggests that single-cell sequencing can be helpful for delineating unique cell type responses in future studies as well.”

And many of the motifs that change by chromVAR are in the Fos/Jun family, which nearly always change in response to stress (often caused by technical covariation and sample handling differences) and are therefore often ignored in downstream analyses.

We used the newly-generated 10x Multiome negative control data (triplicate collections across two different days from the same donors) to examine the question of motif background enrichment for the Fos/Jun family (**Reviewer Figure 3**).

Reviewer Figure 3: Absence of Fos/Jun motif enrichment in Negative Controls. We examined the enrichment profiles for the TF motifs in the snATAC-seq data, which showed no enrichment of the Fos/Jun motifs in the negative controls. Motif enrichment is plotted as $-\log_{10}(\text{adjusted } p\text{-value})$ on the y-axis and the change (delta) in z-score for motif accessibility (x-axis).

Fortunately, we did not observe any significant enrichment of the Fos/Jun motif in any cell type or timepoint comparison. This indicates that the spaceflight-associated motif changes are more

likely related to a response to the stress of the mission than sample handling or technical variation.

Nonetheless, we also added a section in the discussion to note that technical noise should always be considered when comparing between missions, protocols, and cell processing:

“However, this study is not without limitations. Although cell-specific comparisons could be made between the I4 mission and prior studies, the assays and collection protocols were slightly different (e.g. column purification vs. droplet-sorted cells), and comparisons will be imperfect. Also, while the same tube types and methods were employed whenever possible, cross-mission comparisons will inevitably include noise from various other types of technical variation, including batches of library preparation and extraction kits, slightly different collection intervals (L-3 vs. L-10), and/or different sequencing/profiling technologies (e.g. Illumina vs. Ultima or LC-MS/MS vs. NULISASeq).”

Overall, the contextualization of the immediate observations (some cytokines go up, some cell type proportions change, gene expression generally returns to baseline after spaceflight, etc.) into biologically-actionable insights remains lacking. To our ear, there remains a disconnect between the lack of biological findings and interpretations with the breathless tone in which the resource is presented in the text.

We have added a new paragraph at the end of the Discussion to help clarify the purpose of the Atlas and indeed, we do not intend for these data to be the basis of crew intervention or specific countermeasures - yet. They are the first step of mapping the response of the human body to spaceflight, and to prepare for longer missions, which is a long-term goal of space agencies and institutions, and we agree with them that such data are valuable. We have also added RNA-seq data from the MARROW study to the SOMA data portal and had our data reviewed by Canadian Space Agency collaborators from that MARROW study (especially Dr. Guy Trudel). Finally, to help make this clear for readers, we have added this section at the end of the Discussion, to ensure we do not overstate the significance of the results:

“Finally, since our comparisons to other NASA and JAXA datasets span, at most, 46 astronauts, these data are not sufficient to guide medical interventions or inference of mechanisms. As such, the SOMA data and resources should be viewed as preliminary molecular maps of the human body’s response to spaceflight.”

Minor concerns:

There is a dangling clause at the end of the sentence that spans lines 197-202.

Thank you for spotting this. We have removed the dangling clause.

We don't think "Permutation Testing" is likely an appropriate header for the figures in 5G-L. If the results from a number of analyses (e.g. figure 5g-l) all look similar, only one needs to go in the main text and the rest can be relegated to the supplement, unless there are crucial differences between the results from the different panels that merit their inclusion.

We have updated Figure 5 to highlight the consensus pathways that appear beyond the statistical noise, and moved the replicate permutation testing figures to the supplement.

Referee #6 (Remarks to the Author):

I thank the authors for their additional clarifications and adjustments, which have appropriately addressed my remaining concerns, and congratulate them on this impressive piece of work.

Thank you for the congratulatory remarks, and we agree this will be a great resource for the field.

We appreciate all the time and insight from all reviewers.

Reviewer Reports on the Third Revision:

Referee #2:

The authors have addressed my concerns. The claims have been adjusted, and additional comparison and descriptions have been included.

Minor:

- The described change to the Fig. 5 (removing p-values to avoid misinterpreting confidence, describing them in the legend) is not present in Fig. 5 or the corresponding legend.

Referee #5:

We thank the authors for responding to many of the points raised in our previous review. We are mostly satisfied with the current version of the analyses presented in the latest version of the manuscript, and will confine minor comments to the newly added panels from this round of revision.

We agree with the authors that the pathway enrichment analyses on the highly variable genes/analytes is a more natural and relevant thing to present compared to the violin plots of the deltaCVs (although we remain a little mystified as to why, during each successive round of review, the panels we have drawn attention to have been immediately removed from the manuscript and/or relegated to the supplement). The permutation testing and the empirical p-values for these analyses seem well-grounded, and it is interesting to take a look at the pathways that show up in Fig. 5g. (However, we object to identifying the pathways here, but not for example in Fig. 4d, as the “most recurrent pathways associated with spaceflight”, since these are rather the pathways whose members have the most variation across individuals after spaceflight, which seems to be a substantially different kind of analysis than implied by the above statement.) We also can't tell if the genes ranked by deltaCVs were ever used for pathway analysis, since all the 14 enrichments in Fig. 5g seem to be from FP1. We also note that the authors state in their response to Reviewer 2 that they have removed the p-values from the panels in Fig. 5a-f, but they seem to still appear there.

The comments from our last rounds of review have been primarily focused on trying to suss out what exactly can be learned from them at present. Many of these comments have tried to identify biological “through-lines” that are statistically significant (and with reasonable effect sizes), consistent with prior literature, recurrent across multiple assay modalities, and biologically reasonable and interpretable. We recognize that this is hard to do in a paper of this form, one that has many spin-out companion papers, and must be coordinated across multiple centers, labs, and authors. However, our fundamental concern about the impact, coherence, and interpretability of the findings presented in this manuscript remains. For example, the authors have added an updated pathway enrichment analysis to Fig. 5. However, the sentence introducing this new (and important!) panel in the text makes no attempt to connect it to the pathway enrichment analysis in Fig. 4, the motif enrichments from ED Fig. 7, etc. The authors addressed our comment about the biological significance of the m6a findings by doing some pathway enrichments and finding telomere related changes (although we can’t find any reference to this in ED Fig. 4, which also seems to be prematurely clipped), though in response to our question about the significance of the snoRNA changes, the authors just removed this sentence. Furthermore, Reviewer Fig. 3 seems surprising insofar as there does not seem to be any relationship between the Z-scores observed and the P values presented [they are all $\log(\text{Pval}) = 0$ for all observed delta z-scores? But even the $\log(\text{pVal} = .9) = -0.05$ or so. Are all these log-pvals zero because they are corrected? How are they corrected?].

At this late juncture of review, further investigation of these findings and analysis and rebuttals are not likely worthwhile (making these questions above rhetorical I suppose), as ultimately, we do believe that these data will likely be of use to the community to some degree, and the concrete biological claims being made in the paper are relatively minimal, and those that remain have at least some level of empirical support.

Author Rebuttals to Third Revision:

Referees' comments:

Reviewer 2:

The authors have addressed my concerns. The claims have been adjusted, and additional comparison and descriptions have been included.

We thank the reviewer for the positive response and appreciation of the new data and updated text.

Minor:

- The described change to the Figure 5 (removing p-values to avoid misinterpreting confidence, describing them in the legend) is not present in Figure 5 or the corresponding legend.

We have updated the Figure and the legend. Thank you.

Reviewer 5:

We thank the authors for responding to many of the points raised in our previous review. We are mostly satisfied with the current version of the analyses presented in the latest version of the manuscript, and will confine minor comments to the newly added panels from this round of revision.

We are glad the reviewer is satisfied with our updated manuscript, and we also highlight below more clarifications we have now added to update the paper.

We agree with the authors that the pathway enrichment analyses on the highly variable genes/analytes is a more natural and relevant thing to present compared to the violin plots of the deltaCVs (although we remain a little mystified as to why, during each successive round of review, the panels we have drawn attention to have been immediately removed from the manuscript and/or relegated to the supplement).

Indeed, we agree with the reviewer that the new enrichment analyses are the most relevant, and we also think that these will be the most interesting for comparisons in future missions. We have taken the reviewer comments into consideration at each round of review, and simplified figures, updated analyses, and generated the suggested new analyses and data from reviewers. This led to several of the figures changing, or becoming new tables, but these updates have all helped clarify and streamline the paper. Moreover, since some of the panels showed similar results, we followed the reviewers' advice and moved some panels to the supplemental figures.

The permutation testing and the empirical p-values for these analyses seem well-grounded, and it is interesting to take a look at the pathways that show up in 5g. (However, we object to identifying the pathways here, but not for example in 4d, as the “most recurrent pathways associated with spaceflight”, since these are rather the pathways whose members have the most variation across individuals after spaceflight, which seems to be a substantially different kind of analysis than implied by the above statement.) We also can’t tell if the genes ranked by deltaCVs were ever used for pathway analysis, since all the I4 enrichments in 5g seem to be from FP1.

We agree with the reviewer that the empirical values are the most robust, and we have also updated the text for Figure 4d to clarify that the figure represents the most variable pathways, which is distinct from Figure 5. We have also updated the text to clarify the details of the I4 enrichments. Also, the pathway enrichment tables for CV and delta CV values are now reported in Extended Data Table 7. Finally, the current Fig 5g contains most recurrent pathways for differential changes in FP1 across the both gene expression and protein-based ‘omics assays, and are then validated with the NASA Twins and JAXA CFE studies.

We also note that the authors state in their response to Reviewer 2 that they have removed the p-values from the panels in 5a-f, but they seem to still appear there.

This figure has been updated.

The comments from our last rounds of review have been primarily focused on trying to suss out what exactly can be learned from them at present. Many of these comments have tried to identify biological “through-lines” that are statistically significant (and with reasonable effect sizes), consistent with prior literature, recurrent across multiple assay modalities, and biologically reasonable and interpretable. We recognize that this is hard to do in a paper of this form, one that has many spin-out companion papers, and must be coordinated across multiple centers, labs, and authors. However, our fundamental concern about the impact, coherence, and interpretability of the findings presented in this manuscript remains. For example, the authors have added an updated pathway enrichment analysis to figure 5. However, the sentence introducing this new (and important!) panel in the text makes no attempt to connect it to the pathway enrichment analysis in figure 4, the motif enrichments from extended data figure 7, etc.

We originally connected the results in Figure 5 to the literature, especially in the discussion section, but we have now also updated the text to map these results to the other data and evidence of spaceflight responses in our paper, as well as results from related studies that can now be visualized in the SOMA data browser.

The authors addressed our comment about the biological significance of the m6a findings by doing some pathway enrichments and finding telomere related changes (although we can't find any reference to this in extended figure 4, which also seems to be prematurely clipped), though in response to our question about the significance of the snoRNA changes, the authors just removed this sentence.

The heatmap of Extended Data Figure 4 is across all pathways, and is a visual representation of the various pre-flight, in-flight, and post-flight dynamics that were significant and recurrent. The specific telomere results are in the Extended Data Tables, and this was noted in the main text before, but we have now made this more clear. Also, as requested by the reviewer, we still point to snoRNA changes in the text and figure, but limit any speculation. The results are still statistically significant, which is why we have kept them, but we also want to be conservative in our claims of biological significance and focus on the specific observations from the I4 mission and those which were validated in other missions (e.g. JAXA CFE study).

Furthermore, reviewer figure 3 seems surprising insofar as there does not seem to be any relationship between the Z-scores observed and the Pvalues presented [they are all $\log(\text{Pval}) = 0$ for all observed delta z-scores? But even the $\log(\text{pVal} = .9) = -0.05$ or so. Are all these log-pvals zero because they are corrected? How are they corrected?]

This figure was made in response to the reviewer request about the Fos/Jun motif enrichment in the 10x single-cell Multiome Negative Controls. The y-axis represents $-\log(\text{adjusted p-value})$, and all the adjusted p-values for the TFs were 1. In that case, $-\log(1) = 0$. We then did a Wilcoxon Rank sum test for a statistical test, with a Bonferroni correction based on the total number of TFs in the dataset, which confirmed that this pathway is not driven by technical noise of the assay.

At this late juncture of review, further investigation of these findings and analysis and rebuttals are not likely worthwhile (making these questions above rhetorical I suppose), as ultimately, we do believe that these data will likely be of use to the community to some degree, and the concrete biological claims being made in the paper are relatively minimal, and those that remain have at least some level of empirical support.

We agree with the reviewer that the data and this paper will be of use to the community, and also serve as a large resource (both data and samples) and continual source of novel hypotheses. We have updated the manuscript to focus on those claims with the most empirical support, notably across several missions, modalities, and assays, as well as the now updated links to murine, human missions, GeneLab, and other data sets.